# FINE-TUNING BEHAVIORAL CLONING POLICIES WITH PREFERENCE-BASED REINFORCEMENT LEARNING

**Maël Macuglia[1], Paul Friedrich[1,2], Giorgia Ramponi[1,2]**
[1] Department of Informatics, University of Zurich, Switzerland
[2] ETH AI Center, Zurich, Switzerland
`mael.macuglia@icloud.com`, `{paul.friedrich, giorgia.ramponi}@uzh.ch`

## ABSTRACT

Deploying reinforcement learning (RL) in robotics, industry, and health care is blocked by two obstacles: the difficulty of specifying accurate rewards and the risk of unsafe, data-hungry exploration. We address this by proposing a two-stage framework that first learns a safe initial policy from a reward-free dataset of expert demonstrations, then fine-tunes it online using preference-based human feedback. We provide the first principled analysis of this offline-to-online approach and introduce BRIDGE, a unified algorithm that integrates both signals via an uncertainty-weighted objective. We derive regret bounds that shrink with the number of offline demonstrations, explicitly connecting the quantity of offline data to online sample efficiency. We validate BRIDGE in discrete and continuous control MuJoCo environments, showing it achieves lower regret than both standalone behavioral cloning and online preference-based RL. Our work establishes a theoretical foundation for designing more sample-efficient interactive agents.

## 1 INTRODUCTION

Deploying reinforcement learning (RL) (Sutton & Barto, 2018) on physical robots, industrial processes, or in healthcare remains notoriously difficult for two reasons. First, exploration is both risky and data-hungry (Dulac-Arnold et al., 2019): A policy that begins from scratch can damage hardware, or user trust, long before gathering enough experience to learn. Second, reward mis-specification: even experienced domain experts often find it hard to translate informal task goals into a correct and safe numerical reward signal (Leike et al., 2018).

A promising solution addresses both challenges simultaneously: It combines reward-free expert demonstrations with online preference-based feedback, allowing practitioners to leverage safe imitation learning while enabling corrective refinement through simple comparative judgments. This hybrid approach has achieved remarkable empirical success across domains. It is the core technique behind modern dialogue agents like ChatGPT, which is first trained to imitate curated demonstrations of desirable responses and then fine-tuned via RL from human feedback (RLHF) (Ouyang et al., 2022). Similar approaches have achieved near-expert performance in complex games like Atari and control tasks like MuJoCo (Christiano et al., 2017) and enabled safe real-world robot manipulation by ranking tele-operated clips before on-hardware fine-tuning (Brown et al., 2020). Our reward-free setting is distinct from a related line of successful work that also pre-trains offline but assumes access to a ground-truth reward signal for online fine-tuning (Nair et al., 2020; Kostrikov et al., 2022; Tang et al., 2025; Park et al., 2024; Tirinzoni et al., 2025).

However, despite its widespread practical success, the theoretical foundations of offline-to-online preference learning remain unexplored. Existing theory analyzes either imitation learning or preference-based RL in isolation. This leaves fundamental questions unanswered: How exactly do offline demonstrations improve online preference learning? What are the precise trade-offs between the quantity of offline data and the number of online queries? When is this combination provably better than either approach alone? This theoretical gap prevents a principled understanding of the method's limits and leaves practitioners without formal guidance for designing such systems.

---

Code available at `https://github.com/pfriedric/bridge`.

We provide the first theoretical analysis of this empirically important paradigm. We formalize the "offline imitation + online preference fine-tuning" setting and develop rigorous regret bounds that quantify how offline expert data reduces online learning complexity. We introduce a new algorithm, **B**ounded **R**egret with **I**mitation **D**ata and **G**uided **E**xploration (**BRIDGE**) that achieves the predicted theoretical benefits in experiments on discrete and, extending beyond our theory, continuous control tasks. Our key contributions are:

- **The first theoretical framework for offline-to-online preference learning.** We provide the first rigorous regret analysis for this approach. Our framework (Theorem 4.2) uses the Hellinger distance between trajectory distributions to construct confidence sets whose radii, $O(1/\sqrt{n})$, directly connect the quantity of offline data $n$ to online learning efficiency.

- **A regret bound showing offline data reduces online regret.** We prove that our algorithm, BRIDGE (Algorithm 1) achieves an optimal $\sqrt{T}$ regret dependence on the online horizon $T$, while explicitly showing how offline demonstrations improve online performance (Theorem 4.1). Our bound formally shows that as the number of offline demonstrations $n \to \infty$, the online regret approaches zero, theoretically validating that high-quality offline data dramatically improves preference learning efficiency.

We review related work on the offline-to-online paradigm in Section 2 and formalize our problem setting and regret measure in Section 3. We then present our algorithm, BRIDGE, along with its theoretical regret bounds in Section 4. Finally, we validate our theory with experiments on discrete and continuous control tasks in Section 5. We show detailed experiments in Appendix A and all proofs in Appendices B to F.

## 2 RELATED WORK

**Imitation Learning.** Behavioral Cloning (BC), pioneered by road-following systems like ALVINN (Pomerleau, 1988), learns policies from expert data via supervised learning. Recent advances in theory e.g. establish horizon-free sample complexity bounds for BC (Foster et al., 2024). The DAGGER algorithm mitigates *covariate shift* at deployment time (when facing states outside the training data), with iterative expert corrections, achieving no-regret guarantees (Ross et al., 2011). However, its reliance on a constantly available expert is often impractical. Our approach inherits BC's simplicity but replaces this online expert with preference-based refinement.

**Hybrid offline-to-online RL.** Learning entirely online from a cold start is often sample-inefficient and (initially) unsafe. Our work fits into the hybrid paradigm, which avoids this by using offline data to warm-start a policy before online refinement, with early contributions including model-based algorithms (Ross & Bagnell, 2012). This area has seen significant empirical progress (Rajeswaran et al., 2017; Hester et al., 2018; Nair et al., 2018; Vecerik et al., 2017; Lee et al., 2022; Ball et al., 2023) and theoretical advances in statistical approaches to efficiently combine offline and online datasets (Song et al., 2023; Wagenmaker & Pacchiano, 2023; Tang et al., 2023). However, this entire line of work is fundamentally different from ours as it *assumes access to a ground-truth reward function* during online fine-tuning. For instance, prior theory shows that for non-expert offline data, pre-training offers no statistical improvement in this reward-based setting (Xie et al., 2021). We show, in contrast, that offline *expert* data provides a provable statistical advantage when combined with reward-free, online *preference-based* feedback. Appendix A.4 provides a comparison of BRIDGE with modern preference-based fine-tuning approaches.

In summary, prior imitation-only approaches lack robustness outside the demonstration manifold, while existing offline-to-online methods demand ground-truth rewards. Our work bridges these gaps by (i) proving that expert demonstrations plus a modest preference-query budget yield sharper regret bounds, and (ii) showing empirically that preference-guided exploration corrects for blind spots with far fewer risky interactions than pure online RL.

## 3 PROBLEM FORMULATION

We address the challenge of learning optimal policies by combining information from two complementary sources: offline expert demonstrations and online preference feedback. In this hybrid

learning paradigm, we first leverage a dataset of trajectories collected from an expert policy to establish strong priors over the policy space. Then, we strategically utilize these priors to guide an online preference-based learning process, where an expert provides binary feedback comparing pairs of trajectories. This framework enables us to efficiently narrow the search space using offline demonstrations while refining our understanding of the expert's underlying preference model through targeted online queries. We aim to quantify how knowledge from offline demonstrations translates to improved regret bounds in the online preference learning phase.

**Finite MDP setting (reward-free).** Consider a finite-horizon Markov Decision Process (MDP) defined by the tuple $\mathcal{M} = (\mathcal{S}, \mathcal{A}, P, H)$, where $\mathcal{S}$ is a finite state space, $\mathcal{A}$ is a finite action space, $H \in \mathbb{N}$ is the horizon length, and $P = \{P_h\}_{h \in [H]}$ represents the time-dependent transition dynamics, with $P_h(\cdot|s, a)$ denoting the probability distribution over next states given state-action pair $(s, a)$ at step $h$. A policy $\pi = \{\pi_h\}_{h \in [H]}$ consists of a collection of mappings $\pi_h : \mathcal{S} \to \Delta(\mathcal{A})$, where $\Delta(\mathcal{A})$ is the probability simplex over actions. A trajectory $\tau = \{(s_h, a_h)\}_{h \in [H]}$ is a sequence of state-action pairs generated by executing a policy $\pi$ in the environment following dynamics $P$. We denote the space of all possible trajectories of fixed length $H$ as $\mathcal{T}$. We assume the trajectories have a continuous distribution with respect to the counting or Lebesgue measure. We will write $\mathbb{P}_P^\pi$ for the density function of the trajectory distribution induced by policy $\pi$ and dynamics $P$.

**Offline demonstrations.** We assume access to an *offline dataset* $\mathbb{D}_n^H = \{\tau_i\}_{i \in [n]}$ consisting of $n$ independent trajectories of length $H$, where each $\tau_i \sim \mathbb{P}_{P^*}^{\pi^*}$. This is an imitation learning setting where trajectories are generated by an expert policy $\pi^*$ interacting with the true environment dynamics $P^*$.

**Online preference queries.** We formalize preference-based learning through feature embeddings and a probabilistic preference model (Christiano et al., 2017; Saha et al., 2023). We assume the existence of a trajectory embedding function $\phi : \mathcal{T} \to \mathbb{R}^d$ that is known to the learner. The offline demonstrations capture raw expert behavior, while the known embedding function $\phi$ provides the necessary structure for efficient online preference learning. The trajectory embedding function $\phi$ serves a critical purpose in our framework by enabling meaningful preference comparisons that would be difficult to perform on raw trajectories. This embedding approach provides a versatile framework that can accommodate various types of trajectory information. The flexibility of this representation allows our method to adapt to different domains and preference structures without changing the underlying learning algorithm. We define the policy embedding as the expected feature representation: $\phi^P(\pi) = \mathbb{E}_{\tau \sim \mathbb{P}_P^\pi}[\phi(\tau)]$.

We adopt two commonly used assumptions, bounded trajectory embeddings (Saha et al., 2023; Parker-Holder et al., 2020b) and bounded weight vectors (Filippi et al., 2010; Faury et al., 2020).

**Assumption 1.** *We require (i) **bounded features:** $\|\phi(\tau)\|_2 \leq B$ for all $\tau \in \mathcal{T}$ and some known $B < \infty$, and (ii) **bounded weights:** $\mathbf{w}^* \in \{v \in \mathbb{R}^d : \|v\|_2 \leq W\}$ for a known $W < \infty$.*

We measure the degree of non-linearity of the sigmoid $\sigma(x) = (1 + e^{-x})^{-1}$ over the parameter space (where $\sigma'$ is the first derivative of $\sigma$) with $\kappa := \sup_{\boldsymbol{x} \in \mathcal{B}_B(0), \mathbf{w} \in \mathcal{B}_W(0)} \frac{1}{\sigma'(\mathbf{w}^\top \boldsymbol{x})}$.

We use a Bradley-Terry model for the preference feedback. Given two trajectories $\tau^1$ and $\tau^2$, the binary preference outcome $o_{1,2} \in \{0, 1\}$ is modeled as a Bernoulli random variable, indicating with $o_{1,2} = 1$ that $\tau^1$ is preferred over $\tau^2$, and vice-versa with $o_{1,2} = 0$:

$$\mathbb{P}(\tau^1 \succ \tau^2) = \mathbb{P}(o_{1,2} = 1|\tau^1, \tau^2) = \sigma(\langle \phi(\tau^1) - \phi(\tau^2), \mathbf{w}^* \rangle).$$

This corresponds to a latent utility model where the inner product $\langle \phi(\tau), \mathbf{w}^* \rangle$ represents the utility of trajectory $\tau$. We can extend this to policies, defining $\mathbb{P}(\pi^1 \succ \pi^2) = \sigma(\langle \phi(\pi^1) - \phi(\pi^2), \mathbf{w}^* \rangle)$. This represents an expected preference over the distribution of trajectories, and captures the average preference when comparing behaviors induced by different policies. From this model, we derive a score function for trajectories $s(\tau) = \langle \phi(\tau), \mathbf{w}^* \rangle$ and extend it to policies as $s^P(\pi) = \mathbb{E}_{\tau \sim \mathbb{P}_P^\pi}[s(\tau)]$.

**Offline estimation quality.** For the offline phase, we measure the quality of estimation using distributional distance metrics in the space of trajectory distributions. Specifically, we will construct confidence sets in the form of Hellinger balls around our estimated density policy and dynamics. Notably, the Hellinger distance relates directly to the $L^2$ norm between square-root densities, enabling a geometric interpretation of our confidence sets as Euclidean balls in the space of density embeddings, with computational advantages over alternative divergences. The precise construction of these confidence sets and their properties is shown in Section 4.

**Online regret.** We quantify our online learning phase's performance through regret measurement. In each round $t \in [T]$ of online learning, the agent selects policies $\pi_t^1$ and $\pi_t^2$, receives binary preference feedback $o_t \in \{0, 1\}$, and accumulates regret measured against the optimal policy. We specifically use the *pseudo-regret* with respect to the policy class $\Pi$ as in Saha et al. (2023):[1]

$$R_T^{\mathrm{psr}} := \max_{\pi \in \Pi} \sum_{t=1}^{T} \frac{[2\phi^{P^*}(\pi) - \phi^{P^*}(\pi_t^1) - \phi^{P^*}(\pi_t^2)]^\top \mathbf{w}^*}{2} = \sum_{t=1}^{T} \frac{2s^{P^*}(\pi^*) - (s^{P^*}(\pi_t^1) + s^{P^*}(\pi_t^2))}{2},$$

where $\pi^* := \arg\max_{\pi \in \Pi} s(\pi)$. All our performance guarantees will be expressed in terms of the MDP parameters (state space size $|\mathcal{S}|$, action space size $|\mathcal{A}|$, horizon length $H$), offline data quantity $n$, online interaction rounds $T$, and confidence level $\delta$ of the offline estimation - establishing a direct connection between offline data quality and online learning efficiency.

**Notation.** We denote $[H] = \{1, \ldots, H\}$ for $H \in \mathbb{N}$. For probability distributions $P, Q$, $H^2(P, Q)$ is the squared Hellinger distance and $\mathrm{TV}(P, Q)$ the total variation distance. We denote as $\mathcal{B}_R(0) := \{x \in \mathbb{R}^d : \|x\|_2 \leq R\}$ the Euclidean ball of radius $R$, and define $x^{\otimes 2} := xx^\top$ as the outer product.

# 4 BRIDGING OFFLINE BEHAVIORAL CLONING AND ONLINE PREFERENCE-BASED FEEDBACK

Our approach BRIDGE uses offline expert demonstrations to improve the efficiency of online preference learning. The core idea is to use the offline expert data to construct a set in policy space that contains the expert with high confidence, drastically shrinking the search space for the subsequent online learning. This process contains three steps:

1. **Offline imitation:** We first use the offline dataset to learn an initial policy via Behavioral Cloning (BC), and a transition model estimate via maximum likelihood estimation (MLE).

2. **Confidence set construction:** Next, we construct a confidence set $\Pi_{1-\delta}^{\mathrm{offline}}$ centered on the BC policy in trajectory distribution space. We define the set as a ball in the space of trajectory distributions using the Hellinger distance. This provides a clean geometric interpretation, namely a ball defined by a single scalar quantity (its radius), and makes the theoretical analysis tractable. We prove that the radius shrinks at a rate of $\mathcal{O}(1/\sqrt{n})$, where $n$ is the number of offline expert demonstrations. More offline data directly translates to a tighter ball and a smaller, more focused search space for online learning.

3. **Constrained online learning:** Finally, we perform online preference-based RL, but with exploration constrained to policies that lie within the pre-computed Hellinger ball. This prevents the agent from exploring highly suboptimal or unsafe regions of the policy space.

This framework is illustrated in Figure 1, implemented in detail in Algorithm 1, experimentally verified in Section 5 and supported by our main theoretical result. The following theorem formalizes the intuition that more offline data improves online performance by providing a high-probability regret bound that explicitly depends on $n$. The full proof is found in Theorem E.1.

**Theorem 4.1** (Main result: Offline data reduces online regret). *Let $n$ be the number of offline demonstrations from an expert policy satisfying Assumption 3, where $\gamma_{\min} > 0$ is the minimum nonzero visitation probability under the expert policy's distribution. Then, with probability at least $1 - \delta$, the regret of* BRIDGE *is bounded by*

$$R_T \leq \tilde{\mathcal{O}} \left( \sqrt{T} \cdot \sqrt{\log\left(1 + \frac{T}{n}\right)} + \frac{\sqrt{T}}{\sqrt{n \cdot \gamma_{\min}}} \right). \tag{1}$$

This regret bound represents our core theoretical contribution. While the regret's scaling in $\sqrt{T}$ matches Saha et al. (2023), it now also depends inversely on the number of offline demonstrations $n$. **Crucially, the regret vanishes as $n \to \infty$, for a fixed** $T$. This formally captures our claim that high-quality offline data can arbitrarily reduce the complexity of online preference-based learning.

---

[1]They show equivalence of the standard preference-based formulation up to constant factors, if $B, W \leq 1$.

---

**Algorithm 1 BRIDGE:** Bounded Regret with Imitation Data and Guided Exploration

1: **Input: offline dataset $\mathbb{D}_n^H$, no. of iterations** $T$
2: Estimate $\pi^{\text{BC}}$ and $\hat{P}$ via MLE (Eqs. (2) & (3)) and compute confidence set $\Pi_{1-\delta}^{\text{offline}}$ (Theorem 4.2)
3: Initialize model $\hat{P}_1 \leftarrow \hat{P}$, data matrix $\overline{\boldsymbol{V}}_1 \leftarrow \lambda \boldsymbol{I}_d$
4: **for** $t = 1, \ldots, T$ **do**
5:     Compute $\mathbf{w}_t^{\text{proj}}$ via constrained MLE (Equation (4))
6:     Define policy set $\Pi_t$ based on $\Pi_{1-\delta}^{\text{offline}}$ and $\mathbf{w}_t^{\text{proj}}$ (Lemma 4.5)
7:     $(\pi_t^1, \pi_t^2) \leftarrow \arg\max_{\pi^1, \pi^2 \in \Pi_t} \{ \gamma_t \cdot \|\phi^{\hat{P}_t}(\pi^1) - \phi^{\hat{P}_t}(\pi^2)\|_{\overline{\boldsymbol{V}}_t^{-1}}$
                    $+ \hat{B}_t(\pi^1, 2WB, \delta^{\text{online}}) + \hat{B}_t(\pi^2, 2WB, \delta^{\text{online}})\}$
8:     Sample trajectories $\tau_t^1 \sim \mathbb{P}_{\hat{P}_t}^{\pi_t^1}, \tau_t^2 \sim \mathbb{P}_{\hat{P}_t}^{\pi_t^2}$ and obtain preference $o_t = \mathbb{I}(\tau_t^1 \succ \tau_t^2)$
9:     Update matrix $\overline{\boldsymbol{V}}_{t+1} \leftarrow \overline{\boldsymbol{V}}_t + (\phi^{\hat{P}_t}(\pi_t^1) - \phi^{\hat{P}_t}(\pi_t^2))^{\otimes 2}$ and model $\hat{P}_{t+1}$
10: **end for**
11: **return** Best policy from $\Pi_T$ using final weight estimate $\mathbf{w}_t^{\text{proj}}$

---

The remainder of this section substantiates these claims. We first detail the construction of the Hellinger-based offline confidence set in Section 4.1 and then describe how we link it to the online preference-learning algorithm that operates within this set in Sections 4.2 and 4.3. Section 4.4 gives a proof sketch, and Section 4.5 discusses limitations and implications of BRIDGE's assumptions.

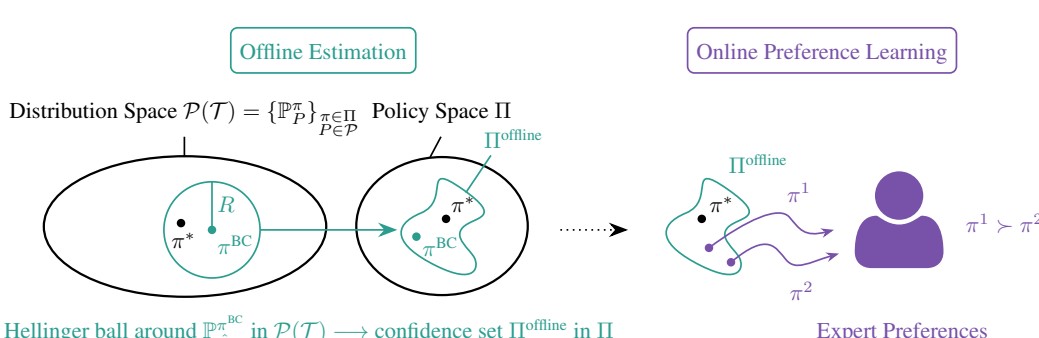

Figure 1: Overview of the BRIDGE framework. Offline estimation derives estimators $\pi^{\text{BC}}$ and $\hat{P}$ using the dataset $\mathbb{D}_n^H$ and constructs a confidence set in trajectory distribution space $\mathcal{P}(\mathcal{T})$ as a Hellinger ball (left), which translates to the offline policy confidence set $\Pi^{\text{offline}}$ in policy space $\Pi$ likely to contain $\pi^*$ (middle). The confidence set $\Pi^{\text{offline}}$ is then used to constrain the online preference learning phase (right), where policies are sampled from within this set and presented to the expert for preference feedback.

## 4.1 STAGE 1: OFFLINE CONFIDENCE SET CONSTRUCTION

The goal of our offline phase is to use the expert dataset $\mathbb{D}_n^H$ to construct a policy confidence set $\Pi_{1-\delta}^{\text{offline}}$ that is guaranteed to contain the expert policy $\pi^*$ with high probability $(1 - \delta)$. The key result of this stage is a computable confidence set whose radius shrinks with increasing amounts of offline data, which we formalize in the following theorem. The proofs of both this theorem and Lemma 4.4 are shown in Appendix C.4.

**Theorem 4.2** (Offline confidence set radius). *Define constants* $\alpha := \sqrt{4 \cdot |\mathcal{S}| \cdot \log(|\mathcal{A}| \cdot 2/\delta)}$ *and* $\beta := \sqrt{4 \cdot |\mathcal{S}|^2 \cdot |\mathcal{A}| \cdot \log(nH \cdot 2/\delta)}$. *Under Assumption 3, the policy set*

$$\Pi_{1-\delta}^{\text{offline}} := \left\{ \pi \mid \sqrt{H^2(\mathbb{P}_{\hat{P}}^{\pi}, \mathbb{P}_{\hat{P}}^{\pi^{\text{BC}}})} \leq \text{Radius} \right\}$$

*is a confidence set of level* $1 - \delta$ *containing* $\pi^*$ *with probability at least* $1 - \delta$, *where*

$$\text{Radius} = \frac{\alpha}{\sqrt{n}} + \frac{\beta}{\sqrt{n}} \cdot \left( 1 + \sqrt{H \cdot \left( 1 + \frac{2\alpha}{\gamma_{\min} \cdot \sqrt{n}} \right)} \right).$$

This result provides the fundamental connection between offline data and online learning efficiency: the confidence set radius scales as $\mathcal{O}(1/\sqrt{n})$ with the offline sample size $n$. As we collect more expert demonstrations, the confidence set shrinks, constraining the online policy search space more tightly. Since our online regret bounds will directly depend on the size of this confidence set, this establishes a quantifiable trade-off between offline data collection and online preference query efficiency, a key contribution of our work. We now detail the components required to construct this set. Relevant corollaries and their proofs are presented in Appendix C.

**Policy and model estimation.** To construct the confidence set, we first obtain estimators for the policy and transition dynamics from the offline data, assuming realizability. We apply maximum likelihood estimation (MLE) on the expert trajectories $\mathbb{D}_n^H$ to learn the BC policy estimator $\pi^{\text{BC}}$ and the transition model estimator $\hat{P}$.

**Assumption 2** (Realizability). *The optimal policy and true transition function belong to their respective function classes: $\pi^* \in \Pi$ and $P^* \in \mathcal{P}$.*

The *BC estimator* $\pi^{\text{BC}}$ is found via log-loss Behavioral Cloning (BC), and the *MLE transition estimator* $\hat{P}$ is found similarly using Maximum Likelihood Estimation (MLE):

$$\pi^{\text{BC}} = \arg\max_{\pi \in \Pi} \sum_{i \in [n]} \sum_{h \in [H]} \log(\pi_h(a_h^i | s_h^i)), \tag{2}$$

$$\hat{P} = \arg\max_{P \in \mathcal{P}} \sum_{i \in [n]} \sum_{h \in [H]} \left( \log[P(s_{h+1}^i | s_h^i, a_h^i)] \right). \tag{3}$$

We provide concentration bounds for these estimators in Appendix C.3, which characterize their error in terms of the Hellinger distance.[2]

**Bounding concentrability.** A key challenge in using these estimators is that our bounds depend on the unknown true dynamics $P^*$. To create a computable confidence set, we must eliminate this dependency. We do so by bounding the *concentrability coefficient*, which measures the maximum divergence between the state-action distributions of an estimated policy $\pi^{\text{BC}}$ and the expert $\pi^*$. Commonly encountered in offline RL literature (Chen & Jiang, 2019), it is defined as:

$$C(\pi^{\text{BC}}, \pi^*) = \sup_{t \in [H]} \sup_{(s,a) \in \mathcal{S} \times \mathcal{A}: d_{P^*}^{\pi^*,t}(s,a) > 0} \frac{d_{P^*}^{\pi^{\text{BC}},t}(s,a)}{d_{P^*}^{\pi^*,t}(s,a)}.$$

To bound this quantity without requiring broad data coverage, we instead make a mild assumption on the expert's policy structure.

**Assumption 3** (Expert policy concentration). *The expert policy $\pi^*$ has a minimum visitation probability $\gamma_{\min} > 0$ for state-actions it visits, i.e., $\min_{(s,a,t): d_{P^*}^{\pi^*,t}(s,a) > 0} d_{P^*}^{\pi^*,t}(s,a) \geq \gamma_{\min}$.*

Intuitively, this assumption characterizes the expert's intrinsic behavior. A smaller $\gamma_{\min}$ corresponds to a more specialized expert with sharp preferences for certain state-actions, while a larger value implies a more uniform visitation pattern. This contrasts a standard assumption in offline RL of a minimum dataset coverage across all state-actions (Levine et al., 2020; Chen & Jiang, 2019).

We can now bound the concentrability coefficient using only the Hellinger error $R$ of our policy estimator and the expert's concentration pattern $\gamma_{\min}$. For the proof, see Appendix C.3.3.

**Lemma 4.3** (Concentrability coefficient bound). *Under Assumption 3, for a policy estimator $\pi^{BC}$ satisfying $H^2(\mathbb{P}_{P^*}^{\pi^{BC}}, \mathbb{P}_{P^*}^{\pi^*}) \leq R$, the concentration coefficient is bounded by*

$$C(\pi^{BC}, \pi^*) \leq 1 + \frac{2\sqrt{R}}{\gamma_{\min}}.$$

---

[2] We present theoretical results for the standard setting of tabular state and action spaces, and finite policy classes $\Pi$. While we leave a full theoretical extension of our framework to the continuous setting for future work, we show experiments and an implementation for continuous MDPs in Section 5 and Appendix A.

**Final confidence set construction.** By combining our concentration results for $\pi^{\mathrm{BC}}$ and $\hat{P}$ (Corollaries C.5 and C.9) with the deterministic bound on concentrability, we can construct the final offline confidence set. The following lemma provides the general form, which uses only quantities computable from our offline data and estimators, leading directly to the explicit radius in Theorem 4.2.

**Lemma 4.4** (Offline policy confidence set). *Let $R_1(\delta/2)$ and $R_2(\delta/2)$ be high-probability upper bounds on the Hellinger estimation errors for the policy and transition model, such that with probability at least $1 - \delta/2$ each:*

$$H^2(\mathbb{P}_{P^*}^{\pi^{BC}}, \mathbb{P}_{P^*}^{\pi^*}) \leq R_1(\delta/2) \quad and \quad H^2(\mathbb{P}_{\hat{P}}^{\pi^*}, \mathbb{P}_{P^*}^{\pi^*}) \leq R_2(\delta/2).$$

*Then, under Assumption 3, the offline confidence policy set*

$$\Pi_{1-\delta}^{offline} := \left\{ \pi \in \Pi \ \Big| \ \sqrt{H^2(\mathbb{P}_{\hat{P}}^{\pi}, \mathbb{P}_{\hat{P}}^{\pi^{BC}})} \leq \sqrt{R_1} + \sqrt{R_2} \cdot \left( 1 + \sqrt{H \cdot \left( 1 + \frac{2\sqrt{R_1}}{\gamma_{\min}} \right)} \right) \right\}$$

*contains the expert policy $\pi^*$ with probability at least $1 - \delta$.*

## 4.2 STAGE 2: CONSTRAINED ONLINE PREFERENCE LEARNING

In the online stage, our goal is to efficiently learn the true preference reward vector $\mathbf{w}^*$ by leveraging the confidence set $\Pi^{\mathrm{offline}}$ constructed previously. Our approach follows the generalized linear model (GLM) framework for preference-based RL of Saha et al. (2023), who themselves adapt GLMs from parametric bandits (Filippi et al., 2010; Faury et al., 2020). We first summarize its core components, and then show our novel adaptations. All derivations and proofs are provided in Appendix D.

**Preference-based online learning framework.** The online algorithm learns a preference vector $\mathbf{w}^*$ by iteratively presenting pairs of trajectories $(\tau^1, \tau^2)$ to an expert and receiving a binary preference. At each step $t$, the method computes a regularized maximum likelihood estimate $\mathbf{w}_t^{\mathrm{MLE}}$, based on past preference queries. Since this initial estimate may not satisfy our boundedness Assumption 1, it is projected onto a valid set. This projection uses the empirical data matrix

$$\boldsymbol{V}_t = \kappa\lambda\boldsymbol{I}_d + \sum_{\ell=1}^{t-1}(\phi(\tau_\ell^1) - \phi(\tau_\ell^2))^{\otimes 2},$$

which captures the information gathered from past queries, and a transformation function given by $g(\mathbf{w}) = \sum_{l=1}^{t-1} \sigma\left(\langle\phi(\tau_l^1) - \phi(\tau_l^2), \mathbf{w}\rangle\right)\left(\phi(\tau_l^1) - \phi(\tau_l^2)\right) + \lambda\mathbf{w}$. The projected estimate $\mathbf{w}_t^{\mathrm{proj}}$ is then found by solving the following optimization problem:

$$\mathbf{w}_t^{\mathrm{proj}} = \arg\min_{\mathbf{w}\in\mathcal{B}_W(0)} \|g_t(\mathbf{w}) - g_t(\mathbf{w}_t^{\mathrm{MLE}})\|_{\boldsymbol{V}_t^{-1}}. \tag{4}$$

The matrix $\boldsymbol{V}_t$ serves a dual role: It defines a confidence ellipsoid around $\mathbf{w}_t^{\mathrm{proj}}$ that contains $\mathbf{w}^*$ with high probability, and it guides exploration towards directions of highest uncertainty via its Mahalanobis norm. To obtain our distributional guarantees, we need to relate this empirical matrix, built from single trajectory realizations, to its expected counterpart $\overline{\boldsymbol{V}}_t$, which averages over the sampling randomness of the trajectories: $\overline{\boldsymbol{V}}_t = \kappa\lambda\boldsymbol{I}_d + \sum_{\ell=1}^{t-1}(\phi^{\hat{P}_t}(\pi_\ell^1) - \phi^{\hat{P}_t}(\pi_\ell^2))^{\otimes 2}$. Relating these matrices allows our bounds to account for both model uncertainty and sampling variance.

Our work introduces two important modifications to this framework to integrate the offline information and guide online exploration.

**1. Offline-online transition model integration.** We improve the online transition model estimator's sample efficiency by pooling offline and online data. We initialize it using the offline MLE estimator from Equation (3), which in the tabular setting is a simple count-based estimator. We then update it at each step $t$ using the combined counts:

$$\hat{P}_t(s'|s,a) = \frac{N_{\mathrm{off}}(s',s,a) + N_t(s',s,a)}{N_{\mathrm{off}}(s,a) + N_t(s,a)}.$$

Consequently, to account for the uncertainty in the estimator, we adapt the exploration bonus from Chatterji et al. (2021) to use these combined counts:

$$\hat{B}_t(\pi,\eta,\delta) = \mathbb{E}_{\tau\sim\mathbb{P}_{\hat{P}_t}^{\pi}}\left[\sum_{h\in[H]}\min\left(2\eta, 4\eta\sqrt{\frac{U_h}{N_{\mathrm{off}}(s_h,a_h) + N_t(s_h,a_h)}}\right)\right],$$

where $U_h$ is a logarithmic term dependent on the state-action space size and the confidence $\delta$ (Lemma E.3). This bonus $\hat{B}_t$ encourages exploring parts of the state-action space where our combined offline-online transition model is less certain.

**2. Constrained and uncertainty-guided policy selection.** We constrain all online exploration to the offline confidence set $\Pi_{1-\delta}^{\text{offline}}$. Within this safe set, the algorithm actively seeks to reduce uncertainty. At each step (line 7 of Algorithm 1), it selects the pair of policies $(\pi_t^1, \pi_t^2)$ that maximizes a total exploration objective. This objective combines the uncertainty in the preference model (the $\|\cdot\|_{\overline{V}_t^{-1}}$ term) with the uncertainty in the transition model (the $\hat{B}_t$ bonus). The following lemma formalizes the adaptive online confidence set $\Pi_t$ from which we sample, guaranteeing that this exploration strategy remains sound. The confidence radius multiplier $\gamma_t$ is defined in Appendix D.4.

**Lemma 4.5** (Online policy confidence set). *With probability at least $1 - \delta^{online}$, the optimal policy $\pi^*$ is contained in the set $\Pi_t \subseteq \Pi_{1-\delta}^{offline}$ for all $t \in [T]$, where $\delta' = \frac{\delta^{online}}{2|\mathcal{A}|^{|\mathcal{S}|}}$ and $\Pi_t$ is defined as*

$$\Pi_t := \Big\{ \pi \in \Pi_{1-\delta}^{offline} \ \Big| \ \forall \pi' \in \Pi_{1-\delta}^{offline} :$$
$$\big\langle \phi^{\hat{P}_t}(\pi) - \phi^{\hat{P}_t}(\pi'), \mathbf{w}_t^{proj} \big\rangle + \gamma_t \cdot \|\phi^{\hat{P}_t}(\pi) - \phi^{\hat{P}_t}(\pi')\|_{\overline{V}_t^{-1}}$$
$$+ \hat{B}_t(\pi, 2WB, \delta') + \hat{B}_t(\pi', 2WB, \delta') \geq 0 \Big\}. \quad (\gamma_t \leftarrow Lemma \ D.7)$$

### 4.3 THE BRIDGE: HOW OFFLINE DATA REDUCES ONLINE REGRET

We now explain the core theoretical mechanism connecting the quality of our offline estimate to the final online regret (cf. Lemmas B.4 and E.6).

Regret bounds in online learning are fundamentally tied to controlling the cumulative exploration variance of queried policy pairs, $\text{tr}(\overline{V}_t) = \sum_{t' < t} \|\phi(\pi_1^{t'}) - \phi(\pi_2^{t'})\|_2^2$. Standard analyses (Lattimore & Szepesvári, 2020) rely on worst-case bounds. Saha et al. (2023), using the bounded feature assumption 1, obtain a worst-case uniform bound on the feature differences $\|\phi(\pi_1) - \phi(\pi_2)\|_2 \leq 2B$, leading to a total variance that scales linearly with the online horizon $T$.

Our key insight is that by constraining policy selection to $\pi \in \Pi_{1-\delta}^{\text{offline}}$ (line 7 of Algorithm 1), we establish a tighter, offline data-dependent bound. The properties of our Hellinger-based confidence set (Lemma 4.4) and the connection between Hellinger distances and feature distances (Appendix F.1) allow us to prove that for any pair of policies $\pi_1, \pi_2 \in \Pi_{1-\delta}^{\text{offline}}$:

$$\|\phi^{\hat{P}}(\pi_1) - \phi^{\hat{P}}(\pi_2)\|_2 \leq \frac{4\sqrt{2}B}{\sqrt{n}}.$$

This result directly injects the offline data size $n$ into the online variance term at each step. It is the central mechanism by which the offline confidence set's $O(1/\sqrt{n})$ radius improves our final regret bound (Theorem 4.1), which formally establishes the trade-off between offline data collection and online query efficiency.

### 4.4 PROOF SKETCH OF OUR MAIN REGRET BOUND (THEOREM 4.1)

We first derive the offline confidence set $\Pi_{1-\delta}^{\text{offline}}$ in Theorem 4.2 / Lemma B.1, proving that $\pi^* \in \Pi_{1-\delta}^{\text{offline}}$ with high probability. For each online iteration $t$, in Lemma D.1 we bound the distance between the learned parameter $\mathbf{w}_t^{\text{proj}}$ and the optimal parameter $\mathbf{w}^*$ under the norm induced by $V_t$, in Lemma D.2 we relate the empirical data matrix $V_t$ to the expected data matrix $\overline{V}_t$, and combine this in Lemma D.3 to bound the distance between $\mathbf{w}_t^{\text{proj}}$ and $\mathbf{w}^*$ under the norm induced by $\overline{V}_t$. To handle the uncertainty of our transition estimator, we define and bound bonus terms $B_t$ in Lemmas D.4 to D.6. Combining these results, we prove Lemma 4.5, which states that with high probability, $\forall t = 1, \ldots, T : \pi^* \in \Pi_t$. In Lemma D.7, we prove the regret bound as a combination of the three terms of the final bound of Theorem E.1, which however still depend on the bonus terms and policy embedding difference under the $\|\cdot\|_{\overline{V}_t}$-norm. Finally, in Theorem E.1 we use auxiliary results to remove that dependence and bound each of these three terms in terms of just the various constants (MDP parameters and assumed embedding & parameter vector bounds), the number of online iterations $T$, and the number of offline data $n$.

### 4.5 LIMITATIONS AND ASSUMPTIONS OF BRIDGE

Here we briefly discuss the implications of our assumptions and resulting limitations of our theory.

**Linear preferences and bounded features.** The chosen feature embedding $\phi$ must be sufficiently expressive to capture the expert's preferences. In case the expert preferences are unknown, one has to resort to naïve policy embeddings. These perform worse than informed ones (as seen in Figure 6), but can still work well enough, as our main experiments show in Figure 2. Future work could investigate learning embeddings from the offline dataset $\mathbb{D}_n^H$ before the online phase begins, using self-supervised objectives van den Oord et al. (2018).

**Realizability and expert data.** We assume the expert policy is realizable, and that the offline data comes from the expert. As a result, the quality of the offline data is key to BRIDGE's performance. If the offline data is noisy or suboptimal, the resulting BC policy will be a poor center for our Hellinger ball and the confidence set may require a larger radius to contain the expert policy - or, worse, may not contain it at all with useful radii. Our ablation on suboptimal offline data in Appendix A.1 confirms that as data is noisier, the effectiveness of our offline filtering (search space reduction) decreases.

**Expert policy concentration.** Our assumption $\gamma_{\min} > 0$ is weaker than comparable ones in the literature that require minimal offline data coverage across all state-action pairs (Rashidinejad et al., 2021). We only require that the expert's policy has a minimum visitation probability for the states it visits. The dependence of regret on the minimum visitation probability $\gamma_{\min}$ arises naturally from concentrating statistical estimation error on states where expert support exists.

## 5 EXPERIMENTS

We conduct experiments to validate our theoretical claims and demonstrate the empirical effectiveness of BRIDGE. We implement our algorithm for both discrete and continuous MDPs. As baselines, we implement Foster et al. (2024)'s offline Behavioral Cloning (BC) and Saha et al. (2023)'s online preference-based RL (PbRL) algorithms, for both of which no implementations are publicly available. **BRIDGE outperforms both baselines' cumulative regret in both discrete and continuous control environments.** We conduct ablation studies comparing the impact of the radius, expert suboptimality, number of offline trajectories, and embedding functions. We refer to the appendix for details on the ablations (A.1), environments (A.2), embeddings (A.3) and descriptions of the algorithm implementations (A.5).

**Discrete and continuous environments.** We provide a separate implementation of BRIDGE and PbRL for both types of environments, detailed descriptions are in Appendix A.5. We evaluate BRIDGE against offline BC and online PbRL baselines in two discrete (`StarMDP` and `Gridworld`) and continuous control (`Reacher` and `Ant`) environments. For each algorithm, we measure regret as the difference in expected reward between the currently selected "best" policy and the expert policy. As shown in Figure 2, BRIDGE achieves lower cumulative regret than both baselines across all environments. Figure 3 shows that BRIDGE refines its policy search space $\Pi_t$ faster than the PbRL baseline.

In the discrete case, the confidence set filtering step (line 6) and optimization step (line 7) require $\mathcal{O}(|\Pi_{1-\delta}^{\text{offline}}|^2)$ and $\mathcal{O}(|\Pi_t|^2)$ many computations, respectively. In continuous control, with its infinitely large policy space, we use a finite approximation of $\Pi$ to render these steps tractable (see Appendix A.5.2). With our implementation, we provide a solution to this practical optimization challenge that most algorithms with theoretical guarantees in imitation learning (Foster et al., 2024) and preference-based RL (Chen et al., 2022; Drago et al., 2025) do not address explicitly. We view this as an additional contribution.

**Ablations.** We performed several ablations, with full details found in Appendix A.1. Our findings show performance is sensitive to the confidence set radius: A large radius is less effective as the search space is poorly constrained, while a radius that is too small can break theoretical guarantees by excluding the optimal policy $\pi^*$ (Lemma 4.5). The quantity and quality of offline data also directly impact performance, as more high-quality trajectories shrink the confidence set and improve performance, while less-optimal data leads to less shrinkage. The choice of feature embedding $\phi$ is critical. Embeddings that are better aligned with the ground-truth expert preferences significantly

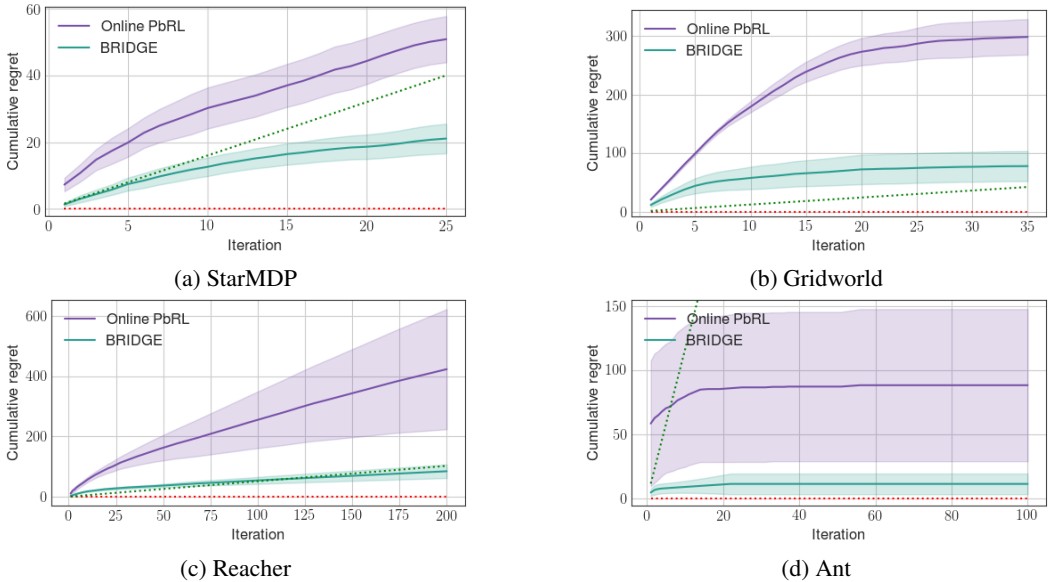

Figure 2: **Cumulative regret versus baselines across four environments.** Our method, BRIDGE, achieves lower regret than the offline BC (Foster et al., 2024) and online PbRL (Saha et al., 2023) baselines in both discrete tasks (a & b) and continuous control tasks (c & d). Dotted lines show BC (green) and expert (red) regret. Mean and 95% CI over 20 seeds. Embeddings used (cf. Appendix A.3): (a) `identity-short`, (b) `state-counts`, (c & d) `average_state-action`. Number of offline demonstrations: (a) 2, (b) 10, (c) 20, (d) 30.

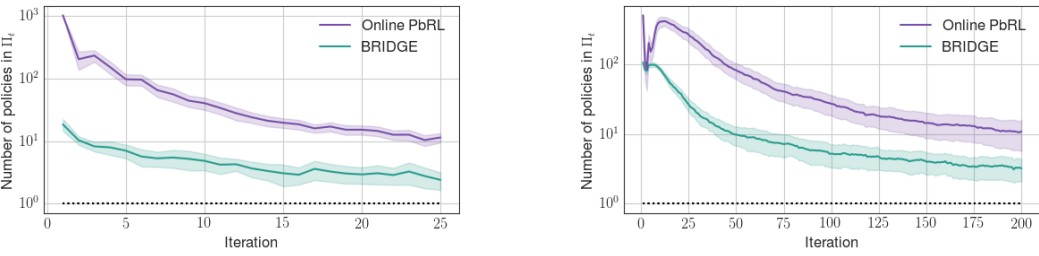

Figure 3: **Policy set size refinement** for discrete (`StarMDP`, left) and continuous (`Reacher`, right) environments. Our BRIDGE rapidly prunes the policy search space compared to the online PbRL baseline, which explores more broadly. Mean and 95% CI over 20 seeds.

improve performance for both our method and the baseline. We verify that empirical regret follows the theoretical dependence on expert minimum visitation probability $\gamma_{\min}$. As $\gamma_{\min}$ shrinks, performance degrades. Finally, we show that injecting noise into the oracle feedback delays convergence and increases regret.

## 6 CONCLUSION

We introduce BRIDGE, an algorithm that addresses the real-world challenges of learning without specifiable reward functions and risky exploration by fine-tuning imitation policies with online preference feedback. We provide the first theoretical regret bound for this hybrid paradigm, proving that an offline-built confidence set shrinks the online search space to provably reduce regret. Our experiments in discrete and continuous control tasks validate this theory, showing BRIDGE achieves lower regret than both offline-only and online-only baselines. Our work opens new directions for developing interactive learning systems that can safely and efficiently improve from human input without explicit reward signals.

ACKNOWLEDGEMENTS

This research was generously supported with funding by the Hasler Foundation, under the project title "Unified Feedback Integration Framework for Reinforcement Learning".

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

## A  EXPERIMENTS AND ADDITIONAL CONTEXT

We compare our algorithm with the log-loss behavioral cloning method of Foster et al. (2024) and the preference-based online learning algorithm of Saha et al. (2023). We could not find publicly available implementations for either of the two, so we made adaptions to achieve a computable implementation. Our separate implementations for discrete and continuous environments are described in Appendix A.5.1 and A.5.2 respectively.

All discrete experiments were run on an M1 Max CPU with 32GB of RAM, with a wall-clock time of roughly 3 seconds per iteration of the online loop for BRIDGE. The main computational bottleneck in the discrete implementation is the simulation of trajectories for approximating the expectation within $\phi(\pi)$, so runtime does not vary significantly between the different environments, if normalized for episode length. Steps 6 and 7 of the algorithm, within its online loop, are the most computationally complex. Step 6, filtering $\Pi_{1-\delta}^{\text{offline}}$ to obtain $\Pi_t$, requires $\mathcal{O}(|\Pi_{1-\delta}^{\text{offline}}|^2)$ many calculations involving the embeddings. Step 7, finding the pair of policies in $\Pi_t$ that maximizes uncertainty, requires $\mathcal{O}(|\Pi_t|^2)$ many. The smaller the initial $\Pi_{1-\delta}^{\text{offline}}$, the shorter the runtime.

Throughout, we use deterministic, tabular policies, i.e., they are represented by a matrix of size $\mathcal{S} \times \mathcal{A}$, where each row is a one-hot vector defining the deterministic action taken in that state. The figures shown display results averaged over 30 seeds, with thick lines representing the average, and shaded areas the results contained within one standard deviation to either side of the average. The continuous control experiments were run on an HPC cluster on a variety of nodes with both AMD and Intel server CPUs of mixed generations (32- to 256-core), on 20 parallel seeds each using a separate core, and using less than 2GB of RAM per core. On these more complex environments, simulating rollouts and filtering the online confidence set at each iteration is considerably more expensive, and observed wall-clock experiment runtime for 200 iterations reached up to 8 hours (much faster, at only small performance loss, if forgoing online confidence set filtering). Runtime strongly varies between environments, as e.g. a higher dimensional state space and more complex transition dynamics increase memory and computation requirements.

Our main regret and search space size figures contain two types of plots. The first (cf. Figure 2) displays the (sub)optimality of the current best policy chosen by each online algorithm at each iteration. At the end of an iteration, this policy is chosen as the one from the offline confidence set $\Pi_{1-\delta}^{\text{offline}}$ which maximizes the learned score function $s^P(\pi) = \mathbb{E}_{\tau \sim \mathbb{P}_{P*}^{\pi}}[\langle \phi(\tau), \mathbf{w}_t^{proj} \rangle]$. Its expected reward is simulated and compared to the optimal policy's (red dotted line) to calculate the regret. The green dotted line is the expected reward of the Behavioral Cloning policy estimated using Foster et al. (2024). The second plot (cf. Figure 3) illustrates the speed at which the algorithms pare down the size of the policy confidence set $\Pi_t$ – once the set contains only a single element, we consider the algorithm converged, as that element is the algorithm's estimate of the optimal policy $\pi^*$.

### A.1  ABLATIONS

**Impact of radius on BRIDGE performance.**  On the `Reacher` continuous control environment with 20 offline trajectories, we vary the radius BRIDGE uses to filter the candidates. Figure 4 shows that higher radii lead to less filtering, and performance that approaches the online PbRL baseline's. Reducing the radius improves performance up to a point – if reduced by too much, the expert may no longer be contained in the filtered $\Pi^{\text{offline}}$ and thus the search space $\Pi_t$, leading to worsening regret.

**Impact of offline data amount and suboptimality on confidence set size.**  This ablation validates our central theoretical contributions: increasing the amount and quality of offline data constrains the policy search space, which in turn improves online regret and enables more sample-efficient preference learning. We conduct the ablation on the `Gridworld` environment. We vary the amount of offline expert trajectories in $\mathbb{D}_n^H$ from $n_{offline} = 10$ to 1000. Additionally, we vary the quality of the data using a noise parameter ranging from $0\%$ to $20\%$ that represents the probability that an expert action in the dataset is corrupted to a random action. Results shown are averaged over 100 random seeds.

Table 1 shows the percentage of policies remaining in the confidence set after Hellinger distance filtering, such that $100\%$ indicates no constraint and no filtering, and lower values show tighter

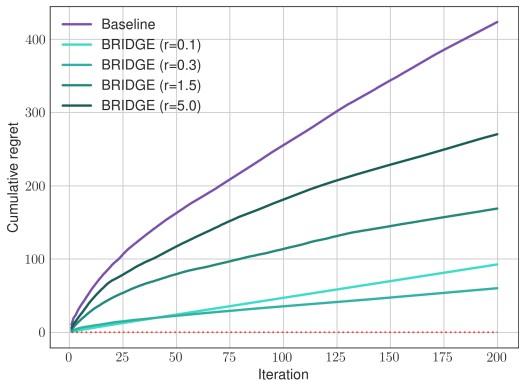

Figure 4: BRIDGE performance for different values of the radius used to filter candidate policies and create the offline confidence set. Higher radii lead to less filtering and performance that approaches the online PbRL baseline's, while a radius too small excludes (near-)optimal candidates, leading to unavoidable regret.

constraints and stronger filtering. We observe that on clean data, a 100-fold increase in training data leads to a $12.4\times$ reduction in search space size ($99.9\% \rightarrow 8.0\%$), roughly a $\mathcal{O}(1/\sqrt{n_{offline}})$ scaling. Our experiment shows that BRIDGE's filtering still works under noisy data, with filtering effectiveness weakening as noise increases.

| $n_{\textbf{offline}}$ | Confidence Set Size (%) | | |
|---|---|---|---|
| | **0% Noise** | **10% Noise** | **20% Noise** |
| 10 | 99.9±0.3 | 99.9±0.3 | 100.0±0.0 |
| 20 | 92.4±8.7 | 96.2±6.4 | 99.0±1.8 |
| 40 | 58.5±16.3 | 80.4±14.3 | 95.3±5.8 |
| 80 | 28.9±17.1 | 66.3±16.3 | 92.0±7.4 |
| 1000 | 8.0±4.8 | 65.6±10.0 | 95.4±3.3 |

Table 1: Empirical validation of confidence set scaling with offline data size and demonstration noise. Results averaged over 100 statistical runs.

**Impact of offline dataset size on BRIDGE performance.** We run an ablation comparing the impact of the amount of offline data $|\mathbb{D}_n^H|$ given on BRIDGE's performance. The experiment is again carried out on the `Reacher` continuous control environment. If given more offline trajectories, the quality of the BC policy $\pi^{BC}$ and thus also BRIDGE's candidate set of policies $\Pi^{offline}$ improves. We observe that BRIDGE quickly converges to the best policy in its candidate set, so more offline data, as expected, leads to a lower regret.

**Impact of choice of embedding $\phi$ on BRIDGE performance.** The choice of embedding has an outsized impact on both PbRL''s and BRIDGE's performance. We illustrate this again on the `Reacher` continuous control environment. We show three embeddings: one that approximates the true reward signal in this environment, and the environment-agnostic *average state-action* and *last state* embeddings. See Appendix A.3 for a detailed description of all embeddings used. Figure 6 illustrates how embeddings that more closely approximate the reward signal improve preference-learning performance. The *average reward*-emulating embedding massively simplifies the learning problem by making it easy to distinguish good from bad policies – the only downside being that it has to be handcrafted for this specific environment, which is harder the less one knows about the nature of the reward signal (but is trivial in a well-specified sim like MuJoCo). The alternative are the two state-agnostic embeddings, which show slower and worse convergence, with the richer *average state-action* embedding showing slightly better convergence of BRIDGE. This environment's rewards contain components that are non-linear in the observations, e.g., the total magnitude of acceleration $\|\mathbf{a}_t\|_2^2$, to punish harsh movements. These two embeddings cannot represent those components. We cannot expect them to fully converge purely from preference signals: their search

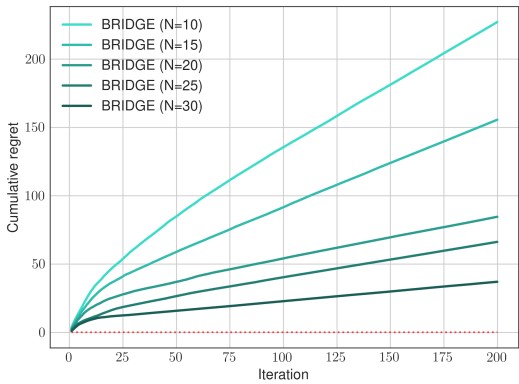

Figure 5: BRIDGE performance for different amounts of offline demonstration trajectories given. As the number of offline trajectories increases, BRIDGE's regret is reduced.

space may contain several potentially optimal policies whose trajectories differ only in those non-linear components and who thus cannot be distinguished using those embeddings and our linear model.

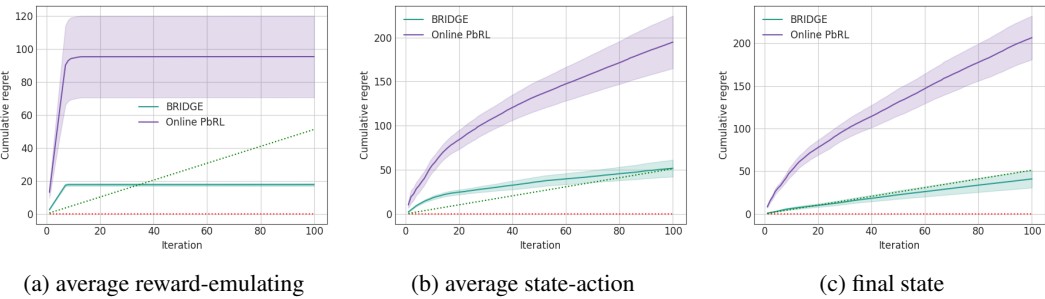

(a) average reward-emulating    (b) average state-action    (c) final state

Figure 6: Ablation showing BRIDGE's performance using three different embeddings in continuous environments. Embeddings that are closer to the true reward signal predictably perform better.

**Impact of minimum expert visitation** $\gamma_{\min}$**.** We run an ablation comparing the impact of different values of $\gamma_{\min}$ between $0.05$ and $0.25$. As $\gamma_{\min}$ shrinks, which corresponds to a more specialized expert, our theoretical regret bound increases. Figure 7 shows that experimentally, this behavior holds true, and smaller $\gamma_{\min}$ lead to a higher regret.

The environment we use for this ablation is a variation of the StarMDP that has two paths to the goal, it is described in Appendix A.1. It allows us to control the expert's $\gamma_{\min}$ via a parameter in the environment definition.

**Impact of noisy expert feedback.** We run an ablation comparing the impact of different levels of noise in the expert feedback, from $0\%$ to $50\%$. As Figure 9 shows, increased noise leads to a longer time until convergence, and higher cumulative regret.

## A.2 ENVIRONMENTS

**StarMDP (custom).** We illustrate the transition dynamics underlying the `StarMDP` in Figure 10. This environment features 5 states and 4 actions $a_0, a_1, a_2, a_3$ that correspond to `right`, `left`, `up` and `down` respectively. Actions have a probability of 0.7 of success, with an agent being moved to a different, random state with a probability of 0.3. Taking an "impossible" action such as going `left` in state $s_4$ will result in not moving with probability 1. Episodes have length $H = 8$ and start from $s_0$. The offline expert's dataset consists of 2 trajectories (re-drawn each seed).

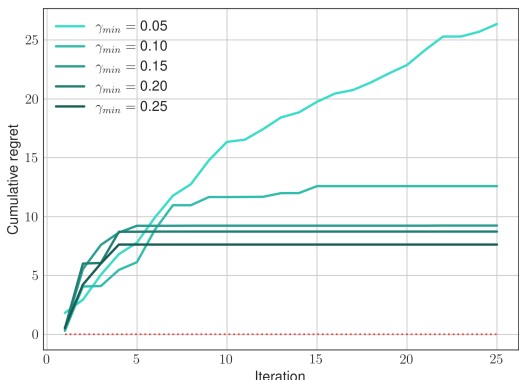

Figure 7: Ablation showing BRIDGE's performance with different levels of minimum expert visitation probability $\gamma_{\min}$. As $\gamma_{\min}$ shrinks, our regret bounds get looser, and empirical performance deteriorates.

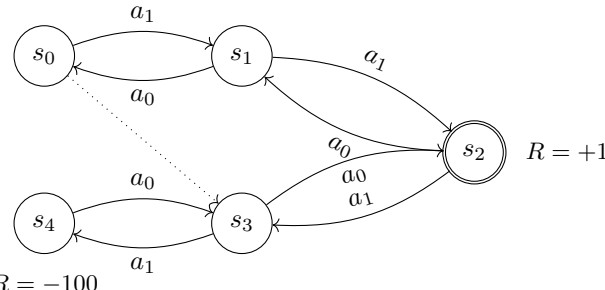

Figure 8: Environment used for the ablation on $\gamma_{\min}$. The starting state is $s_0$, when taking any action in $s_0$ there is a probability equal to $\gamma_{\min}$ to be transported to $s_3$ (dotted line). The only states with nonzero rewards are $s_2$ ($R = +1$) and $s_4$ ($R = -100$). The expert will always move along the trajectory $(s_0, s_1, s_2)$, or, if they randomly get transported to $s_3$, along $(s_0, s_3, s_2)$. Thus, the expert's minimum nonzero visitation probability $\gamma_{\min}$ is equal to the probability of getting transported.

**Gridworld (custom).** We illustrate the gridworld environment in Figure 11. The environment consists of a $4 \times 4$ grid with states associated with different rewards, including a negative-reward region in the top-right corner, a high-reward but unreachable state, and a moderate-reward goal state

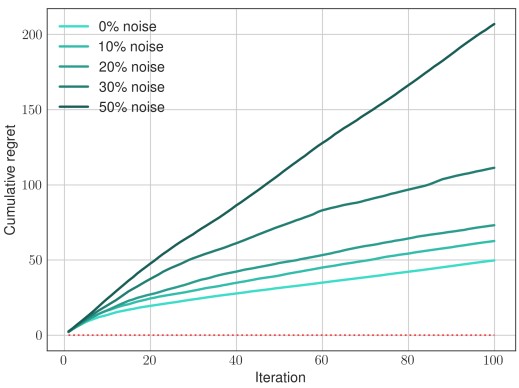

Figure 9: Ablation showing BRIDGE's performance with different levels of noise in the oracle feedback. At a given level, the oracle has a chance equivalent to the noise to return the wrong preference. With higher noise, convergence slows down, leading to higher cumulative regret.

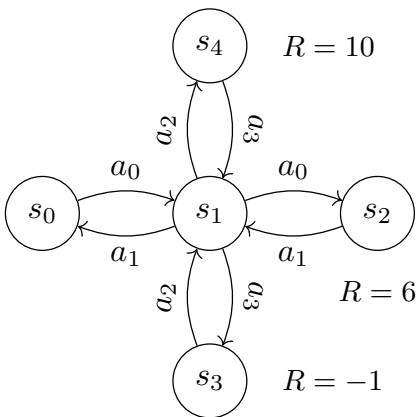

Figure 10: Star MDP. Transition probabilities are 0.7 for all solid arrows, otherwise the action takes the agent randomly to one of the other states.

at the bottom right corner. Each episode has length $H = 10$ and starts in the top-left corner. Each of the four actions (up, left, down, right) has a success probability of 0.8, whereas with probability 0.2 a randomly chosen different action is executed. Action stay remains in the current state with probability 1. Transitions beyond the grid limits or through obstacles have probability zero, with the remainder of the probability mass for each action being distributed among other directions equally. The offline dataset consists of 10 expert trajectories (re-drawn each seed).

| Start | | $-1$ | $-1$ |
|---|---|---|---|
| | | $-1$ | $-1$ |
| | 20 | | |
| | | | 10 |

Figure 11: Gridworld environment. Rewards at every state are indicated if non-zero. Transition probabilities are 0.9. Thick lines indicate an obstacle, through which state transitions have probability zero.

**Reacher (MuJoCo, v5).** This environment is part of the MuJoCo continuous control suite, which we use via *Gymnasium* (Towers et al., 2024). The agent controls a two-jointed robot arm on a 2D plane and needs to move its tip to a location that is sampled at random at the start of each episode. Rewards are a weighted combination of the distance between tip of the arm and the target, and a penalty term given as the euclidian norm of the action. The environment features a 10-dimensional observation space and 2-dimensional action space (torque at each joint). We train our own expert on this environment, using a *Stable-Baselines 3* PPO agent with training hyperparameters taken from *RL Baselines3 Zoo*'s (Raffin, 2020) reference implementations. We use it to generate an offline dataset of $n = 20$ trajectories (re-drawn each seed), each of length $H = 50$ (the default).

**Ant (MuJoCo, v5).** This is the second of our two continuous control environments, again accessed via *Gymnasium*. It is more complex as a control task than Reacher, but contains less stochastic elements. The agent is a 3D quadruped robot with four legs, that each feature two controllable joints. The aim is to move across a plane, with a slightly randomized initial location and orientation. Rewards are given for achieving a maximal distance in direction of the x-axis, with penalties for large action amplitudes and a bonus for survival (not flipping over). Our goal in selecting this environment is that the survival aspect allows behavioral cloning to quickly achieve a close-to-optimal policy,

but to have the remaining nonzero probability of sudden catastrophic failure require many more offline trajectories to fully converge. The action space has 8 dimensions. The 105-dimensional observation space is much bigger than in 'Ant'. *RL Baselines3 Zoo* provides pre-trained experts for many MuJoCo environments, but these are based on `-v3` versions of the environments, which in `Ant`'s case features a slightly different action space than our `-v5` version, thus making the agent incompatible. Just like in the `Reacher` environment, we used their training hyperparameters to train our own expert to convergence with TD3. The expert is used to generate an offline dataset of $n = 30$ trajectories (re-drawn each seed), each of length 100 (truncating from the default 1000).

## A.3 EMBEDDINGS

The choice of embedding function $\phi$ has implications on computational complexity and learning speed. Concretely, both a small dimension $d$ and upper bound $B$ for the norm of embedded trajectories are desirable. We present embeddings for both discrete (tabular) and continuous environments. See Pacchiano et al. (2020) and Parker-Holder et al. (2020a) for more possible embedding functions and analyses of their performance in different RL tasks.

Our experiments use the true reward signal to model the preferences. A general observation we make is that confirming intuition, the more closely an embedding approximates the true reward, the easier the learning problem is and the faster preference learning (both BRIDGE and the PbRL baseline) converges. If one has information about the nature of the true preferences (in our case, rewards), it seems helpful to incorporate those by crafting environment-specific embeddings, which we have done in the continuous case.

**Discrete environments.** We considered four options, defined on the space of trajectories. In the experiments shown we use two embeddings that strike a good balance between dimension, norm bound, and expressiveness. The `StarMDP` experiments use the `identity_short` embedding. The `Gridworld` experiments use the `state_counts` embedding. States and actions are represented as one-hot vectors.

Table 2: Discrete embedding definitions and properties

| Name | Definition $\phi(\tau)$ | d | B |
|------|------------------------|---|---|
| `identity_long` | $(s_0, a_0, \ldots, s_H, a_H)$ | $H(|\mathcal{S}| + |\mathcal{A}|)$ | $\sqrt{2H}$ |
| `identity_short` | $\sum_{t \leq H}(s_t, a_t)$ | $|\mathcal{S}| + |\mathcal{A}|$ | $\sqrt{2H}$ |
| `state_counts` | $\sum_{t \leq H}(s_t)$ | $|\mathcal{S}|$ | $H$ |
| `final_state` | $s_H$ | $|\mathcal{S}|$ | 1 |

**Continuous environments.** We use both environment-agnostic, and environment-specific embeddings, as shown in Table 3. Our main experiments for both `Reacher` and `Ant` (Section 5) use the `average_state-action` embedding, which is similar to the discrete `identity_short` embedding. We show the impact of using the env-agnostic `final_state` and env-specific `reacher_reward` embeddings in an ablation in Appendix A.1.

Table 3: Continuous embedding definitions and properties

| Name | Definition $\phi(\tau)$ | d | B |
|------|------------------------|---|---|
| `average_state-action` | $\frac{1}{H}\sum_{h \leq H}(s_h, a_h)$ | $|\mathcal{S}| + |\mathcal{A}|$ | $\sqrt{|\mathcal{S}| + |\mathcal{A}|}$ |
| `final_state` | $s_H$ | $|\mathcal{S}|$ | $\sqrt{|\mathcal{S}|}$ |
| `reacher_reward` | $\frac{1}{H}\sum_{h \leq H}(\|\text{dist-to-target}_h\|^2, \|a_h\|^2)$ | 2 | $2\sqrt{2}$ |

## A.4 COMPARING BRIDGE WITH MODERN PREFERENCE-BASED FINE-TUNING ALGORITHMS

We see three main points of comparison to BRIDGE in the space of preference-based fine-tuning algorithms: Reinforcement Learning from Human Feedback (RLHF) Ouyang et al. (2022), Direct

Preference Optimization (DPO) Rafailov et al. (2024), and online-preference based RL (PbRL) Saha et al. (2023).

**RLHF** first trains a static reward model (RM) from an offline dataset of preferences, then runs an RL algorithm (like PPO) against this reward model. This two-stage process can suffer from RM mis-specification or "reward hacking." Furthermore, it typically relies on a "soft" KL-divergence penalty to the initial policy as a heuristic for safety or style. BRIDGE differs in two ways: First, we avoid a static RM by *iteratively* learning the latent preference vector $\mathbf{w}^*$ within the online loop. Second, we replace the heuristic KL-constraint with a *provable* one: our online exploration is "hard" constrained to the offline Hellinger confidence set $\Pi_{1-\delta}^{\text{offline}}$, which is guaranteed to contain the expert policy $\pi^*$ with high probability.

**DPO** is a powerful offline method that learns directly from a static, pre-collected set of preferences. BRIDGE is designed for a different, online problem setting. Our online stage is an active learning loop that intelligently selects which new queries to make to a live expert to minimize online regret. Moreover, like RLHF, DPO's offline phase requires preferences, whereas BRIDGE's offline phase uses reward-free expert demonstrations.

**Online-only PbRL** is what BRIDGE's online component builds on. Our core contribution is adapting it into a hybrid framework. BRIDGE improves on the PbRL baseline in two ways: First, we constrain policy selection to $\Pi_{1-\delta}^{\text{offline}}$, which establishes a tighter, data-dependent exploration variance bound that shrinks with number of offline demonstrations $n$, formally connecting offline data to online regret. Second, we "pool" both offline and online data to create a more sample-efficient online transition model estimator.

### A.5 PRACTICAL IMPLEMENTATIONS OF BRIDGE

We provide two different implementations of BRIDGE, one for tabular, and the other for continuous environments.[3] The discrete implementation aims to implement BRIDGE as close to the theoretical description as possible, while the continuous one implements its main ideas, but has to take more liberties in details to stay computable. Appendix A.5.1 shows the discrete implementation, Appendix A.5.2 the continuous one, and Appendix A.5.3 presents a computationally efficient way to calculate the Hellinger distance in the discrete case, which we use in our discrete implementation.

### A.5.1 DISCRETE IMPLEMENTATION

**Offline learning** For both our testing environments `StarMDP` and `Gridworld`, we obtain the (tabular) optimal policy $\pi^*$ by solving a linear program using `cvxopt`. We sample trajectories from this policy to create a dataset of offline trajectories $\mathbb{D}_n^H$. The learned transition models are trained on the offline trajectory dataset. The model for `StarMDP` is a Maximum Likelihood Estimator (MLE) based on the state visitation counts, while `Gridworld` is a 2-layer MLP with a hidden dimension of 32 and ReLu activations trained to predict next states with a cross-entropy loss. We estimate the optimal policy on the offline dataset with log-loss Behavioral Cloning (`LogLossBC` in Foster et al. (2024)) using Adam, resulting in $\hat{\pi}$.

To obtain $\Pi_{1-\delta}^{\text{offline}}$, we use rejection sampling, although the search space of policies depends on the MDP. In `StarMDP`, we construct all $|\Pi| = 1024$ deterministic, stationary policies and iterate through each of them, calculating its Hellinger distance to $\hat{\pi}$ and adding it to $\Pi_{1-\delta}^{\text{offline}}$ if the distance is less than $R$. In larger MDPs this is infeasible as $|\Pi|$ quickly grows. In `Gridworld`, we sample $500,000$ random policies, and build $\Pi_{1-\delta}^{\text{offline}}$ by iterating through that sample. This sample is large enough to contain close-to-optimal policies with near certainty while staying computationally feasible to exhaustively check. Larger MDPs may require larger samples.

The purely online baseline PbRL in principle searches the space of all (deterministic, stationary) policies $\Pi$. This is feasible in `StarMDP`, but in `Gridworld`, we have to make a pragmatic adaption. We define PbRL's search space as $\Pi_{1-\delta}^{\text{offline}}$ (which is on the order of $< 50$ policies), augmented by random policies to reach a set of size $1000$.

---

[3]Code: https://github.com/pfriedric/bridge.

**Online, preference-based learning**   In the online loop, to estimate $\phi(\pi) = \mathbb{E}_{\tau \sim \mathbb{P}_{P^*}}[\phi(\tau)]$ for any $\pi$, we sample 100 trajectories $\tau$ and average the returned embeddings. To start the online loop, we initialize $\mathbf{w}_0^{proj}$ as a vector of random normal values with mean 0 and variance 1. In subsequent iterations $t$, $\mathbf{w}_t^{MLE}$ is initialized as a normalized vector of ones (this does improve convergence compared to random initialization) and trained on all online trajectories observed so far using a regularized binary cross-entropy loss (as in Saha et al. (2023), Section 3.1) and Adam for 10 episodes. After preferences have been collected, we update the learned transition model, obtaining $\hat{P}_{t+1}$ by retraining from scratch the same models and losses as described in the offline part on all online trajectories observed so far. At the end of each iteration, we find the policy with the highest predicted score $\langle \phi(\pi), \mathbf{w}_t^{proj} \rangle$ and calculate its average reward as well as the true optimal policy $\pi^*$'s over 1000 sampled trajectories under the true transitions and compare the two in our suboptimality plots.

### A.5.2   Continuous implementation

**Offline learning.**   For both environments `Reacher` and `Ant`, we train agents using the hyperparameters from *RL Baselines3 Zoo*. We sample trajectories from this policy to create a dataset of offline trajectories $\mathbb{D}_n^H$. Unlike in the discrete implementation, for simplicity, we do not implement a learned transition model (implementation would work exactly the same as in the discrete case). We again obtain an estimate $\pi^{BC}$ of the optimal policy on the offline dataset with log-loss Behavioral Cloning using Adam (`LogLossBC` in Foster et al. (2024)). Policies are modeled as Gaussian policies with a 2-layer MLP of 64 (`Reacher`) or 256 (`Ant`) neurons per layer.

As the size of the true policy space is infinite and the rejection sampling from random policies (our approach in discrete MDPs) is computationally infeasible, we obtain the offline confidence set $\Pi_{1-\delta}^{\text{offline}}$ constructively. An alternative approach would be to discretise the state- and action space to land back at a discrete setting, but we show here how to adapt BRIDGE to the fully continuous setting.

We first construct a proxy $\tilde{\Pi}$ to the true policy space $\Pi$. The learning problem for BRIDGE and PbRL is fundamentally to learn to distinguish between "good" and "bad" policies, in our case in terms of expected reward. Policies with zero or near-zero expected reward are easily distinguishable from the others by both algorithms and regardless of embedding. They also form the overwhelming majority of all policies in the true $\Pi$ or a random sample of it. Including them thus simply increases computation time without meaningfully impacting both algorithms' dynamics. Our goal is to construct a proxy for $\Pi$ that is computationally feasible to search and contains policies ranging in performance from near-zero to near-optimal or even optimal in roughly even proportions, skewed toward including more worse policies, but not to the extremely lopsided degree of the true $\Pi$. Our solution is to construct a union of two sets:

$$\tilde{\Pi} := \{\pi^{BC}, \pi^{BC} + \text{small noise}\} \cup \{\pi^{BC} + \text{large noise}\}.$$

The first set contains policies that are close to the BC policy in terms of both reward and distance in trajectory distribution space, and is expected to also contain near-optimal or optimal policies that improve on $\pi^{BC}$. We obtain it by adding a small amount of Gaussian noise to the BC policy's parameters. The second set is meant to represent the rest of the policy space that achieves rewards ranging from zero to decent, but not near-optimal. It is constructed similar to the first, but with much higher levels of noise. By tuning the noise level, this approach results in policies that cover the remaining spectrum of performance (decent to near-zero) and distance to $\pi^{BC}$ (in trajectory distribution space).

We then define BRIDGE's filtered offline confidence set

$$\Pi^{\text{offline}} := \left\{ \pi \in \tilde{\Pi} \mid \|\phi(\pi) - \phi(\pi^{BC})\|_2 < \text{radius} \right\},$$

using the L2, (rather than Hellinger) distance in trajectory distribution space for computability.

**Online, preference-based learning.**   We estimate $\phi(\pi) = \mathbb{E}_{\tau \sim \mathbb{P}_{P^*}}[\phi(\tau)]$ by sampling 200 trajectories $\tau$ and averaging the returned embeddings. For massively increased performance, we do this

only once at the start of the online loop and then use cached versions. The order of operations in the online loop is slightly different than as stated in theory and in the discrete case.

We first filter $\Pi^{\text{offline}}$ to obtain the online confidence set,

$$
\Pi_t = \left\{ \pi \in \Pi^{\text{offline}} \mid \forall \pi' \in \Pi^{\text{offline}} : \langle \phi(\pi) - \phi(\pi'), \mathbf{w}_t \rangle + \gamma \| \phi(\pi) - \phi(\pi) \|_{\bar{\mathbf{V}}_t^{-1}} \geq 0 \right\}.
$$

As we assume a known transition model, there are no bonus terms. The exploration scaling factor $\gamma$ can be chosen to increase (smaller $\gamma$) or reduce (larger $\gamma$) the speed at which the confidence set is pruned.

Then, we sample a pair of policies from the set, $(\pi^1, \pi^2) \in \Pi_t$. There are several ways to implement sampling that follow the spirit of the theoretical algorithm. We have tested three:

- $\pi^1 = \arg\max\langle \phi(\pi), \mathbf{w}_t \rangle$ and $\pi^2 = \text{random}$, picking a pair of the current estimated optimal policy and a random other,
- $(\pi^1, \pi^2) = \arg\max\langle \phi(\pi^1) - \phi(\pi^2), \mathbf{w}_t \rangle + \beta \| \phi(\pi^1) - \phi(\pi^2) \|_{\bar{\mathbf{V}}_t^{-1}}$, spiritually similar to UCB, which picks the pair maximizing a $\beta$-weighted combination of the difference in estimated win probabilities and the uncertainty of that estimate,
- and perhaps closest to the theoretical algorithm, picking a pair purely based on the the uncertainty, $(\pi^1, \pi^2) = \arg\max\| \phi(\pi^1) - \phi(\pi^2) \|_{\bar{\mathbf{V}}_t^{-1}}$.

Although they show similar performance, on our environments and embeddings, the first performed slightly better and is the one we pick throughout.

As in theory, we then sample a trajectory from the pair, receive the oracle preference $o_t = \mathbb{I}(\tau_t^1 \succ \tau_t^2)$ (in our case, the higher true trajectory reward), and store the tuple of embedding differences $\Delta \phi_t = \phi(\pi^1) - \phi(\pi^2)$ and preference signal $(\Delta \phi_t, o_t)$ in the online preference buffer $\mathbb{D}^{\text{pref}}$. To increase convergence speed, we repeat this $N_{\text{rollouts}} = 10$ many times for the same policy pair each iteration.

We then learn $\mathbf{w}_t$ on the preference buffer $\mathbb{D}^{\text{pref}}$ using MLE and continual training (rather than starting from scratch every episode) of 100 epochs per iteration.

Finally, we update the data matrix $\mathbf{V}_t = \mathbf{V}_{t-1} + (\Delta \phi_t)^{\otimes 2}$.

### A.5.3 An efficient calculation of the (squared) Hellinger distance in the discrete case

Here, we demonstrate how under the assumptions of our model there is a computationally tractable method of calculating the Hellinger distances we need that avoids (intractable) iteration over the entire trajectory space.

**Reducing Hellinger distance to a recursive scheme.** The Hellinger distance between two distributions $\mathbb{P}_{P^1}^{\pi^1}, \mathbb{P}_{P^2}^{\pi^2}$ is a measure of distance over the space of trajectories. Its square is defined as

$$
H^2(\mathbb{P}_{P^1}^{\pi^1}, \mathbb{P}_{P^2}^{\pi^2}) = 1 - \sum_{\text{trajectories } \tau} \sqrt{\mathbb{P}_{P^1}^{\pi^1}(\tau) \mathbb{P}_{P^2}^{\pi^2}(\tau)}
$$
$$
= 1 - BC(\mathbb{P}_{P^1}^{\pi^1}, \mathbb{P}_{P^2}^{\pi^2}).
$$

where the term of the sum is called the Bhattacharyya coefficient. Calculating this sum is normally intractable, as the space of trajectories is too large to exhaustively compute anything over. In our case, there is a way to not just calculate this sum (and therefore the Hellinger distance), but do so very efficiently, and we show it here.

In an abuse of notation, we use the fact that we assume stationary, deterministic policies, to write $\pi(s_t)$ to refer to the action $\pi$ chooses with probability 1 at state $s_t$. We first note that $\mathbb{P}_{P^{1,2}}^{\pi^{1,2}} = d_0(s_0) \prod_{t=0}^{H-1} \pi^{1,2}(a_t|s_t) P^{1,2}(s_{t+1}|s_t, a_t)$, where $d_0(\cdot)$ is the initial state distribution.

Our ultimate goal is to efficiently calculate the Bhattacharyya coefficient. Let $\tau_t = (s_0, a_0, \ldots, a_{t-1}, s_t)$ be a trajectory of length $t$ that ends in $s_t$. Let us write out the square-root term

$$\sqrt{\mathbb{P}_{P^1}^{\pi^1}(\tau_t)\mathbb{P}_{P^2}^{\pi^2}(\tau_t)} = \sqrt{d_0(s_0) \prod_{j=0\ldots t-1} \pi^1(a_t|s_t)P^1(s_{j+1}|s_t, a_t) \cdot d_0(s_0) \prod_{j=0\ldots t-1} \pi^2(a_t|s_t)P^2(s_{j+1}|s_t, a_t)}$$

$$= d_0(s_0)\sqrt{\prod_{j=0\ldots t-1} \pi^1(a_t|s_t)P^1(s_{j+1}|s_t, a_t)\pi^2(a_t|s_t)P^2(s_{j+1}|s_t, a_t)}.$$

Since $\pi^1$ and $\pi^2$ are deterministic, we can simplify it. Whenever $\pi^1(s) \neq \pi^2(s)$ for any $s$ in the trajectory $\tau_t$, either $\pi^1(s)$ or $\pi^2(s)$ are zero, which reduces their product to $0$ and thus zeroes out the entire term. Thus,

$$\sqrt{\mathbb{P}_{P^1}^{\pi^1}(\tau_t)\mathbb{P}_{P^2}^{\pi^2}(\tau_t)} = d_0(s_0) \prod_{j=0\ldots t-1} \sqrt{P^1(s_{j+1}|s_t, \pi^1(s_t))P^2(s_{j+1}|s_t, \pi^2(s_t))}\mathbb{1}\{\pi^1(s_t) = \pi^2(s_t)\}.$$

We define $X_t(s)$ as the sum of the square root of the product of $P^1, P^2$ for all partial trajectories $\tau_t$ of length $t$ ending in state $s$, i.e.:

$$X_t(s) := \sum_{\tau_t \text{ ending in } s} \sqrt{\mathbb{P}_{P^1}^{\pi^1}(\tau_t)\mathbb{P}_{P^2}^{\pi^2}(\tau_t)}.$$

**Claim.** There is a recursive relationship:

$$X_{t+1}(s_{t+1}) = \sum_{s_t \in \mathcal{S}} X_t(s_t)\sqrt{P^1(s_{t+1}|s_t, \pi^1(s_t))P^2(s_{t+1}|s_t, \pi^2(s_t))}\mathbb{1}\{\pi^1(s_t) = \pi^2(s_t)\}.$$

*Proof.* For $t = 0$, the trajectories of length $0$ are just the initial states, so by definition, $X_0(s) = d_0(s)$. Note that we can write $X_0$ as a 1D vector of length $|\mathcal{S}|$.

For the induction step, by definition:

$$X_{t+1}(s_{t+1}) = \sum_{\tau_{t+1} \text{ ending in } s_{t+1}} \left( d_0(s_0) \prod_{j=0}^{t} \sqrt{\mathbb{P}_{P^1}^{\pi^1}(s_{j+1}|s_j, a_j)\mathbb{P}_{P^2}^{\pi^2}(s_{j+1}|s_j, a_j)} \right)$$

We can split the product inside the parentheses into $\prod_{j=0,\ldots,t-1}(\ldots) \cdot \sqrt{\mathbb{P}_{P^1}^{\pi^1}(s_{t+1}|s_t, a_t)\mathbb{P}_{P^2}^{\pi^2}(s_{t+1}|s_t, a_t)}$, and split the sum $\sum_{\tau_t \text{ ending in } s_{t+1}} = \sum_{s_t}\sum_{a_t}\sum_{s_0, a_0, \ldots, s_{t-1}, a_{t-1}}$ by first summing over history up to time:

$$X_{t+1}(s_{t+1}) = \sum_{s_t, a_t} \left( \sum_{s_0, a_0, \ldots, s_{t-1}, a_{t-1}} d_0(s_0) \prod_{j=0}^{t-1} \ldots \right) \cdot \sqrt{\mathbb{P}_{P^1}^{\pi^1}(s_{t+1}|s_t, a_t)\mathbb{P}_{P^2}^{\pi^2}(s_{t+1}|s_t, a_t)}$$

First, note that the term in brackets is exactly the definition of $X_t(s_t)$, so we can substitute it. Second, we can again use the fact that policies are deterministic, and that $\pi^1(s)\pi^2(s)$ is non-zero ($= 1$) if and only if the two policies agree on that state, and thus

$$\sqrt{\mathbb{P}_{P^1}^{\pi^1}(s_{t+1}|s_t, a_t)\mathbb{P}_{P^2}^{\pi^2}(s_{t+1}|s_t, a_t)} = \sqrt{P^1(s_{t+1}|s_t, \pi^1(s_t))P^2(s_{t+1}|s_t, \pi^2(s_t))} \cdot \mathbb{1}\{\pi^1(s_t) = \pi^2(s_t) = a_t\}.$$

Combining these two insights, we get exactly the claim. $\square$

Using the claim, if we have $X_t(s_H)$ for all $t = 0, \ldots, H$ and states $s_H \in \mathcal{S}$, we can calculate the Bhattacharyya coefficient:

$$\sum_{s_H \in \mathcal{S}} X_H(s_H) = \sum_{s_H \in \mathcal{S}} \sum_{\tau_H \text{ ending in } s_H} \sqrt{\mathbb{P}_{P^1}^{\pi^1}(\tau_H)\mathbb{P}_{P^2}^{\pi^2}(\tau_H)}$$

$$= \sum_{\text{trajectories } \tau \text{ of length } H} \sqrt{\mathbb{P}_{P^1}^{\pi^1}(\tau_H)\mathbb{P}_{P^2}^{\pi^2}(\tau_H)}$$

$$= BC(\mathbb{P}_{P^1}^{\pi^1}, \mathbb{P}_{P^2}^{\pi^2}).$$

**Efficiently computing $H^2$.** We can use the recursive scheme we just proved above to efficiently compute $X_H(s_H)$ for all $s_H$, and thus also the Bhattacharyya coefficient $BC(...)$ and finally the Hellinger distance $H^2(...)$. First, treat $X_t$ as a vector of length $|\mathcal{S}|$, where the $i-th$ entry is $X_t(s_i)$. Then, define a matrix $M$ such that

$$M_{s,s'} := \sqrt{P^1(s'|s, \pi^1(s))P^2(s'|s, \pi^2(s))}\mathbb{1}\{\pi^1(s) = \pi^2(s)\}.$$

Then, we have $X_1 = M \cdot X_0$, $X_2 = M \cdot X_1 = M^2 \cdot X_0$, ..., $X_H = M^H \cdot X_0$, and $X_0$ is simply the vector of probabilities of the initial state distribution $d_0$. Putting this all together, we get

$$H^2(\mathbb{P}_{P^1}^{\pi^1}, \mathbb{P}_{P^2}^{\pi^2}) = 1 - \sum_{i=0}^{|\mathcal{S}|} \left[M^H d_0\right]_i.$$

This way, we avoid having to do any computations over the entire trajectory space. Computational cost is merely building the matrix $M$ once at the beginning based on $\pi^1, \pi^2, P^1$ and $P^2$ ($\mathcal{O}(|\mathcal{S}|^2)$), computing $M^H$ ($\mathcal{O}(\log H)$ matrix multiplications of $\mathcal{O}(|\mathcal{S}|^{\log_2 7})$ each), and calculating $X_H = M^H X_0$ with one final matrix multiplication, for a total computational complexity of $\mathcal{O}(\log H |\mathcal{S}|^{\log_2 7})$ and memory complexity of $\mathcal{O}(|\mathcal{S}|^2)$ – tractable for moderate-sized MDPs.

**Intuition in our case of $P^1 = P^2$.** Our BRIDGE algorithm computes $H^2(\mathbb{P}_{\hat{P}}^{\pi^1}, \mathbb{P}_{\hat{P}}^{\pi^2})$ with the same underlying transition distribution $\hat{P}$. In that case,

$$\sqrt{\mathbb{P}_{\hat{P}}^{\pi^1}(\tau_H)\mathbb{P}_{\hat{P}}^{\pi^2}(\tau_H)} = d_0(s_0) \prod_{j=0}^{H-1} \sqrt{\hat{P}(s_{j+1}|s_j, \pi^1(s_j))\hat{P}(s_{j+1}|s_j, \pi^2(s_j))}\mathbb{1}\{\pi^1(s_t) = \pi^2(s_t) = a_t\}$$

$$= d_0(s_0) \prod_{j=0}^{H-1} \hat{P}(s_{j+1}|s_j, a_t)\mathbb{1}\{\pi^1(s_t) = \pi^2(s_t) = a_t\}$$

$$= \hat{P}(\tau_H)\mathbb{1}\{\pi^1 \text{ and } \pi^2 \text{ agree on } \tau_H\}.$$

If a trajectory passes only through states where $\pi^1$ and $\pi^2$'s actions agree, we can call it an *agreement trajectory*. Then, the squared Hellinger distance has a direct interpretation using the total probability mass of agreement trajectories:

$$H^2(\mathbb{P}_{\hat{P}}^{\pi^1}, \mathbb{P}_{\hat{P}}^{\pi^2}) = 1 - BC(\mathbb{P}_{\hat{P}}^{\pi^1}, \mathbb{P}_{\hat{P}}^{\pi^2})$$

$$= 1 - \sum_{\tau_H} \sqrt{\mathbb{P}_{\hat{P}}^{\pi^1}(\tau_H)\mathbb{P}_{\hat{P}}^{\pi^2}(\tau_H)}$$

$$= 1 - Prob(\text{agreement trajectories under } \hat{P}).$$

The complex interpretation of Hellinger distance thus becomes a simple question:

> *What is the probability that a trajectory evolves for $H$ steps without ever hitting a state where $\pi^1$ and $\pi^2$ diverge?*

## B  SIMPLIFIED SETUP FOR UNDERSTANDING REGRET ANALYSIS

In this section, we propose an analysis of the regret under a simplified setting, where the underlying dynamics $P^*$ are known. We aim to build understanding of how the construction of the confidence set over the policies from the offline learning estimation helps to reduce the number of policies to draw from in the online learning setting. By ignoring the added complexity of the transition estimation, we can highlight which part of our methods applies to the policies. The goal is to prepare the reader for the proof of our algorithm BRIDGE in Appendix E.

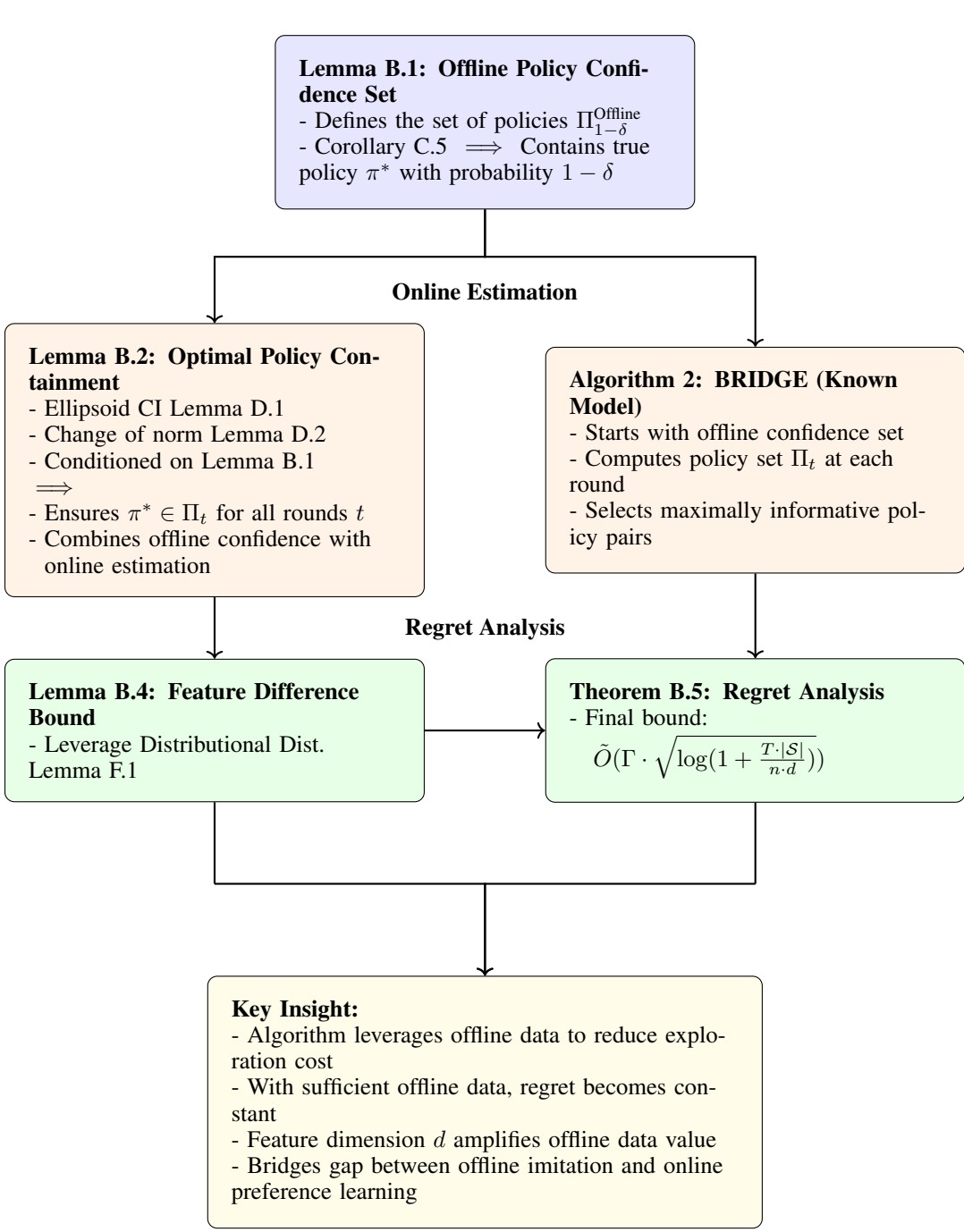

Figure 12: Proof Overview for BRIDGE Algorithm with Known Dynamics

## B.1 Setup for Known Dynamics

### B.1.1 Offline Estimation with Known Dynamics

Assume we get the offline data $\mathbb{D}_n^H = \{\tau_i\}_{i \in [n]}$. The underlying object describing the trajectories is a finite MDP, reward-free setting as in the main paper. Assume that the set of possible policies is

stationary and deterministic. Then assuming the underlying dynamics are known, the confidence set from Theorem 4.2 reduces to the following, by direct application of Corollary C.5, i.e., setting the radius around the MLE estimate $\pi^{BC}$ from Equation (7).

We formalize this into the following Lemma:

**Lemma B.1** (Offline policy confidence set under known dynamics). *Let $\pi^{BC}$ be the log-loss BC estimator defined in Equation* (7).
*The policy set*

$$\Pi_{1-\delta}^{\text{Offline}} := \left\{ \pi : H(\mathbb{P}_{P^*}^{\pi}, \mathbb{P}_{P^*}^{\pi^{BC}}) \leq \sqrt{\frac{6 \cdot |\mathcal{S}| \cdot \log(|\mathcal{A}| \cdot \delta^{-1})}{n}} \right\}$$

*contains $\pi^*$ with probability at least $1 - \delta$.*

*Proof.* Note that by symmetry

$$H(\mathbb{P}_{P^*}^{\pi^*}, \mathbb{P}_{P^*}^{\pi^{BC}}) = H(\mathbb{P}_{P^*}^{\pi^{BC}}, \mathbb{P}_{P^*}^{\pi^*})$$

Then the result follows from Corollary C.5. $\square$

### B.1.2 ONLINE LEARNING WITH KNOWN DYNAMICS

Here, we adapt our algorithm BRIDGE to the setting with known transition dynamics $P^*$. We adapt the approach from Saha et al. (2023) under known dynamics to constrain the set of policies to choose from to our offline confidence set $\Pi_{1-\delta}^{\text{offline}}$ described in the previous section.

First, since the transitions are known, for this section we define:

$$\phi^{P^*}(\pi) := \phi(\pi) = \mathbb{E}_{\tau \sim \mathbb{P}_{P^*}^{\pi}}[\phi(\tau)]$$

We also define the expected data matrix $\overline{V}_t^{P^*}$ under the true transition dynamics $P^*$ as follows (see Appendix D for an overview of results about data matrices):

$$\overline{V}_t^{P^*} = \kappa\lambda\mathbf{I}_d + \sum_{\ell=1}^{t-1} \left(\phi(\pi_\ell^1) - \phi(\pi_\ell^2)\right)\left(\phi(\pi_\ell^1) - \phi(\pi_\ell^2)\right)^\top.$$

Then we define the set of policies $\Pi_t$ to draw from during the online iterations as:

$$\alpha_{d,T}(\delta) := 20BW\sqrt{d\log(T(1+2T)/\delta)}, \quad \text{(cf. Lemma D.2)}$$
$$\gamma_t^{\text{known}} := 4\kappa\beta_t(\delta) + \alpha_{d,T}(\delta),$$
$$\Pi_t := \left\{ \pi \in \Pi_{1-\delta}^{\text{Offline}} \mid \forall \pi' \in \Pi_{1-\delta}^{\text{Offline}} : \right.$$
$$\left. \langle\phi(\pi) - \phi(\pi'), \mathbf{w}_t^{\text{proj}}\rangle + \gamma_t^{\text{known}} \cdot \|\phi(\pi) - \phi(\pi')\|_{(\overline{V}_t^{P^*})^{-1}} \geq 0 \right\}.$$

**Lemma B.2** (Optimal policy containment). *Conditioned on $E_{w^*} \cap E_{\overline{V}_T^{P^*}} \cap E_{\text{offline}}$ where:*

- *$E_{w^*}$ is the event defined in Lemma D.1*

- *$E_{\overline{V}_T^{P^*}}$ is the event defined in Lemma D.2*

- *$E_{\text{offline}} := \{\pi^* \in \Pi_{1-\delta}^{\text{Offline}}\}$*

*then*

$$\pi^* \in \Pi_t \quad \forall t \in [T].$$

*Proof.* This follows directly from Lemma 2 in Saha et al. (2023). We adapt the probability parameter $\delta$ to account for the additional condition that $\pi^* \in \Pi_{1-\delta}^{\text{Offline}}$, which holds with probability at least $1-\delta$ according to Lemma C.13. $\square$

We now present the adapted version of BRIDGE for the known transition model:

---

**Algorithm 2 BRIDGE (known model):** **B**ounded **R**egret with **I**mitation **D**ata and **G**uided **E**xploration

---

1: **Input:** Offline dataset $\mathbb{D}_n^H$, time horizon $T$, true dynamics $P^*$
2: Compute confidence set $\Pi_{1-\delta}^{\text{Offline}}$ using Lemma B.1
3: Initialize $\overline{\mathbf{V}}_1^{P^*} \leftarrow \kappa\lambda\mathbf{I}_d$                     ▷ Initialize data matrix
4: **for** $t = 1, \ldots, T$ **do**
5:     Compute $\mathbf{w}_t^{\text{proj}}$ via constrained MLE (Equation (4))
6:     Define policy set $\Pi_t$ based on $\Pi_{1-\delta}^{\text{Offline}}$ and $\mathbf{w}_t^{\text{proj}}$
7:     $(\pi_t^1, \pi_t^2) \leftarrow \arg\max_{\pi^1, \pi^2 \in \Pi_t}\{\|\phi(\pi^1) - \phi(\pi^2)\|_{(\overline{\mathbf{V}}_t^{P^*})^{-1}}\}$
8:     Sample trajectories $\tau_t^1 \sim \mathbb{P}_{P^*}^{\pi_t^1}, \tau_t^2 \sim \mathbb{P}_{P^*}^{\pi_t^2}$ and obtain preference $o_t = \mathbb{I}(\tau_t^1 \succ \tau_t^2)$
9:     Update matrix: $\overline{\mathbf{V}}_{t+1}^{P^*} \leftarrow \overline{\mathbf{V}}_t^{P^*} + (\phi(\pi_t^1) - \phi(\pi_t^2))(\phi(\pi_t^1) - \phi(\pi_t^2))^\top$
10: **end for**
11: **return** Best policy from $\Pi_T$ using final weight estimate $\mathbf{w}_T^{\text{proj}}$

---

## B.2 REGRET ANALYSIS: BRIDGE (KNOWN MODEL)

We now present a regret analysis of the BRIDGE algorithm under known transition. We start by stating the following lemma:

**Lemma B.3.** *The regret of* BRIDGE *under known dynamics is bounded from above by:*

$$R_T \leq 2\gamma_T^{known} \sum_{t \in [T]} \|\phi(\pi_t^1) - \phi(\pi_t^2)\|_{(\overline{V}_t^{P^*})^{-1}}$$

*Proof.* Let $\Delta\phi_t^{i,j} := \phi(\pi_t^i) - \phi(\pi_t^j)$. First, we bound the instantaneous regret, using Lemma D.2 in the last line:

$$2r_t = \langle\Delta\phi_t^{*,1}, \mathbf{w}^*\rangle + \langle\Delta\phi_t^{*,2}, \mathbf{w}^*\rangle$$

$$\leq \langle\Delta\phi_t^{*,1}, \mathbf{w}_t^{\text{proj}}\rangle + \langle\Delta\phi_t^{*,2}, \mathbf{w}_t^{\text{proj}}\rangle + \|w^* - \mathbf{w}_t^{\text{proj}}\|_{\overline{V}_t^{P^*}}\left(\|\Delta\phi_t^{*,1}\|_{(\overline{V}_t^{P^*})^{-1}} + \|\Delta\phi_t^{*,2}\|_{(\overline{V}_t^{P^*})^{-1}}\right)$$

$$\leq \langle\Delta\phi_t^{*,1}, \mathbf{w}_t^{\text{proj}}\rangle + \langle\Delta\phi_t^{*,2}, \mathbf{w}_t^{\text{proj}}\rangle + \gamma_t^{\text{known}}\left(\|\Delta\phi_t^{*,1}\|_{(\overline{V}_t^{P^*})^{-1}} + \|\Delta\phi_t^{*,2}\|_{(\overline{V}_t^{P^*})^{-1}}\right).$$

Since we chose $\pi_t^1, \pi_t^2 = \arg\max\|\Delta\phi^{i,j}\|$ and know that $\pi^* \in \Pi_t$ (Lemma B.2), we get

$$2r_t \leq \langle\Delta\phi_t^{*,1}, \mathbf{w}_t^{\text{proj}}\rangle + \langle\Delta\phi_t^{*,2}, \mathbf{w}_t^{\text{proj}}\rangle + 2\gamma_t^{\text{known}}\left(\|\Delta\phi^{1,2}\|_{(\overline{V}_t^{P^*})^{-1}}\right).$$

Next, using the fact that $\pi_t^1, \pi_t^2, \pi^* \in \Pi_t$, we have the following constraints:

$$\langle\Delta\phi_t^{*,i}, \mathbf{w}_t^{\text{proj}}\rangle + \gamma_t^{\text{known}}\|\Delta\phi_t^{*,i}\|_{(\overline{V}_t^{P^*})^{-1}} \geq 0 \quad i \in \{1, 2\}$$

$$\Leftrightarrow \langle\Delta\phi_t^{*,i}, \mathbf{w}_t^{\text{proj}}\rangle \leq \gamma_t^{\text{known}}\|\Delta\phi_t^{*,i}\|_{(\overline{V}_t^{P^*})^{-1}} \quad i \in \{1, 2\}$$

which lead to

$$2r_t \leq \gamma_t^{\text{known}}\left(\|\Delta\phi_t^{*,1}\|_{(\overline{V}_t^{P^*})^{-1}} + \|\Delta\phi_t^{*,2}\|_{(\overline{V}_t^{P^*})^{-1}}\right) + 2\gamma_t^{\text{known}}\|\Delta\phi_t^{1,2}\|_{(\overline{V}_t^{P^*})^{-1}}$$

$$\leq 4\gamma_t^{\text{known}}\|\Delta\phi_t^{1,2}\|_{(\overline{V}_t^{P^*})^{-1}},$$

hence

$$R_T = \sum_{t \in [T]} r_t \leq 2\gamma_T^{\text{known}} \sum_{t \in [T]} \|\Delta\phi_t^{1,2}\|_{(\overline{V}_t^{P^*})^{-1}}.$$

$\square$

The remaining step in our analysis is to bound the term:

$$\sum_{t \in [T]} \|\phi(\pi_t^1) - \phi(\pi_t^2)\|_{(\overline{V}_t^{P^*})^{-1}} = \sum_{t \in [T]} \left\| \mathbb{E}_{\tau \sim \mathbb{P}_{P^*}^{\pi_t^1}}[\phi(\tau)] - \mathbb{E}_{\tau \sim \mathbb{P}_{P^*}^{\pi_t^2}}[\phi(\tau)] \right\|_{(\overline{V}_t^{P^*})^{-1}}.$$

A simple approach would be to use Assumption 1, which states that the feature map $\phi$ is bounded in $\ell_2$-norm by $B$. However, our offline confidence set construction in Lemma B.1 provides a more powerful result: policies in our set have distributions that are close not only in Hellinger distance but also in the resulting feature expectations.

This is precisely why we formulated our confidence set constraint using the square root of the squared Hellinger distance - it yields a bound on the $L_2$ norm of distribution differences. Through Lemma F.1, we can translate bounds on Hellinger distance into bounds on the difference of feature expectations in the $\ell_2$-norm.

We formalize this connection in the following lemma:

**Lemma B.4** (Feature difference bound Under offline constraints). *For policies $\pi_t^1, \pi_t^2 \in \Pi_{1-\delta}^{Offline}$ selected by our algorithm at each round $t \in [T]$, the sum of feature differences measured in the data matrix norm is bounded as:*

$$\sum_{t \in [T]} \|\phi(\pi_t^1) - \phi(\pi_t^2)\|_{(\overline{V}_t^{P^*})^{-1}} \leq \sqrt{2d \cdot \log\left(1 + \frac{192 B^2 T |\mathcal{S}| \log(|\mathcal{A}| \cdot \delta^{-1})}{n \cdot d \cdot \lambda}\right)},$$

*where $d$ is the feature dimension, $B$ is the feature norm bound, $|\mathcal{S}|$ and $|\mathcal{A}|$ are the state and action space sizes, and $n$ is the number of offline samples.*

*Proof.* Again let $\Delta\phi_t^{1,2} := \phi(\pi_t^1) - \phi(\pi_t^2)$. First, we use Cauchy-Schwarz on the vectors $\mathbf{a} := (1)_{t \in [T]}$, $\mathbf{b} := (\|\Delta\phi_t^{1,2}\|)_{t \in [T]}$ and take a square root on both sides to bound:

$$\langle \mathbf{a}, \mathbf{b} \rangle = \sum_{t \in [T]} \|\Delta\phi_t^{1,2}\|_{(\overline{V}_t^{P^*})^{-1}} \leq \sqrt{T \cdot \sum_{t \in [T]} \|\Delta\phi_t^{1,2}\|_{(\overline{V}_t^{P^*})^{-1}}^2} = \|\mathbf{a}\|\|\mathbf{b}\|.$$

Then, we use the inequality

$$u \leq 2\log(1 + u) \quad u \geq 1 \implies \sum_{t \in [T]} \|\Delta\phi_t^{1,2}\|_{(\overline{V}_t^{P^*})^{-1}}^2 \leq 2 \cdot \sum_{t \in [T]} \log(1 + \|\Delta\phi_t^{1,2}\|_{(\overline{V}_t^{P^*})^{-1}}^2).$$

Using the definition of $\overline{V}_t^{P^*}$, we have

$$\overline{V}_{t+1}^{P^*} = \lambda \cdot I_{d \times d} + \sum_{s \in [t]} (\Delta\phi_s^{1,2})(\Delta\phi_s^{1,2})^\top$$

$$= \overline{V}_t^{P^*} + (\Delta\phi_t^{1,2})(\Delta\phi_t^{1,2})^\top$$

$$= (\overline{V}_t^{P^*})^{1/2}\left(I + (\overline{V}_t^{P^*})^{-1/2}(\Delta\phi_t^{1,2})(\Delta\phi_t^{1,2})^\top(\overline{V}_t^{P^*})^{-1/2}\right)(\overline{V}_t^{P^*})^{1/2}.$$

Using the properties of the determinant:

$$\det(\overline{V}_{t+1}^{P^*}) = \det(\overline{V}_t^{P^*}) \cdot \det(I + (\overline{V}_t^{P^*})^{-1/2}(\Delta\phi_t^{1,2})(\Delta\phi_t^{1,2})^\top(\overline{V}_t^{P^*})^{-1/2})$$

$$= \det(\overline{V}_t^{P^*}) \cdot (1 + \|\Delta\phi_t^{1,2}\|_{(\overline{V}_t^{P^*})^{-1}}^2)$$

$$= \det(V_0) \cdot \prod_{s \in [t]} (1 + \|\Delta\phi_s^{1,2}\|_{(\overline{V}_s^{P^*})^{-1}}^2).$$

Taking the log on both sides, this holds iff

$$\log\left[\frac{\det(\overline{V}_{t+1}^{P^*})}{\det(V_0)}\right] = \sum_{s\in[t]}\log(1+\|\Delta\phi_s^{1,2}\|^2_{(\overline{V}_s^{P^*})^{-1}}).$$

It also holds that:

$$\det(\overline{V}_{t+1}^{P^*}) = \prod_{i\in[d]}\lambda_i \le \left(\frac{1}{d}\cdot\mathrm{Tr}\{\overline{V}_{t+1}^{P^*}\}\right)^d.$$

Using linearity of trace:

$$\mathrm{Tr}\{\overline{V}_{t+1}^{P^*}\} = \mathrm{Tr}\{\lambda I\} + \sum_{s\in[t]}\mathrm{Tr}\{(\Delta\phi_s^{1,2})(\Delta\phi_s^{1,2})^\top\}$$

$$= d\lambda + \sum_{s\in[t]}\|\Delta\phi_s^{1,2}\|_2^2.$$

Applying the corrected bound from Lemma F.1:

$$\|\Delta\phi_t^{1,2}\|_2^2 \le (2\sqrt{2}\cdot B\cdot\sqrt{H^2(\mathbb{P}_{P^*}^{\pi_t^1},\mathbb{P}_{P^*}^{\pi_t^2})})^2$$

$$\le 8B^2\cdot\frac{24\cdot|\mathcal{S}|\cdot\log(|\mathcal{A}|\cdot\delta^{-1})}{n}$$

$$= \frac{192B^2|\mathcal{S}|\log(|\mathcal{A}|\cdot\delta^{-1})}{n}.$$

Using this tighter bound in our trace calculation:

$$\mathrm{Tr}\{\overline{V}_{t+1}^{P^*}\} \le d\cdot\lambda + t\cdot\frac{192B^2|\mathcal{S}|\log(|\mathcal{A}|\cdot\delta^{-1})}{n}$$

$$= d\lambda\left(1+\frac{192B^2t|\mathcal{S}|\log(|\mathcal{A}|\cdot\delta^{-1})}{nd\lambda}\right).$$

Hence:

$$\log\left[\frac{\det(\overline{V}_{t+1}^{P^*})}{\det(V_0)}\right] \le d\cdot\log\left(\frac{\mathrm{Tr}\{\overline{V}_{t+1}^{P^*}\}}{d}\right)$$

$$= d\cdot\log\left(\lambda\left(1+\frac{192B^2t|\mathcal{S}|\log(|\mathcal{A}|\cdot\delta^{-1})}{n\cdot d\cdot\lambda}\right)\right)$$

$$= d\cdot\log(\lambda) + d\cdot\log\left(1+\frac{192B^2t|\mathcal{S}|\log(|\mathcal{A}|\cdot\delta^{-1})}{nd\lambda}\right).$$

Since $\det(V_0) = \lambda^d$, the first logarithmic term cancels out:

$$\log\left[\frac{\det(\overline{V}_{t+1}^{P^*})}{\det(V_0)}\right] = d\cdot\log\left(1+\frac{192B^2t|\mathcal{S}|\log(|\mathcal{A}|\cdot\delta^{-1})}{nd\lambda}\right).$$

Therefore:

$$\sum_{t\in[T]}\|\Delta\phi_t^{1,2}\|^2_{(\overline{V}_t^{P^*})^{-1}} \le 2\cdot\log\left[\frac{\det(\overline{V}_{T+1}^{P^*})}{\det(V_0)}\right]$$

$$\le 2d\cdot\log\left(1+\frac{192B^2T|\mathcal{S}|\log(|\mathcal{A}|\cdot\delta^{-1})}{nd\lambda}\right).$$

Taking the square root:

$$\sum_{t \in [T]} \|\Delta\phi_t^{1,2}\|_{(\overline{V}_t^{P^*})^{-1}} \leq \sqrt{2d \cdot \log\left(1 + \frac{192B^2 T|\mathcal{S}| \log(|\mathcal{A}| \cdot \delta^{-1})}{nd\lambda}\right)}.$$

$\square$

**Theorem B.5** (Regret Analysis for BRIDGE under Known Model). *Let $\delta \leq 1/e$ and $\lambda \geq \frac{B}{\kappa}$. Then, with probability at least $1 - \delta$, the expected regret of Algorithm 2 is bounded by:*

$$R_T \leq (2\kappa\beta_T(\delta) + \alpha_{d,T}(\delta))\sqrt{2d \cdot \log\left(1 + \frac{192B^2 T|\mathcal{S}| \log(|\mathcal{A}| \cdot \delta^{-1})}{nd\lambda}\right)}$$

*In asymptotic notation, this becomes:*

$$R_T = \tilde{\mathcal{O}}\left(\left(W\sqrt{\kappa B} + WB\right) d \log(TB/\kappa\delta)\sqrt{\log\left(1 + \frac{T|\mathcal{S}|}{nd}\right)}\right)$$

*where the probability parameter $\delta$ accounts for the events*

$$
\begin{aligned}
E_{w^*} &\quad\rightarrow\quad Lemma\ D.1\\
E_{\overline{V}_T^{P^*}} &\quad\rightarrow\quad Lemma\ D.2\\
E_{offline} := \{\pi^* \in \Pi_{1-\delta}^{Offline}\} &\quad\rightarrow\quad Lemma\ B.1
\end{aligned}
$$

*Remark* B.6. This result demonstrates a significant improvement over Saha et al. (2023)'s bound of $\tilde{\mathcal{O}}\left(\left(W\sqrt{\kappa B} + WB\right) d \log(TB/\kappa\delta)\sqrt{T}\right)$. The key advantage lies in the term $\sqrt{\log(1 + \frac{T|\mathcal{S}|}{n})}$, which approaches zero as $nd \to \infty$, potentially yielding constant regret.

### B.3 PRACTICAL REGRET ANALYSIS WITH FIXED OFFLINE DATA (KNOWN MODEL)

For a fixed offline dataset of size $n$, our regret bound scales with horizon $T$ as:

$$R_T = \tilde{\mathcal{O}}\left(\Gamma \cdot \sqrt{\log\left(1 + \frac{CT|\mathcal{S}|}{n \cdot d}\right)}\right)$$

where $\Gamma = (W\sqrt{\kappa B} + WB)d \log(TB/\kappa\delta)$. This bound reveals three distinct regimes:

1. **Small $T$ Regime** ($T|\mathcal{S}| \ll n \cdot d$): Using $\log(1 + x) \approx x$ for small $x$:
$$R_T = \mathcal{O}\left(\Gamma \cdot \sqrt{\frac{T|\mathcal{S}|}{n \cdot d}}\right) = \mathcal{O}\left(\Gamma \cdot \sqrt{T} \cdot \frac{|\mathcal{S}|^{1/2}}{\sqrt{n \cdot d}}\right)$$

2. **Transition Regime** ($T|\mathcal{S}| \approx n \cdot d$):
$$R_T = \mathcal{O}(\Gamma) = \mathcal{O}\left((W\sqrt{\kappa B} + WB)d \log(TB/\kappa\delta)\right)$$

3. **Large $T$ Regime** ($T|\mathcal{S}| \gg n \cdot d$):
$$R_T = \mathcal{O}\left(\Gamma \cdot \sqrt{\log(T)}\right)$$

These regimes highlight two key insights: (1) with sufficient offline data ($n = \Omega(\frac{T|\mathcal{S}|}{d})$), regret dramatically improves from $\mathcal{O}(\sqrt{\log(T)})$ to $\mathcal{O}(1)$ in the dependence on $T$; and (2) feature dimension $d$ amplifies the value of offline data, allowing the same regret reduction with $\sqrt{d}$ times less data. This explains why high-dimensional problems may benefit more significantly from offline data.

As $n$ increases, regret transitions from logarithmic ($\mathcal{O}(\log(T))$) to sublinear ($\mathcal{O}(\sqrt{T/n})$) and eventually approaches $\mathcal{O}(1)$ when $n \gg \frac{T|\mathcal{S}|}{d}$. In the limiting case where $n \to \infty$, exploration becomes unnecessary, and regret is bounded only by statistical error in the offline estimation.

## C  OFFLINE ESTIMATION

### C.1  MAXIMUM LIKELIHOOD FOR DENSITY ESTIMATION

In this section, we present Maximum Likelihood Estimation (MLE) for density estimation that forms the foundation of our concentration results. While these results are presented more extensively in Foster et al. (2024), we include them here for completeness and readability.

The analysis of MLE relies on standard concentration techniques following the well-established work of van de Geer (2000) and Zhang (2006), enhanced by new Freedman-type concentration inequalities developed in Foster et al. (2024) (Appendix B).

The key proof strategy connects MLE analysis to information-theoretic measures via *Rényi divergence* of order $1/2$, written as $D_{1/2}(P\|Q)$. Specifically, the approach bounds expressions of the form $-n \cdot \log(\mathbb{E}_{z \sim g^*}[e^{\frac{1}{2}\log(g(z)/g^*(z))}])$, which equals $\frac{n}{2} \cdot D_{1/2}(g\|g^*)$. This term is bounded using Freedman-type inequalities for adapted sequences, which provide high-probability bounds of the form $\sum_{t=1}^{T'} -\log(\mathbb{E}_{t-1}[e^{-X_t}]) \leq \sum_{t=1}^{T'} X_t + \log(\delta^{-1})$. When combined with union bounds over $\varepsilon$-nets, this yields tight concentration results for the entire function class. The approach also leverages connections to Hellinger distance through the identity $H^2(g, g^*) = 1 - \int \sqrt{g(z)g^*(z)}dz$, providing geometrically interpretable guarantees.

To handle infinite classes, we introduce a tailored notion of covering number for log-loss:

**Definition C.1** (Log-Covering Number). For a class $\mathcal{G} \subset \Delta(\mathcal{X})$, the class $\mathcal{G}' \subset \mathcal{X}$ is an $\epsilon-$cover if for all $g \in \mathcal{G}$, there exists $g' \in \mathcal{G}'$ such that $\forall x \in \mathcal{X}$

$$\log(g(x)/g'(x)) \leq \epsilon$$

The size of such cover is defined by $\mathcal{N}_{\log}(\mathcal{G}, \epsilon)$.

Consider the data $\mathbb{D}_n = \{x_i\}_{i \in [n]}$ consisting of i.i.d copies of $x \sim g^*$ where $g^* \in \Delta(\mathcal{X})$. We have a class $\mathcal{G} \subseteq \Delta(\mathcal{X})$ that may or may not contain $g^*$. The density MLE estimator is defined as

$$\hat{g} = \arg\max_{g \in \mathcal{G}} \sum_{i \in [n]} \log(g(x_i)) \tag{5}$$

**Lemma C.2** (Maximum Likelihood Estimator Bound). *The maximum likelihood estimator in Equation* (5) *has that with probability at least $1 - \delta$,*

$$H^2(\hat{g}, g^*) \leq \inf_{\varepsilon > 0} \left\{ \frac{6\log(2\mathcal{N}_{\log}(\mathcal{G}, \varepsilon)/\delta^{-1})}{n} + 4\varepsilon \right\} + 2\inf_{g \in \mathcal{G}} \log(1 + D_{\chi^2}(g^*\|g))$$

*In particular, if $\mathcal{G}$ is finite, the maximum likelihood estimator satisfies*

$$H^2(\hat{g}, g^*) \leq \frac{6\log(2|\mathcal{G}|/\delta^{-1})}{n} + 2\inf_{g \in \mathcal{G}} \log(1 + D_{\chi^2}(g^*\|g))$$

*Note that the term $\inf_{g \in \mathcal{G}} \log(1 + D_{\chi^2}(g^*\|g))$ corresponds to the mis-specification error, and is zero if $g^* \in \mathcal{G}$.*

### C.2  MLE OBJECTIVE OF DATASET OF INDEPENDENT TRAJECTORIES

Given a data set of reward free trajectories $\mathbb{D}_n^H = \{\tau_i\}_{i \in [H]}$ of $n$ trajectories of length $H$ where $\{\tau_i\} \sim_{i.i.d} \tau \sim \mathbb{P}_{P^*}^{\pi^*}$. The distribution $\mathbb{P}_{P^*}^{\pi^*}$ is assumed to be continuous w.r.t to the Lebesgue measure. It is characterized by the policy density $\pi = \{\pi_i\}_{i \in [H]} \in \Pi$ and the stationary transition density $P = \mathcal{P}$ where $\Pi, \mathcal{P}$ characterize the policy and transition density spaces. The log-likelihood of the set with for a policy $\pi$ and a transition $P$ reads:

$$l_n(\pi, P) = \frac{1}{n} \sum_{i \in [n]} \log \left[ P(s_1^i) \cdot \pi_1(a_1^i, s_1^i) \prod_{1 < j \leq H} P(s_j^i|s_{j-1}^i, a_{j-1}^i)\pi_j(a_j^i|s_j^i) \right]$$

The maximum likelihood objective over the density class $\{\mathbb{P}_P^\pi\}_{\pi \in \Pi, P \in \mathcal{P}}$ for the dataset $\mathbb{D}_n^H$

$$\arg\max_{\pi \in \Pi, P \in \mathcal{P}} \sum_{i \in [n]} \sum_{j \in [H]} \left( \log[\pi_i(a_j^i|s_j^i)] \right) + \sum_{i \in [n]} \sum_{j=0}^{H} \left( \log[P(s_{j+1}^i|s_j^i, a_j^i)] \right) \tag{6}$$

## C.3 CONCENTRATION BOUNDS

In this section, we provide concentration bounds for the MLE estimators of the policies and the transition model, as well as for our notion of concentrability coefficient. The important takeaway is that the control of the error, i.e., the decay of these concentration bounds depends only on values known to the user. This will allow us to compute confidence policy sets based on these bounds.

### C.3.1 POLICY ESTIMATION

Define the log-loss behavioral cloning estimator for dataset $\mathbb{D}_n^H$ as described in Appendix C.2 as

$$\pi^{\mathrm{BC}} = \arg\max_{\pi \in \Pi} \sum_{i \in [n]} \sum_{h \in [H]} \log(\pi_h(a_h^i | s_h^i)) \tag{7}$$

which is from Equation (6) equivalent to performing maximum density estimation over the density class $\{\mathbb{P}_{P^*}^\pi\}_{\pi \in \Pi}$. Similar to Theorem C.1 (cf. Foster et al. (2024)), define the following

**Definition C.3** (Policy covering number). For a class $\Pi \subset \{\pi_h : \mathcal{S} \to \Delta(\mathcal{A})\}$, we say that $\Pi' \subset \{\pi_h : \mathcal{S} \to \Delta(\mathcal{A})\}$ is an $\epsilon-$cover if for all $\pi \in \Pi$ there exists $\pi' \in \Pi'$ such that

$$\log\left(\frac{\pi_h(a|s)}{\pi_h'(a|s)}\right) \leq \epsilon \quad \forall (s, a, h) \in \mathcal{S} \times \mathcal{A} \times [H]$$

We denote the size of the smallest such cover as $\mathcal{N}_{pol}(\Pi, \epsilon)$

We state the following theorem from (Foster et al., 2024, Appendix C):

**Theorem C.4** (Generalization bound for log-loss BC, general policies). *The log-loss BC estimator (Equation (7)) satisfies with probability $\geq 1 - \delta$*

$$H^2(\mathbb{P}_{P^*}^{\pi^{BC}}, \mathbb{P}_{P^*}^{\pi^*}) \leq \inf_{\epsilon}\left\{\frac{6\log(2\mathcal{N}_{pol}(\Pi, \epsilon/H)\delta^{-1})}{n} + \epsilon\right\}$$

*in particular, if $\Pi$ is finite,*

$$H^2(\mathbb{P}_{P^*}^{\pi^{BC}}, \mathbb{P}_{P^*}^{\pi^*}) \leq \frac{6 \cdot \log(2 \cdot |\Pi| \cdot \delta^{-1})}{n}.$$

*Proof.* See (Foster et al., 2024, Appendix C). □

**Corollary C.5** (Generalization bound for log-loss BC, deterministic, stationary & tabular policies). *If $\Pi = \Pi_S^D$, i.e the set of deterministic tabular policies, for the log-loss BC estimator Equation (7) it holds that with probability at least $1 - \delta$,*

$$H^2(\mathbb{P}_{P^*}^{\pi^{BC}}, \mathbb{P}_{P^*}^{\pi^*}) \leq \frac{6 \cdot |\mathcal{S}| \cdot \log(|\mathcal{A}| \cdot \delta^{-1})}{n}.$$

*Proof.* We have $|\Pi_S^D| = |\mathcal{A}|^{|\mathcal{S}|}$. □

If we have stochastic rather than deterministic policies, we need to determine $\log(\mathcal{N}_{pol}(\Pi_S, \epsilon))$. This can be accomplished using a discretisation argument, where we create a finite $\epsilon$-net that approximates the continuous space of stochastic policies within the desired error tolerance.

### C.3.2 TRANSITION MODEL ESTIMATION

Here we can give a similar argument as for the policy log loss BC estimator. We define the following estimator

$$\hat{P} = \arg\max_{P \in \mathcal{P}} \sum_{i \in [n]} \sum_{j=0}^{H}\left(\log[P(s_{j+1}^i | s_j^i, a_j^i)]\right) \tag{8}$$

which is from Eq. 6 equivalent to performing maximum density estimation over the density class $\{\mathbb{P}_P^{\pi^*}\}_{P \in \mathcal{P}}$. Similarly, we define the following notion of covering

**Definition C.6** (Log covering number for stationary transitions). For a class of stationary transition probability functions $\mathcal{P} \subset \{P : \mathcal{S} \times \mathcal{A} \to \Delta(\mathcal{S})\}$ we define that $\mathcal{P}' \subset \{P : \mathcal{S} \times \mathcal{A} \to \Delta(\mathcal{S})\}$ is an $\epsilon$-cover if for all $P \in \mathcal{P}$ there exists $P' \in \mathcal{P}'$ such that

$$\log\left(\frac{P(s'|s,a)}{P'(s'|s,a)}\right) \leq \epsilon \quad \forall(s',s) \in \mathcal{S}, a \in \mathcal{A}$$

We denote the size of the smallest such cover by $\mathcal{N}_{trans}(\mathcal{P}, \epsilon)$.

**Assumption 4** (Realisability of transitions). *We assume the true transition density to be in the model class i.e $P^* \in \mathcal{P}$*

We can now give a similar guarantee as for the log loss policy estimate but for the transition estimate

**Theorem C.7** (Generalisation bound for MLE transition estimator). *The MLE for transitions (Eq. (8)) satisfies with probability at least $1 - \delta$*

$$H^2(\mathbb{P}_{\hat{P}}^{\pi^*}, \mathbb{P}_{P^*}^{\pi^*}) \leq \inf_\epsilon \left\{ \frac{6 \log(2\mathcal{N}_{trans}(\Pi, \epsilon/H)\delta^{-1})}{n} + \epsilon \right\}$$

*Proof.* Given a valid $\epsilon$-cover of $\mathcal{P}$ from definition C.6, we have

$$\log\left(\frac{\mathbb{P}_P^\pi}{\mathbb{P}_{P'}^\pi}\right) = \sum_{h=1}^H \log\left(\frac{P(s_{h+1}|s_h, a_h)}{P'(s_{h+1}|s_h, a_h)}\right) \leq \epsilon \cdot H.$$

This means that we get a valid $\epsilon \cdot H$ cover for the trajectory density class. The bound follows as a direct application of Lemma C.2. □

**Lemma C.8** (Log covering number for stationary, tabular & stochastic transitions). *For a class of stationary transition probability functions $\mathcal{P} \subset \{P : \mathcal{S} \times \mathcal{A} \to \Delta(\mathcal{S})$ where $|\mathcal{S} \times \mathcal{A}|$ is finite (tabular MDP), the $\epsilon$-cover from Definition C.6 satisfies:*

$$\log(\mathcal{N}_{trans}(\mathcal{P}, \epsilon)) \leq |\mathcal{S}| \cdot |\mathcal{A}| \cdot (|\mathcal{S}| - 1) \cdot \log\left(\frac{1}{\epsilon} + 1\right)$$

*Proof.* The proof follows a standard geometric discretization argument for a finite class of functions (see Chapter 5 Wainwright (2019)) . For a given $\epsilon > 0$ we construct a geometric grid:

$$\mathcal{G}_\epsilon = \left\{\delta, \delta \cdot \exp(\epsilon/2), \delta \exp(\epsilon), \delta \exp(3\epsilon/2), \ldots, \delta \exp(k\epsilon/2)\right\}$$

where $\delta > 0$ is the minimum probability and $k$ is chosen such that the grid represents a discretization of the continuous interval $[0, 1]$ i.e

$$\delta \exp(k\epsilon/2) \implies k \geq \frac{2\log(1/\delta)}{\epsilon}$$

Thus the grid size is at most:

$$|\mathcal{G}_\epsilon| \leq \left\lceil \frac{2\log(1/\delta)}{\epsilon} \right\rceil + 1$$

For each state action pair $(s, a)$ define $P(s_i|s, a) = p_i$ for $i \in |\mathcal{S}|$. Note that for the first $1, \ldots, |\mathcal{S}| - 1$ there exist $q_i := P'(s_i|s, a) \in \mathcal{G}_\epsilon$ that satisfies by construction

$$\exp(-\epsilon/2) \leq \frac{p_i}{q_i} \leq \exp(\epsilon/2)$$

For the last state $i = |\mathcal{S}|$, we need to determine $q_{|\mathcal{S}|}$ close enough to $p_{|\mathcal{S}|}$.
Let define $S_q := \sum_i^{|\mathcal{S}-1|} q_i$ and $S_p := \sum_i^{|\mathcal{S}-1|} p_i$ we have the constraint

$$p_{|\mathcal{S}|} = 1 - S_p$$
$$q_{|\mathcal{S}|} = 1 - S - q$$

From the bound on the first $1 - |\mathcal{S}|$ elements we have

$$S_q \exp(-\epsilon/2) \leq S_p \leq S_q \exp(\epsilon/2)$$

For the ratio of the last probability

$$\frac{p_{|\mathcal{S}|}}{q_{|\mathcal{S}|}} = \frac{1 - S_p}{1 - S_q},$$

we have the following condition such that we have $\frac{p_{|\mathcal{S}|}}{q_{|\mathcal{S}|}} \geq \exp(-\epsilon)$

$$S_q \leq \frac{1 - \exp(-\epsilon)}{\exp(\epsilon/2) - \exp(-\epsilon)}.$$

Similarly for the upper bound, we have the condition

$$S_q \leq \frac{\exp(\epsilon) - 1}{\exp(\epsilon) - \exp(-\epsilon/2)}.$$

Combining both constraints, we get

$$S_q \leq \min \left\{ \frac{\exp(\epsilon) - 1}{\exp(\epsilon) - \exp(-\epsilon/2)}, \frac{\exp(\epsilon) - 1}{\exp(\epsilon) - \exp(-\epsilon/2)} \right\}.$$

By Taylor approximation, this boils down to

$$S_q \leq \frac{2}{3}.$$

Hence we select only the combination of points that satisfies

$$\frac{2}{3} \leq S_q \leq 1 - \delta.$$

It remains to count the number of points in our cover, i.e, the first $|\mathcal{S} - 1|$ that satisfy our constraint

$$(\text{number of grid points})^{|\mathcal{S}-1|} \leq |\mathcal{G}_\epsilon|^{|\mathcal{S}|-1} \leq \left( \left\lceil \frac{2 \log(1/\delta)}{\epsilon} \right\rceil + 1 \right)^{|\mathcal{S}|-1}.$$

Going across all state action pairs, and taking the logarithm, we get

$$\log(\mathcal{N}_{trans}(\mathcal{P}, \epsilon)) \leq |\mathcal{S}||\mathcal{A}|(\mathcal{S} - 1) \log \left( \left\lceil \frac{2 \log(1/\delta)}{\epsilon} \right\rceil + 1 \right).$$

Choosing $\delta = \mathcal{O}(\epsilon)$ yield the result. $\qquad \square$

**Corollary C.9** (Stochastic, stationary, tabular transition setting). *For finite $|\mathcal{S} \times \mathcal{A}|$ (tabular setting) and assuming the transition density class to be stochastic and stationary we have with probability at least $1 - \delta$*

$$H^2(\mathbb{P}_{\hat{P}}^{\pi^*}, \mathbb{P}_{P^*}^{\pi^*}) \leq \mathcal{O} \left( \frac{|\mathcal{S}|^2 |\mathcal{A}| \log(nH\delta^{-1})}{n} \right)$$

*where for the theoretical optimal constant $C_{theory} \approx 6$*

*Proof.* From our lemma on the covering number of transition functions, we have:

$$\log \mathcal{N}_{\text{trans}}(\mathcal{P}, \varepsilon/H) \leq |\mathcal{S}||\mathcal{A}|(|\mathcal{S}| - 1) \log \left( \frac{H}{\varepsilon} + 1 \right)$$

For large $\frac{H}{\varepsilon}$, we can approximate:

$$\log \left( \frac{H}{\varepsilon} + 1 \right) \approx \log \left( \frac{H}{\varepsilon} \right)$$

Substituting this into our bound:

$$H^2(\mathbb{P}_{\hat{P}}^{\pi^*}, \mathbb{P}_{P^*}^{\pi^*}) \leq \inf_{\varepsilon > 0} \left\{ \frac{6 \log(2) + 6|\mathcal{S}||\mathcal{A}|(|\mathcal{S}| - 1) \log \left( \frac{H}{\varepsilon} \right) + 6 \log(\delta^{-1})}{n} + \varepsilon \right\}$$

$$= \inf_{\varepsilon > 0} \left\{ \frac{6 \log(2) + 6D \log(H) - 6D \log(\varepsilon) + 6 \log(\delta^{-1})}{n} + \varepsilon \right\}$$

where $D = |\mathcal{S}||\mathcal{A}|(|\mathcal{S}| - 1)$ for brevity.

To find the optimal $\varepsilon$, we differentiate with respect to $\varepsilon$ and set to zero:

$$\frac{d}{d\varepsilon}\left[\frac{6\log(2) + 6D\log(H) - 6D\log(\varepsilon) + 6\log(\delta^{-1})}{n} + \varepsilon\right] = -\frac{6D}{n\varepsilon} + 1 = 0$$

$$\Rightarrow \varepsilon_{\text{opt}} = \frac{6D}{n} = \frac{6|\mathcal{S}||\mathcal{A}|(|\mathcal{S}| - 1)}{n}$$

Substituting this optimal value back:

$$H^2(\mathbb{P}_{\hat{P}}^{\pi^*}, \mathbb{P}_{P^*}^{\pi^*}) \leq \frac{6\log(2) + 6D\log(H) - 6D\log\left(\frac{6D}{n}\right) + 6\log(\delta^{-1})}{n} + \frac{6D}{n}$$

$$= \frac{6\log(2) + 6D\log(H) - 6D\log(6D) + 6D\log(n) + 6\log(\delta^{-1}) + 6D}{n}$$

$$= \frac{6\log(2) + 6\log(\delta^{-1}) + 6D\log\left(\frac{nH}{6D}\right) + 6D}{n}$$

For large state spaces where $|\mathcal{S}| - 1 \approx |\mathcal{S}|$, and defining $\tilde{D} = |\mathcal{S}|^2|\mathcal{A}|$, this becomes:

$$H^2(\mathbb{P}_{\hat{P}}^{\pi^*}, \mathbb{P}_{P^*}^{\pi^*}) \leq \frac{6\log(2) + 6\log(\delta^{-1}) + 6\tilde{D}\log\left(\frac{nH}{6\tilde{D}}\right) + 6\tilde{D}}{n}$$

For large $n$ and $\tilde{D}$, the dominant term is $\frac{6\tilde{D}\log(nH)}{n}$, and we can combine the logarithmic terms to get:

$$H^2(\mathbb{P}_{\hat{P}}^{\pi^*}, \mathbb{P}_{P^*}^{\pi^*}) = \mathcal{O}\left(\frac{|\mathcal{S}|^2|\mathcal{A}|\log(nH\delta^{-1})}{n}\right)$$

Note that the constant 6 appears in the full derivation. This completes the proof. $\square$

### C.3.3 CONCENTRABILITY COEFFICIENT UPPER BOUND

**Definition C.10** (Concentrability coefficient). We define the following quantity as the *concentrability coefficient*:

$$C(\pi^{\text{BC}}, \pi^*) = \sup_{t \in [H]} \sup_{(s,a) \in \mathcal{S} \times \mathcal{A}: d_{P^*}^{\pi^*,t}(s,a) > 0} \frac{d_{P^*}^{\pi,t}(s,a)}{d_{P^*}^{\pi^*,t}(s,a)}.$$

It measures the maximum ratio between the state-action distributions induced by policies $\pi^{\text{BC}}$ and $\pi^*$ under the true dynamics $P^*$.

**Assumption 5** (Minimum visitation probability). *There exists a constant $\gamma_{\min} > 0$ such that for all state-action-time tuples with non-zero probability under the optimal policy:*

$$\min_{(s,a,t): d_{P^*}^{\pi^*,t}(s,a) > 0} d_{P^*}^{\pi^*,t}(s,a) \geq \gamma_{\min}$$

**Lemma C.11** (Concentrability coefficient bound). *Consider a policy estimator $\pi^{BC}$ satisfying*

$$H^2(\mathbb{P}_{P^*}^{\pi^{BC}}, \mathbb{P}_{P^*}^{\pi^*}) \leq R.$$

*Then, under Assumption 3, the concentration coefficient is bounded by:*

$$C(\pi^{BC}, \pi^*) \leq 1 + \frac{2\sqrt{R}}{\gamma_{\min}}.$$

*Proof.* We will begin by upper bounding the numerator using the condition on the Hellinger distance, followed by lower bounding the denominator using concentration.

For the upper bound, note that

$$\sup_{a,s} |d_{P*}^{\pi^{\mathrm{BC}},t} - d_{P*}^{\pi^*,t}| = 2 \cdot TV(d_{P*}^{\pi^{\mathrm{BC}},t}, d_{P*}^{\pi^*,t}).$$

Recall that the state-action distribution $d_{P*}^{\pi,t}(s,a)$ is the marginal distribution of the trajectory distribution at timestep $t$. Explictly:

$$d_{P*}^{\pi,t}(s,a) = \int_{\tau_{-t}} \mathbb{P}_{P*}^{\pi}(\tau) \, d\tau_{-t} =: \mathbb{P}_{P*}^{\pi}(s_t = s, a_t = a),$$

where $\tau_{-t}$ denotes all time steps in the trajectory except for time $t$, and $\mathbb{P}_{P*}^{\pi}(\tau)$ is the probability of trajectory $\tau$ under policy $\pi$ and dynamics $P^*$.

It follows that

$$\begin{aligned} TV(d_{P*}^{\pi^{\mathrm{BC}},t}, d_{P*}^{\pi^*,t}) &= 2 \cdot TV(\mathbb{P}_{P*}^{\pi^{\mathrm{BC}}}(s_t = s, a_t = a), \mathbb{P}_{P*}^{\pi^*}(s_t = s, a_t = a)) \\ &\leq 2 \cdot TV(\mathbb{P}_{P*}^{\pi^{\mathrm{BC}}}, \mathbb{P}_{P*}^{\pi^*}) \\ &\leq 2 \cdot \sqrt{H^2(\mathbb{P}_{P*}^{\pi^{\mathrm{BC}}}, \mathbb{P}_{P*}^{\pi^*})} \leq 2\sqrt{R}. \end{aligned}$$

By assumption 1, we have a lower bound on the minimum state-action visitation probability:

$$\min_{(s,a,t):d_{P*}^{\pi^*,t}(s,a)>0} d_{P*}^{\pi^*,t}(s,a) \geq \gamma_{\min}.$$

Finally, we combine the upper bound on the numerator and the lower bound from assumption 1 to get $\forall (a,s,t) \quad \text{s.t} \quad d_{P*}^{\pi^*,t}(a,s) > 0$:

$$\begin{aligned} C(\pi^{\mathrm{BC}}, \pi^*) &= 1 + \frac{\sup_{a,s,t} |d_{P*}^{\pi^{\mathrm{BC}},t}(s,a) - d_{P*}^{\pi^*,t}(s,a)|}{\inf_{a,s,t} d_{P*}^{\pi^*,t}(s,a)} \\ &\leq 1 + \frac{2\sqrt{R}}{\inf_{a,s,t} d_{P*}^{\pi^*,t}(s,a)} \\ &\leq 1 + \frac{2\sqrt{R}}{\gamma_{\min}}. \end{aligned}$$

This completes the proof, giving us a deterministic bound on the concentration coefficient that depends on the Hellinger distance between trajectory distributions and the minimum visitation probability of the optimal policy. $\square$

### C.3.4 CONFIDENCE SET CONSTRUCTION

In this section, we will derive a distributional confidence set on the trajectory space in the form of a hellinger ball, accounting for the error of the MLE density estimates $\pi^{\mathrm{BC}}$ and $\hat{P}$. We start by presenting the following in between result

**Lemma C.12** (Technical results)**.** *Assume a finite state-action space $\mathcal{S} \times \mathcal{A}$ (tabular setting). $\forall \pi \in \Pi$, where $\pi^*$ is the true policy and $P^*, \hat{P}$ are the true and estimated transition models, the following bound holds:*

$$H^2(\mathbb{P}_{\hat{P}}^{\pi}, \mathbb{P}_{P*}^{\pi}) \leq H \cdot C(\pi, \pi^*) \cdot H^2(\mathbb{P}_{\hat{P}}^{\pi^*}, \mathbb{P}_{P*}^{\pi^*}),$$

*where*

$$C(\pi, \pi^*) = \sup_{t \in [H]} \sup_{(s,a) \in \mathcal{S} \times \mathcal{A}: d_{P*}^{\pi^*,t}(s,a)>0} \frac{d_{P*}^{\pi,t}(s,a)}{d_{P*}^{\pi^*,t}(s,a)}.$$

*Proof.* We derive the proof in three steps:

Step 1:
$$H^2(\mathbb{P}_{\hat{P}}^{\pi}, \mathbb{P}_{P^*}^{\pi}) \leq \sum_{t \in [H-1]} \mathbb{E}_{(s_t, a_t) \sim d_{P^*}^{\pi, t}} \left[ H^2\left( \hat{P}(\cdot|s_t, a_t), P^*(\cdot|s_t, a_t) \right) \right].$$

Step 2:
$$\mathbb{E}_{(s_t, a_t) \sim d_{P^*}^{\pi, t}} \left[ H^2\left( \hat{P}(\cdot|s_t, a_t), P^*(\cdot|s_t, a_t) \right) \right]$$
$$\leq C(\pi, \pi^*) \cdot \mathbb{E}_{(s_t, a_t) \sim d_{P^*}^{\pi^*, t}} \left[ H^2\left( \hat{P}(\cdot|s_t, a_t), P^*(\cdot|s_t, a_t) \right) \right].$$

Step 3:
$$H^2(\mathbb{P}_{\hat{P}}^{\pi^*}, \mathbb{P}_{P^*}^{\pi^*}) \geq \frac{1}{H} \cdot \mathbb{E}_{(s_t, a_t) \sim d_{P^*}^{\pi^*, t}} \left[ H^2\left( \hat{P}(\cdot|s_t, a_t), P^*(\cdot|s_t, a_t) \right) \right].$$

Proof Step 1:

$$H^2(\mathbb{P}_{\hat{P}}^{\pi}, \mathbb{P}_{P^*}^{\pi}) = 1 - \int_{\mathcal{T}} 1 - \mu_0(s_0) \prod_{t=0}^{H-1} \pi(a_t|s_t) \sqrt{\hat{P}(s_{t+1}|a_t, s_t) P^*(s_{t+1}|s_t, a_t)} d\tau$$

$$= 1 - \int_{\mathcal{T}} p_{P^*}^{\pi}(\tau) \cdot \frac{\mu_0(s_0) \prod_{t=0}^{H-1} \pi(a_t|s_t) \sqrt{\hat{P}(s_{t+1}|a_t, s_t) P^*(s_{t+1}|s_t, a_t)}}{\mu_0(s_0) \prod_{t=0}^{H-1} \pi(a_t|s_t) P^*(s_{t+1}|s_t, a_t)} d\tau$$

$$= 1 - \int_{\mathcal{T}} p_{P^*}^{\pi}(\tau) \cdot \prod_{t=0}^{H-1} \sqrt{\frac{\hat{P}(s_{t+1}|s_t, a_t)}{P^*(s_{t+1}, s_t, a_t)}} d\tau.$$

Next, we define

$$\alpha_t(s_{t+1}, a_t, s_t) := \sqrt{\frac{\hat{P}(s_{t+1}|s_t, a_t)}{P^*(s_{t+1}, s_t, a_t)}},$$

$$\gamma_t(s_t, a_t) := \int_{s'} \sqrt{\hat{P}(s_{t+1}|s_t, a_t) P^*(s_{t+1}, s_t, a_t)} ds' = \int_{s'} P^*(s'|s_t, a_t) \cdot \alpha_t(s', s_t, a_t) ds'.$$

Notice that $\gamma_t$ is a BC coefficient, i.e,
$$1 - \gamma_t(s_t, a_t) = H^2\left( \hat{P}(\cdot|s_t, a_t), P^*(\cdot|s_t, a_t) \right)$$

Using notation above:

$$H^2(\mathbb{P}_{\hat{P}}^{\pi}, \mathbb{P}_{P^*}^{\pi}) = 1 - \mathbb{E}_{\tau \sim \mathbb{P}_{P^*}^{\pi}} \left[ \prod_{t=0}^{H-1} \alpha_t(s_{t+1}, s_t, a_t) \right]$$

Using conditional expectation (law of iterated expectation) we change the distribution in the expectation from $\mathbb{P}_{P^*}^{\pi}$ to the so called state-action distribution $d_{P^*}^{\pi, t}$. To show this argument, we show it for state action pair $(a_0, s_0, s_1)$. The rest follows by using the same idea:

$$\mathbb{E}_{\tau \sim \mathbb{P}_{P^*}^{\pi}} \left[ \prod_{t=0}^{H-1} \alpha_t(s_{t+1}, s_t, a_t) \right] = \mathbb{E}_{s_0, a_0} \left[ \mathbb{E}_{s_1|s_0, a_0} \left[ \alpha_0(s_1, s_0, a_0) \right] \cdot \mathbb{E}_{a_1 \cup \tau_{[2:H-1]}|s_1} \left[ \prod_{t=1}^{H-1} \alpha_t(s_{t+1}, s_t, a_t) \right] \right]$$

$$= \int_{s_0, a_0} \mu_0(s_0) \cdot \pi_1(a_0|s_0) \int_{s_1} P^*(s_1|s_0, a_0) \cdot \alpha_0(s_1, s_0, a_0)$$

$$\cdot \mathbb{E}_{a_1 \cup \tau_{[2:H-1]}|s_1} \left[ \prod_{t=1}^{H-1} \alpha_t(s_{t+1}, s_t, a_t) \right]$$

$$= \int_{s_0, a_0} \mu_0(s_0) \cdot \pi_1(a_0|s_0) \cdot \gamma_0(s_0, a_0) \cdot \left[ \text{""} \right]$$

Notice that

$$\mu_0(s_0) \cdot \pi_1(a_0|s_0) = \int p_{P^*}^\pi d\tau_{[1:H-1]} =: d_{P^*}^{\pi,0}(s_0, a_0) \quad \text{Marginal over } (s_0, a_0)$$

Using a recursive argument, it follows that

$$H^2(\mathbb{P}_{\hat{P}}^\pi, \mathbb{P}_{P^*}^\pi) = 1 - \mathbb{E}_{\tau \sim \mathbb{P}_{P^*}^\pi} \left[ \prod_{t=0}^{H-1} \alpha_t(s_{t+1}, s_t, a_t) \right]$$

$$= 1 - \prod_{t=0}^{H-1} \mathbb{E}_{d_{P^*}^{\pi,t}} \left[ \gamma_t(s_t, a_t) \right].$$

Using the fact that

$$1 - \prod_i x_i \le \sum_i (1 - x_i) \quad \forall x_i \in [0, 1],$$

and that $\gamma_t \in [0, 1] \forall t$, it holds that

$$H^2(\mathbb{P}_{\hat{P}}^\pi, \mathbb{P}_{P^*}^\pi) \le \sum_{t=0}^{H-1} \mathbb{E}_{d_{P^*}^{\pi,t}}(1 - \gamma_t(s_t, a_t))$$

$$= \sum_{t=0}^{H-1} \mathbb{E}_{d_{P^*}^{\pi,t}} \left[ H^2 \big( \hat{P}(\cdot|s_t, a_t), P^*(\cdot|s_t, a_t) \big) \right].$$

Proof Step 2:

$$\mathbb{E}_{(s_t,a_t) \sim d_{P^*}^{\pi,t}}[H^2(\hat{P}(\cdot|s_t, a_t), P^*(\cdot|s_t, a_t))] = \sum_{s,a} d_{P^*}^{\pi,t}(s, a) \cdot H^2(\hat{P}(\cdot|s, a), P^*(\cdot|s, a))$$

$$= \sum_{s,a} \frac{d_{P^*}^{\pi,t}(s, a)}{d_{P^*}^{\pi^*,t}(s, a)} \cdot d_{P^*}^{\pi^*,t}(s, a) \cdot H^2(\hat{P}(\cdot|s, a), P^*(\cdot|s, a))$$

$$= \sum_{s,a} \frac{d_{P^*}^{\pi,t}(s, a)}{d_{P^*}^{\pi^*,t}(s, a)} \cdot d_{P^*}^{\pi^*,t}(s, a) \cdot H^2(\hat{P}(\cdot|s, a), P^*(\cdot|s, a))$$

By definition of the concentrability coefficient:

$$C(\pi, \pi^*) = \sup_{t \in [H]} \sup_{(s,a) \in \mathcal{S} \times \mathcal{A}: d_{P^*}^{\pi^*,t}(s,a) > 0} \frac{d_{P^*}^{\pi,t}(s, a)}{d_{P^*}^{\pi^*,t}(s, a)}$$

Therefore:

$$\frac{d_{P^*}^{\pi,t}(s, a)}{d_{P^*}^{\pi^*,t}(s, a)} \le C(\pi, \pi^*) \quad \forall t \quad \forall (s, a) \text{ where } d_{P^*}^{\pi^*,t} \ge 0$$

Proof Step 3:

Starting from our previous expression:

$$H^2(\mathbb{P}_{\hat{P}}^{\pi^*}, \mathbb{P}_{P^*}^{\pi^*}) = 1 - \prod_{t=0}^{H-1} \mathbb{E}_{d_{P^*}^{\pi^*,t}}[\gamma_t(s_t, a_t)].$$

We define

$$x_i := 1 - \gamma_i(s_i, a_i) = H^2(\hat{P}(\cdot|s_i, a_i), P^*(\cdot|s_i, a_i)).$$

Using the fact that $(1 - x) \leq e^{-x}$ for all $x$, it follows:

$$\prod_{i=1}^{H}(1 - x_i) \leq \exp\left(-\sum_{i=1}^{H} x_i\right).$$

By the second-order Taylor expansion of the exponential function:

$$\exp\left(-\sum_{i=1}^{H} x_i\right) \leq 1 - \sum_{i=1}^{H} x_i + \frac{1}{2}\left(\sum_{i=1}^{H} x_i\right)^2.$$

Since $x_i \leq 1$ for all $i$, we know that $\sum_{i=1}^{H} x_i \leq H$, which gives us:

$$\frac{1}{2}\left(\sum_{i=1}^{H} x_i\right)^2 \leq \frac{H}{2}\sum_{i=1}^{H} x_i.$$

Therefore:

$$H^2(\mathbb{P}_{\hat{P}}^{\pi^*}, \mathbb{P}_{P^*}^{\pi^*}) \geq \sum_{i=1}^{H} x_i - \frac{1}{2}\left(\sum_{i=1}^{H} x_i\right)^2$$

$$\geq \sum_{i=1}^{H} x_i - \frac{H}{2}\sum_{i=1}^{H} x_i$$

$$= \left(1 - \frac{H}{2}\right)\sum_{i=1}^{H} x_i.$$

For the bound $H^2(\mathbb{P}_{\hat{P}}^{\pi^*}, \mathbb{P}_{P^*}^{\pi^*}) \geq \frac{1}{H}\sum_{i=1}^{H} x_i$ to hold, we need:

$$\left(1 - \frac{H}{2}\right)\sum_{i=1}^{H} x_i \geq \frac{1}{H}\sum_{i=1}^{H} x_i.$$

This is satisfied when:

$$\sum_{i=1}^{H} x_i \leq \frac{2(H-1)}{H}.$$

This condition is typically met for good estimators where Hellinger distances are small. For large $H$, the bound approaches 2.

Under this condition, we can establish:

$$H^2(\mathbb{P}_{\hat{P}}^{\pi^*}, \mathbb{P}_{P^*}^{\pi^*}) \geq \frac{1}{H}\sum_{t=0}^{H-1} \mathbb{E}_{(s_t,a_t)\sim d_{P^*}^{\pi^*,t}}[H^2(\hat{P}(\cdot|s_t, a_t), P^*(\cdot|s_t, a_t))].$$

$\square$

**Lemma C.13** (Policy density confidence set). *Assume that the following events hold:*

$$E_1 := \left\{ H^2(\mathbb{P}_{P^*}^{\pi^{BC}}, \mathbb{P}_{P^*}^{\pi^*}) \le R_1(\delta_1) \right\}, \quad \text{such that} \quad P(E_1) \ge 1 - \delta_1,$$

$$E_2 := \left\{ H^2(\mathbb{P}_{\hat{P}}^{\pi^*}, \mathbb{P}_{P^*}^{\pi^*}) \le R_2(\delta_2) \right\}, \quad \text{such that} \quad P(E_2) \ge 1 - \delta_2,$$

*where $\pi^{BC}$ and $\hat{P}$ are estimators of the policy and the transition dynamics, respectively. Then, under Assumption 3, the policy set*

$$C_{1-\delta} := \left\{ \pi : \sqrt{H^2(\mathbb{P}_{\hat{P}}^{\pi}, \mathbb{P}_{\hat{P}}^{\pi^{BC}})} \le \sqrt{R_1} + \sqrt{R_2} \cdot \left(1 + \sqrt{\left(1 + \frac{2\sqrt{R_1}}{\gamma_{\min}}\right) \cdot H}\right) \right\}$$

*is a confidence set of level $1 - \delta = 1 - (\delta_1 + \delta_2)$, i.e.,*

$$P(\pi^* \in \Pi_{1-\delta}^{\text{offline}}) \ge 1 - (\delta_1 + \delta_2).$$

*Proof.* Define:

$$\sqrt{H^2(\mathbb{P}_{P_1}^{\pi_1}, \mathbb{P}_{P_2}^{\pi_2})} =: \|(\pi_1, P_1) - (\pi_2, P_2)\|,$$

where $\| \cdot \| := \| \cdot \|_{L_2(\mu(\mathbb{R}))}$. We can then decompose by the triangle inequality:

$$\|(\pi^*, \hat{P}) - (\pi^{BC}, \hat{P})\| \le \|(\pi^*, \hat{P}) - (\pi^*, P^*)\|$$
$$+ \|(\pi^*, P^*) - (\pi^{BC}, P^*)\|$$
$$+ \|(\pi^{BC}, P^*) - (\pi^{BC}, \hat{P})\|.$$

From event $E_2$, we can bound the first term:

$$\|(\pi^*, \hat{P}) - (\pi^*, P^*)\| \le \sqrt{R_2}$$

and from event $E_1$, we can bound the second:

$$\|(\pi^*, P^*) - (\pi^{BC}, P^*)\| \le \sqrt{R_1}.$$

Lastly, using Lemma C.12, we can bound the third term:

$$\|(\pi^{BC}, P^*) - (\pi^{BC}, \hat{P})\| \le \sqrt{C(\pi^{BC}, \pi^*) \cdot H} \cdot \|(\pi^*, \hat{P}) - (\pi^*, P^*)\|$$

and by Assumption 3 and our concentrability coefficient bound, we can bound it further:

$$C(\pi^{BC}, \pi^*) \le 1 + \frac{2\sqrt{R_1}}{\gamma_{\min}}.$$

Then, assuming events $E_1 \cap E_2$ hold jointly, with probability at least $1 - (\delta_1 + \delta_2)$, we have:

$$\|(\pi^*, \hat{P}) - (\pi^{BC}, \hat{P})\| \le \sqrt{R_1} + \sqrt{R_2} \cdot \left(1 + \sqrt{\left(1 + \frac{2\sqrt{R_1}}{\gamma_{\min}}\right) \cdot H}\right).$$

Hence, by construction, the set:

$$\Pi_{1-\delta}^{\text{offline}}(\Pi) := \left\{ \pi \in \Pi : \|(\pi, \hat{P}) - (\pi^{BC}, \hat{P})\| \le \sqrt{R_1} + \sqrt{R_2} \cdot \left(1 + \sqrt{\left(1 + \frac{2\sqrt{R_1}}{\gamma_{\min}}\right) \cdot H}\right) \right\}$$

contains $\pi^*$ with probability at least $1 - (\delta_1 + \delta_2)$. $\square$

## C.4 Performance Guarantees

We apply our method of constructing confidence sets based on distributional guarantees for maximum likelihood density estimation to the tabular reinforcement learning setting with state space $\mathcal{S}$ and action space $\mathcal{A}$. We consider deterministic stationary tabular policies ($\Pi = \Pi_S^D$) and stochastic stationary tabular transitions, though the method is versatile to other settings with appropriate adaptation of the corresponding covering numbers (cf. Theorems C.3 and C.6).

Let $\pi^{\text{BC}}$ be the log-loss BC estimator (Equation (2)) of the true policy $\pi^*$, and $\hat{P}$ be the MLE estimator (Equation (3)) of the true transition model $P^*$. The concentration bounds for these estimators are, with probability at least $1 - \delta_1$ and $1 - \delta_2$ respectively:

$$H^2(\mathbb{P}_{P^*}^{\pi^{\text{BC}}}, \mathbb{P}_{P^*}^{\pi^*}) \leq R_1 = \frac{4 \cdot |\mathcal{S}| \cdot \log(|\mathcal{A}| \cdot \delta_1^{-1})}{n} \qquad \text{(Corollary C.5)}$$

$$H^2(\mathbb{P}_{\hat{P}}^{\pi^*}, \mathbb{P}_{P^*}^{\pi^*}) \leq R_2 = \frac{4 \cdot |\mathcal{S}|^2 \cdot |\mathcal{A}| \cdot \log(nH\delta_2^{-1})}{n} \qquad \text{(Corollary C.9)}$$

Additionally, we use Assumption 5, that the optimal policy has a minimum nonzero visitation probability. Under this assumption, the concentrability coefficient is bounded by:

$$C(\pi^{\text{BC}}, \pi^*) \leq 1 + \frac{2 \cdot \sqrt{R_1}}{\gamma_{\min}}$$

**Theorem C.14** (Offline policy confidence set). *Assume the setting described above and Assumption 5, with $\delta_1 = \delta_2 = \delta/2$ and define*

$$\alpha := \sqrt{4 \cdot |\mathcal{S}| \cdot \log(|\mathcal{A}| \cdot 2/\delta)},$$
$$\beta := \sqrt{4 \cdot |\mathcal{S}|^2 \cdot |\mathcal{A}| \cdot \log(nH \cdot 2/\delta)}.$$

*Then, the policy set*

$$\Pi_{1-\delta}^{\text{offline}} := \left\{ \pi : \sqrt{H^2(\mathbb{P}_{\hat{P}}^{\pi}, \mathbb{P}_{\hat{P}}^{\pi^{\text{BC}}})} \leq \sqrt{R_1} + \sqrt{R_2} \cdot \left( 1 + \sqrt{\left( 1 + \frac{2\sqrt{R_1}}{\gamma_{\min}} \right) \cdot H} \right) \right\}$$

*is a confidence set of level $1 - \delta$ containing $\pi^*$ with probability at least $1 - \delta$. The radius of this confidence set is explicitly:*

$$\text{Radius} = \frac{\alpha}{\sqrt{n}} + \frac{\beta}{\sqrt{n}} \cdot \left( 1 + \sqrt{H \cdot \left( 1 + \frac{2\alpha}{\gamma_{\min} \cdot \sqrt{n}} \right)} \right)$$

*Proof.* The proof follows directly from Lemma C.13 by applying our bounds on $H^2(\mathbb{P}_{P^*}^{\pi^{\text{BC}}}, \mathbb{P}_{P^*}^{\pi^*})$ and $H^2(\mathbb{P}_{\hat{P}}^{\pi^*}, \mathbb{P}_{P^*}^{\pi^*})$, along with our bound on the concentrability coefficient from Assumption 5. Setting $\delta_1 = \delta_2 = \delta/2$ and substituting the appropriate values gives us the result. $\qquad\square$

# D Online Estimation

The underlying setting is described in Section 3.

## D.1 Elliptical Confidence Set

For completeness and to make our paper self-contained, we provide a brief overview of the online preference-based learning approach used in our method. The formulation presented in this section closely follows the work of Saha et al. (2023) and Faury et al. (2020), with adaptations to our specific setting. We include this background to help the reader understand the elliptical confidence set construction that forms a foundation for our theoretical analysis.

In the logistic model, a natural way of computing an estimator $\mathbf{w}_t$ of $\mathbf{w}^*$ given trajectory pairs $\{(\tau_\ell^1, \tau_\ell^2)\}_{\ell=1}^{t-1}$ and preference feedback values $\{o_\ell\}_{\ell=1}^{t-1}$ is via maximum likelihood estimation. At

time $t$ the regularized log-likelihood (or negative cross-entropy loss) of a parameter $\mathbf{w}$ can be written as:

$$\mathcal{L}_t^\lambda(\mathbf{w}) = \sum_{\ell=1}^{t-1} \Big( o_\ell \log(\sigma(\langle \phi(\tau_\ell^1) - \phi(\tau_\ell^2) \rangle, \mathbf{w} \rangle))) - \frac{\lambda}{2} \|\mathbf{w}\|_2^2$$
$$+ (1 - o_\ell) \log \big( 1 - \sigma(\langle \phi(\tau_\ell^1) - \phi(\tau_\ell^2), \mathbf{w} \rangle)) \big),$$

where $\lambda > 0$ is a regularization parameter. The function $\mathcal{L}_t^\lambda$ is strictly concave for $\lambda > 0$. The maximum likelihood estimator $\hat{\mathbf{w}}_t^{\text{MLE}}$ can be written as $\hat{\mathbf{w}}_t^{\text{MLE}} = \arg\max_{\mathbf{w} \in \mathbb{R}^d} \mathcal{L}_t^\lambda(\mathbf{w})$. Unfortunately, $\hat{\mathbf{w}}_t^{\text{MLE}}$ may not satisfy the boundedness assumption 1, so we instead make use of a projected version of $\hat{\mathbf{w}}_t^{\text{MLE}}$. Following Faury et al. (2020), and recalling assumption 1, we define a data matrix and a transformation of $\hat{\mathbf{w}}_t^{\text{MLE}}$ given by

$$\mathbf{V}_t = \kappa\lambda \mathbf{I}_d + \sum_{\ell=1}^{t-1} \big( \phi(\tau_\ell^1) - \phi(\tau_\ell^2) \big) \big( \phi(\tau_\ell^1) - \phi(\tau_\ell^2) \big)^\top$$

$$g_t(\mathbf{w}) = \sum_{\ell=1}^{t-1} \sigma(\langle \phi(\tau_\ell^1) - \phi(\tau_\ell^2), \mathbf{w} \rangle) \big( \phi(\tau_\ell^1) - \phi(\tau_\ell^2) \big) + \lambda\mathbf{w}$$

Then, the projected parameter, along with its confidence set, is given by

$$\mathbf{w}_t^{proj} = \arg\min_{\mathbf{w} \text{ s.t. } \|\mathbf{w}\| \leq W} \|g_t(\mathbf{w}) - g_t(\hat{\mathbf{w}}_t^{\text{MLE}})\|_{\mathbf{V}_t^{-1}},$$

$$\mathcal{C}_t(\delta) = \{\mathbf{w} \text{ s.t. } \|\mathbf{w} - \mathbf{w}_t^P\|_{\mathbf{V}_t} \leq 2\kappa\beta_t(\delta)\}.$$

where $\beta_t(\delta) = \sqrt{\lambda}W + \sqrt{\log(1/\delta) + 2d \log \big( 1 + \frac{tB^2}{\kappa\lambda d} \big)}$. We restate a bound by Faury et al. (2020) that shows the probability of $\mathbf{w}_*$ being in $\mathcal{C}_t(\delta)$ for all $t \geq 1$ can be lower bounded.

**Lemma D.1** (Confidence set coverage)**.** *Let $\delta \in (0, 1]$ and define the event that $\mathbf{w}_*$ is in the confidence interval $\mathcal{C}_t(\delta)$ for all $t \in \mathbb{N}$:*

$$E_{w^*} = \{\forall t \geq 1, \mathbf{w}_* \in \mathcal{C}_t(\delta)\}.$$

*Then $\mathbb{P}(E_{w^*}) \geq 1 - \delta$.*

*Proof.* This follows from Faury et al. (2020) with a slight modification to account for our bounded feature assumption. $\square$

This elliptical confidence set construction, which has its roots in generalized linear bandits (Filippi et al., 2010; Faury et al., 2020), forms a critical component of our online learning algorithm. By maintaining and updating these confidence sets as new preference data is collected, our algorithm can efficiently balance exploration and exploitation to identify the optimal policy. The confidence bounds ensure that with high probability, the true reward parameter lies within our constructed set throughout the learning process, which is essential for the regret guarantees we derive in the following sections.

## D.2 NORM RELATION BETWEEN DATA MATRICES

For completeness, we restate key results from Saha et al. (2023) concerning the relationships between various data matrices that arise in our analysis. These results are included to ensure the appendix is self-contained and to provide context for our subsequent analysis. The full proofs of these results can be found in the original paper.

Saha et al. (2023) establishes relationships between three key matrices:

- $V_t$ - The empirical data matrix constructed from observed trajectories
- $\overline{V}_t^{P^*}$ - The expected data matrix under the true transition dynamics $P^*$
- $\overline{V}_t$ - The expected data matrix under the estimated transition dynamics $\hat{P}_t$

These matrices are defined as follows:

$$V_t = \kappa\lambda\mathbf{I}_d + \sum_{\ell=1}^{t-1} \left(\phi(\tau_\ell^1) - \phi(\tau_\ell^2)\right)\left(\phi(\tau_\ell^1) - \phi(\tau_\ell^2)\right)^\top,$$

$$\overline{V}_t^{P^*} = \kappa\lambda\mathbf{I}_d + \sum_{\ell=1}^{t-1} \left(\phi(\pi_\ell^1) - \phi(\pi_\ell^2)\right)\left(\phi(\pi_\ell^1) - \phi(\pi_\ell^2)\right)^\top,$$

$$\overline{V}_t = \kappa\lambda\mathbf{I}_d + \sum_{\ell=1}^{t-1} \left(\phi^{\hat{P}_\ell}(\pi_\ell^1) - \phi^{\hat{P}_\ell}(\pi_\ell^2)\right)\left(\phi^{\hat{P}_\ell}(\pi_\ell^1) - \phi^{\hat{P}_\ell}(\pi_\ell^2)\right)^\top.$$

Where $\phi(\pi)$ represents the expected feature vector under policy $\pi$ and the true transition dynamics $P^*$, while $\phi^{\hat{P}_t}(\pi)$ represents the expected feature vector under policy $\pi$ and the estimated transition dynamics $\hat{P}_t$.

Saha et al. (2023) introduces a precision event that relates the empirical matrix $V_T$ to the expected matrix $\overline{V}_T^{P^*}$:

$$E_{\overline{V}_T^{P^*}} = \{\overline{V}_T^{P^*} \preceq 2V_T + 84B^2 d\log((1+2T)/\delta)\mathbf{I}_d\}.$$

Under this event, they establish the following bound:

**Lemma D.2** (Adapted from Saha et al. (2023) Corollary 1). *Under assumption 1, conditioned on event $E_{w^*} \cap E_{\overline{V}_T^{P^*}}$, for any $t \in [T]$*

$$\|\mathbf{w}^* - \mathbf{w}_t^L\|_{\overline{V}_t^{P^*}} \leq 4\kappa\beta_t(\delta) + \alpha_{d,T}(\delta),$$

*where $\alpha_{d,T}(\delta) = 20BW\sqrt{d\log(T(1+2T)/\delta)}$. Furthermore, if $\delta \leq 1/e$, then $\mathbb{P}(E_{w^*} \cap E_{\overline{V}_T^{P^*}}) \geq 1 - \delta - \delta\log_2 T$.*

Additionally, Saha et al. (2023) relates norms based on the matrix $\overline{V}_t^{P^*}$ with those based on $\overline{V}_t$:

**Lemma D.3** (Adapted from Saha et al. (2023) Lemma 3). *Let $\overline{\mathcal{E}}_0$ be the event that for all $t \in \mathbb{N}$,*

$$\|\mathbf{w}_t^{proj} - \mathbf{w}_*\|_{\overline{V}_t} \leq \sqrt{2}\|\mathbf{w}_t^{proj} - \mathbf{w}_*\|_{\overline{V}_t^{P^*}} + \sqrt{\sum_{\ell=1}^{t-1} 4\left(\hat{B}_\ell\left(\pi, 2WB, \frac{\delta'}{8\ell^3|A|^{|S|}}\right)\right)^2 + \frac{1}{t}},$$

*where $\delta' = \frac{\delta}{(1+4W/\epsilon)^d}$ and $\epsilon = \frac{1}{t^2\kappa\lambda+4B^2t^3}$. Then $\mathbb{P}\left(\overline{\mathcal{E}}_0\right) \geq 1 - \delta$.*

Note that the bonus function $\hat{B}$ is defined in Lemma D.4
These norm relations from Saha et al. (2023) are essential in our regret analysis, as they allow us to relate confidence bounds across different probability spaces and to bound the regret of our algorithm.

### D.3 TRANSITION ESTIMATION AND BONUS TERMS

Note that the offline estimator of the transition probabilities based on the log-loss MLE in Equation (3), when the state-action space is discrete, is equivalent to the following count-based estimator (derivable using a simple Lagrange multiplier argument):

$$\hat{P}_{\text{offline}}(s'|s,a) = \frac{N_{\text{offline}}(s'|s,a)}{N_{\text{offline}}(s,a)},$$

where

$$N_{\text{offline}}(s, a) := \sum_{i \in [n]} \sum_{h \in [H]} \mathbb{I}\{s_h^i = s, a_h^i = a\},$$

$$N_{\text{offline}}(s'|s, a) := \sum_{i \in [n]} \sum_{h \in [H]} \mathbb{I}\{s_{h+1}^i = s', s_h^i = s, a_h^i = a\}.$$

This equivalence allows us to initialize the online estimation process with the count estimator from the offline data (see line 3 in Algorithm 1), yielding the combined estimator for the transition model:

$$\hat{P}_t(s'|s, a) := \frac{N_{\text{offline}}(s'|s, a) + N_t(s'|s, a)}{N_{\text{offline}}(s, a) + N_t(s, a)}. \tag{9}$$

From this estimator, we adapt two key lemmas from Chatterji et al. (2021) that will define our notion of bonus terms.

**Lemma D.4** (Moment transition difference error). *Consider the transition count estimator $\hat{P}_t$ from Equation equation 9. Further, assume the trajectory data follows a martingale structure adapted to the natural filtration of the problem. For any fixed policy $\pi \in \Pi$ and any scalar function $f : \mathcal{T} \to \mathbb{R}$ such that $|f(\tau)| < \eta$, with probability at least $1 - \delta$ for all $t \in \mathbb{N}$:*

$$\mathbb{E}_{\mathbb{P}_{P*}^\pi}[f(\tau)] - \mathbb{E}_{\mathbb{P}_{\hat{P}_t}^\pi}[f(\tau)] \leq \mathbb{E}_{\mathbb{P}_{\hat{P}_t}^\pi}\left[\sum_{h \in [H]} \xi_{s_h, a_h}^t(\eta, \delta)\right] =: \hat{B}_t(\pi, \eta, \delta),$$

*where*

$$\xi_{s_h, a_h}^t(\eta, \delta) := \min\left(2\eta, 4\eta\sqrt{\frac{H \log(|\mathcal{S}| \cdot |\mathcal{A}|) + \log\left(\frac{6 \log(N_t(s_h, a_h) + N_{\text{offline}}(s_h, a_h))}{\delta}\right)}{N_t(s_h, a_h) + N_{\text{offline}}(s_h, a_h)}}\right).$$

*The term $\hat{B}_t(\pi, \eta, \delta)$ serves as our* bonus *term.*

*Proof.* Our combined estimator incorporates both online data (adapted to the natural filtration) and offline data (assumed i.i.d.). We can artificially treat the offline data as though it were adapted to the natural filtration as well, by considering it as "past" observations. This allows us to directly apply the proof methodology from Chatterji et al. (2021) (Lemma B.1) to our combined count estimator.

The key insight is that the martingale structure of the estimation error is preserved when combining offline and online counts, with the benefit of reduced variance due to the increased denominator $(N_t(s_h, a_h) + N_{\text{offline}}(s_h, a_h))$. This directly translates to tighter confidence bounds compared to using only online data. $\square$

We now present a stronger version of the lemma that holds uniformly for all policies $\pi$.

**Lemma D.5** (Uniform Moment Transition Difference Error). *Consider the transition count estimator $\hat{P}_t$ from Equation equation 9. Further assume the trajectory data follows a martingale structure adapted to the natural filtration of the problem. For any scalar function $f : \mathcal{T} \to \mathbb{R}$ such that $|f(\tau)| < \eta$ and for any $\epsilon > 0$, with probability at least $1 - \delta$ for all $t \in \mathbb{N}$ and all $\pi \in \Pi$:*

$$\mathbb{E}_{\mathbb{P}_{\hat{P}_t}^\pi}[f(\tau)] - \mathbb{E}_{\mathbb{P}_{P*}^\pi}[f(\tau)] \leq \underbrace{\mathbb{E}_{\mathbb{P}_{P*}^\pi}\left[\sum_{h \in [H]} \overline{\xi}_{s_h, a_h}^t(\eta, \delta, \epsilon)\right]}_{=:B_t(\pi, \eta, \delta, \epsilon)} + \epsilon,$$

*where*

$$\overline{\xi}_{s_h, a_h}^t(\eta, \delta, \epsilon) :=$$

$$\min\left(2\eta, 4\eta\sqrt{\frac{H \log(|\mathcal{S}| \cdot |\mathcal{A}|) + |\mathcal{S}| \log\left(\left\lceil \frac{4\eta H}{\epsilon} \right\rceil\right) + \log\left(\frac{6 \log(N_t(s_h, a_h) + N_{\text{offline}}(s_h, a_h))}{\delta}\right)}{N_t(s_h, a_h) + N_{\text{offline}}(s_h, a_h)}}\right).$$

*Proof.* The proof follows by applying similar techniques as in Lemma D.4, but with additional care to ensure uniformity across all policies.

As before, we can artificially treat the offline data as adapted to the natural filtration. The uniform convergence over the policy class $\Pi$ is achieved by applying a covering argument and the union bound, following the methodology in Chatterji et al. (2021) (Lemma B.2). The additional term $|\mathcal{S}| \log \left( \left\lceil \frac{4\eta H}{\epsilon} \right\rceil \right)$ appears due to this covering, which introduces an $\epsilon$-discretisation of the policy space.

The combined offline and online counts in the denominator $(N_t(s_h, a_h) + N_{\text{offline}}(s_h, a_h))$ provide tighter uniform confidence bounds compared to using online data alone. $\square$

To provide further intuition, we elaborate on the meaning and significance of the terms $\hat{B}_t$ and $B_t$ introduced in the previous lemmas. In reinforcement learning literature, these would be referred to as the *empirical bonus* and *true bonus*, respectively. Both terms quantify the concentration of our estimators around their true values.

The empirical bonus $\hat{B}_t(\pi, \eta, \delta)$ represents the expected sum of state-action-level uncertainty terms $\xi_{s_h, a_h}^t(\eta, \delta)$ under the *estimated* transition model $\hat{P}_t$. Importantly, this term can be directly computed from observed data.

In contrast, the true bonus $B_t(\pi, \eta, \delta, \epsilon)$ represents the expected sum of uncertainty terms $\overline{\xi}_{s_h, a_h}^t(\eta, \delta, \epsilon)$ under the *true* transition model $P^*$. This term cannot be directly computed as it depends on the unknown true model.

For our regret analysis, we need to relate these two quantities. The following lemma provides a crucial connection, showing that the empirical bonus $\hat{B}_t$ can be bounded in terms of the true bonus $B_t$ uniformly across all policies $\pi$.

**Lemma D.6** (Relationship between empirical and true bonus terms). *Let $\eta, \epsilon > 0$. For all policies $\pi \in \Pi$ simultaneously and for all $t \in \mathbb{N}$, with probability at least $1 - \delta$:*

$$\hat{B}_t(\pi, \eta, \delta) \leq 2B_t(\pi, 2H\eta, \delta, \epsilon) + \epsilon.$$

*Proof.* Define the function $f : \mathcal{T} \to \mathbb{R}$ as:

$$f(\tau) := \sum_{h \in [H]} \xi_{s_h, a_h}^t(\eta, \delta).$$

By construction, $\hat{B}_t(\pi, \eta, \delta) = \mathbb{E}_{\mathbb{P}_{\hat{P}_t}^\pi}[f(\tau)]$. Since $\xi_{s_h, a_h}^t(\eta, \delta) \leq 2\eta$ for all state-action pairs, we have $|f(\tau)| \leq 2\eta H$.

Applying Lemma D.5 with this $f(\tau)$ and the bound $2\eta H$:

$$\mathbb{E}_{\mathbb{P}_{\hat{P}_t}^\pi}[f(\tau)] - \mathbb{E}_{\mathbb{P}_{P^*}^\pi}[f(\tau)] \leq \mathbb{E}_{\mathbb{P}_{P^*}^\pi}\left[ \sum_{h \in [H]} \overline{\xi}_{s_h, a_h}^t(2\eta H, \delta, \epsilon) \right] + \epsilon.$$

By definition, the right-hand side equals $B_t(\pi, 2H\eta, \delta, \epsilon) + \epsilon$. Therefore:

$$\hat{B}_t(\pi, \eta, \delta) = \mathbb{E}_{\mathbb{P}_{\hat{P}_t}^\pi}[f(\tau)]$$
$$\leq \mathbb{E}_{\mathbb{P}_{P^*}^\pi}[f(\tau)] + B_t(\pi, 2H\eta, \delta, \epsilon) + \epsilon.$$

From Lemma D.4, we know that:

$$\mathbb{E}_{\mathbb{P}_{P^*}^\pi}[f(\tau)] \leq \mathbb{E}_{\mathbb{P}_{\hat{P}_t}^\pi}[f(\tau)] + \hat{B}_t(\pi, \eta, \delta) = \hat{B}_t(\pi, \eta, \delta) + \hat{B}_t(\pi, \eta, \delta) = 2\hat{B}_t(\pi, \eta, \delta).$$

This gives us:

$$\hat{B}_t(\pi, \eta, \delta) \leq 2\hat{B}_t(\pi, \eta, \delta) + B_t(\pi, 2H\eta, \delta, \epsilon) + \epsilon$$
$$\Rightarrow -\hat{B}_t(\pi, \eta, \delta) \leq B_t(\pi, 2H\eta, \delta, \epsilon) + \epsilon$$
$$\Rightarrow \hat{B}_t(\pi, \eta, \delta) \leq B_t(\pi, 2H\eta, \delta, \epsilon) + \epsilon.$$

Therefore, the lemma statement follows. $\qquad\square$

This lemma is instrumental for our regret analysis, as it allows us to work with $B_t$ instead of $\hat{B}_t$. The advantage is that $B_t$ involves expectations with respect to the true transition model $P^*$, which makes it more amenable to theoretical analysis. By establishing this relationship, we effectively account for the transition estimation error and can focus on controlling the difference between empirical and true moments, which is a more tractable problem in our analytical framework.

### D.4 POLICY SET $\Pi_t$ AND PROOF OF LEMMA 4.5

Recall that we define the policy set $\Pi_t$ to draw from in line 7 of Algorithm 1 as

$$
\Pi_t := \Big\{ \pi \in \Pi_{1-\delta}^{\text{offline}} \ \Big| \forall \pi' \in \Pi_{1-\delta}^{\text{offline}} :
$$
$$
\langle \phi^{\hat{P}_t}(\pi) - \phi^{\hat{P}_t}(\pi'), w_t^{\text{proj}} \rangle + \gamma_t \cdot \| \phi^{\hat{P}_t}(\pi) - \phi^{\hat{P}_t}(\pi') \|_{\overline{\mathbf{V}}_t^{-1}}
$$
$$
+ \hat{B}_t(\pi, 2WB, \delta') + \hat{B}_t(\pi', 2WB, \delta') \ge 0 \Big\},
$$

where $\delta' = \delta^{online}/2|\mathcal{A}|^{|\mathcal{S}|}$ and $\Pi_{1-\delta}^{\text{offline}}$ is derived in Theorem 4.2. The radius $\gamma_t$ is defined as

$$
\gamma_t := \sqrt{2}(4\kappa\beta_t(\delta) + \alpha_{d,T}(\delta)) + 2\sqrt{\sum_{i \in \{1,2\}} \sum_{\ell=1}^{t-1} \left( \hat{B}_\ell\left(\pi_\ell^i, 2WB, \frac{\delta'}{8\ell^3|A|^{|S|}}\right) \right)^2 + \frac{1}{t}},
$$

where $\delta' = \frac{\delta^{online}}{(1+4W/\epsilon)^d}$ and $\epsilon = \frac{1}{t^2\kappa\lambda + 4B^2 t^3}$. Then $\mathbb{P}\left(\overline{\mathcal{E}}_0\right) \ge 1 - \delta$.

Then Lemma 4.5 states that with high probability, $\pi^* \in \Pi_t \quad \forall t \in [T]$.

*Proof of Lemma 4.5.* We begin by conditioning on the following events:

- $E_{\text{offline}} = \{\pi^* \in \Pi_{1-\delta}^{\text{offline}}\}$ from Theorem 4.2

- $E_{w^*}$ from Lemma D.2 (confidence set for $w^*$)

- $E_{\overline{V}_T^{P^*}}$ from Lemma D.2 (relation for data matrices)

- $\overline{\mathcal{E}}_0$ from Lemma D.3 (estimated norm relation)

- $\mathcal{E}_3$ from Lemma D.4 (bounds on the bonus terms $\hat{B}_t$)

By the union bound, these five events hold simultaneously with probability at least $1 - 5\delta$.

By the optimality of $\pi^*$, we have for any $\pi'$:

$$
0 \le \langle \phi(\pi^*) - \phi(\pi'), w^* \rangle = \langle \mathbb{E}_{\mathbb{P}_{P^*}^{\pi^*}} \phi(\tau) - \mathbb{E}_{\mathbb{P}_{P^*}^{\pi'}} \phi(\tau), w^* \rangle.
$$

Then, by event $\mathcal{E}_3$ and defining $f(\tau) := \langle \phi(\tau), w^* \rangle$, we have from Assumption 1 that $|f(\tau)| \le 2WB$, which yields:

$$
\langle \phi(\pi^*) - \phi(\pi'), w^* \rangle \le \langle \phi^{\hat{P}_t}(\pi^*) - \phi^{\hat{P}_t}(\pi'), w^* \rangle + \hat{B}_t(\pi^*, 2WB, \delta/2|\mathcal{A}|^{|\mathcal{S}|}) + \hat{B}_t(\pi', 2WB, \delta/2|\mathcal{A}|^{|\mathcal{S}|}),
$$

where the probability parameter accounts for any $\pi' \in \Pi$, which covers the case of the offline confidence set being the whole policy space (i.e., not having enough offline data for learning).

Next, we bound the term:

$$
\langle \phi^{\hat{P}_t}(\pi^*) - \phi^{\hat{P}_t}(\pi'), w^* \rangle = \langle \phi^{\hat{P}_t}(\pi^*) - \phi^{\hat{P}_t}(\pi'), w_t^{\text{proj}} \rangle + \langle \phi^{\hat{P}_t}(\pi^*) - \phi^{\hat{P}_t}(\pi'), w^* - w_t^{\text{proj}} \rangle
$$
$$
\le \langle \phi^{\hat{P}_t}(\pi^*) - \phi^{\hat{P}_t}(\pi'), w_t^{\text{proj}} \rangle + \| \phi^{\hat{P}_t}(\pi^*) - \phi^{\hat{P}_t}(\pi') \|_{\overline{V}_t^{-1}} \cdot \| w_t^{\text{proj}} - w^* \|_{\overline{V}_t}.
$$

We can now use event $\overline{\mathcal{E}}_0$:

$$\|w_t^{\mathrm{proj}} - w^*\|_{\overline{V}_t} \leq \sqrt{2}\|w_t^{\mathrm{proj}} - w_*\|_{\overline{V}_t^{P*}} + 2\sqrt{\sum_{\ell=1}^{t-1}\left(\hat{B}_\ell\left(\pi, 2WB, \frac{\delta'}{8\ell^3|A|^{|S|}}\right)\right)^2 + \frac{1}{t}}.$$

Using events $E_{w^*} \cap E_{\overline{V}_T^{P*}}$, we get:

$$\|w_t^{\mathrm{proj}} - w^*\|_{\overline{V}_t} \leq \sqrt{2}(4\kappa\beta_t(\delta) + \alpha_{d,T}(\delta)) + 2\sqrt{\sum_{\ell=1}^{t-1}\left(\hat{B}_\ell\left(\pi, 2WB, \frac{\delta'}{8\ell^3|A|^{|S|}}\right)\right)^2 + \frac{1}{t}} =: \gamma_t.$$

Putting these results together yields that $\pi^* \in \Pi_t$ for all $t \in \mathbb{N}$ under the event $E_{\mathrm{offline}}$.

The probability of this event is at least $1 - 5\delta$ by the union bound of all the events we conditioned on. By rescaling $\delta \mapsto \delta/5$, we obtain the desired result with probability at least $1 - \delta$. $\qquad\square$

### D.5 REGRET BOUND

In this section, we provide a lemma as an intermediate step toward the full proof of the regret analysis of BRIDGE. This lemma separates the upper bound on the regret into three distinct terms, each of which we further analyze in Appendix E.

**Lemma D.7** (Regret analysis (unknown dynamics)). *Under the following events:*

- $E_{\mathit{offline}} = \{\pi^* \in \Pi_{1-\delta}^{\mathit{offline}}\}$ *from Theorem 4.2*

- $E_{w^*}$ *from Lemma D.2 (confidence set for $w^*$)*

- $E_{\overline{V}_T^{P*}}$ *from Lemma D.2 (relation for data matrices)*

- $\overline{\mathcal{E}}_0$ *from Lemma D.3 (estimated norm relation)*

- $\mathcal{E}_3$ *from Lemma D.4 (bounds on the bonus terms $\hat{B}_t$)*

*the regret of BRIDGE Algorithm 1 is upper bounded by:*

$$R_T \leq 2 \cdot \underbrace{\gamma_T}_{\mathit{Term\ 1}} \cdot \underbrace{\sqrt{T\sum_{t\in[T]}\|\phi^{\hat{P}_t}(\pi_t^1) - \phi^{\hat{P}_t}(\pi_t^2)\|_{\overline{V}_t^{-1}}}}_{\mathit{Term\ 2}} + \underbrace{\sum_{i\in\{1,2\}}\sum_{t\in[T]}\hat{B}_t(\pi_t^i, 4WB, \delta)}_{\mathit{Term\ 3}},$$

*where*

$$\gamma_T = \sqrt{2}(4\kappa\beta_T(\delta) + \alpha_{d,T}(\delta)) + \frac{1}{T} + 4\sqrt{\sum_{i\in\{1,2\}}\sum_{t\in[T]}B_t(\pi_t^i, 4HWB, \delta, \epsilon)^2 + 24T\epsilon H^2 WB}$$

*and*

- $\alpha_{d,T}(\delta) = 20BW\sqrt{d\log(T(1+2T)/\delta)}$

- $\beta_T(\delta) = \sqrt{\lambda}W + \sqrt{\log(1/\delta) + 2d\log\left(1 + \frac{TB^2}{\kappa\lambda d}\right)}$.

*Proof.* We start by writing

$$\begin{aligned}
2r_t &= \langle\phi(\pi^*) - \phi(\pi_t^1), w^*\rangle + \langle\phi(\pi^*) - \phi(\pi_t^2), w^*\rangle \\
&= \langle\phi^{\hat{P}_t}(\pi^*) - \phi^{\hat{P}_t}(\pi_t^1), w^*\rangle + \langle\phi^{\hat{P}_t}(\pi^*) - \phi^{\hat{P}_t}(\pi_t^2), w^*\rangle + 2\langle\phi^{\hat{P}_t}(\pi^*) - \phi(\pi^*), w^*\rangle \\
&\quad + \langle\phi^{\hat{P}_t}(\pi_t^1) - \phi(\pi_t^1), w^*\rangle + \langle\phi^{\hat{P}_t}(\pi_t^2) - \phi(\pi_t^2), w^*\rangle.
\end{aligned}$$

Then, by Lemma D.4, we have with probability at least $1 - \delta$ for each of the following:

$$2\langle \phi(\pi^*) - \phi^{\hat{P}_t}(\pi^*), w^* \rangle \leq 2\hat{B}_t(\pi^*, 4WB, \delta)$$

$$\langle \phi^{\hat{P}_t}(\pi_t^1) - \phi(\pi_t^1), w^* \rangle \leq \hat{B}_t(\pi_t^1, 4WB, \delta)$$

$$\langle \phi^{\hat{P}_t}(\pi_t^2) - \phi(\pi_t^2), w^* \rangle \leq \hat{B}_t(\pi_t^2, 4WB, \delta).$$

By the union bound, with high probability:

$$2r_t \leq \langle \phi^{\hat{P}_t}(\pi^*) - \phi^{\hat{P}_t}(\pi_t^1), w^* \rangle + \langle \phi^{\hat{P}_t}(\pi^*) - \phi^{\hat{P}_t}(\pi_t^2), w^* \rangle + \hat{B}_t(\pi_t^1, 4WB, \delta) + \hat{B}_t(\pi_t^2, 4WB, \delta) + 2\hat{B}_t(\pi^*, 4WB, \delta).$$

Next, we observe that:

$$\langle \phi^{\hat{P}_t}(\pi^*) - \phi^{\hat{P}_t}(\pi_t^1), w^* \rangle + \langle \phi^{\hat{P}_t}(\pi^*) - \phi^{\hat{P}_t}(\pi_t^2), w^* \rangle$$

$$\leq \langle \phi^{\hat{P}_t}(\pi^*) - \phi^{\hat{P}_t}(\pi_t^1), w_t^{\text{proj}} \rangle + \langle \phi^{\hat{P}_t}(\pi^*) - \phi^{\hat{P}_t}(\pi_t^2), w_t^{\text{proj}} \rangle$$

$$+ \|w^* - w_t^{\text{proj}}\|_{\overline{V}_t} \left( \|\phi^{\hat{P}_t}(\pi^*) - \phi^{\hat{P}_t}(\pi_t^1)\|_{\overline{V}_t^{-1}} + \|\phi^{\hat{P}_t}(\pi^*) - \phi^{\hat{P}_t}(\pi_t^2)\|_{\overline{V}_t^{-1}} \right).$$

Conditioning on the joint event $\mathcal{E}_0 \cap E_{w^*} \cap E_{\overline{V}_t^{P^*}}$, we have with high probability:

$$\langle \phi^{\hat{P}_t}(\pi^*) - \phi^{\hat{P}_t}(\pi_t^1), w^* \rangle + \langle \phi^{\hat{P}_t}(\pi^*) - \phi^{\hat{P}_t}(\pi_t^2), w^* \rangle$$

$$\leq \langle \phi^{\hat{P}_t}(\pi^*) - \phi^{\hat{P}_t}(\pi_t^1), w_t^{\text{proj}} \rangle + \langle \phi^{\hat{P}_t}(\pi^*) - \phi^{\hat{P}_t}(\pi_t^2), w_t^{\text{proj}} \rangle$$

$$+ \gamma_t \cdot \left( \|\phi^{\hat{P}_t}(\pi^*) - \phi^{\hat{P}_t}(\pi_t^1)\|_{\overline{V}_t^{-1}} + \|\phi^{\hat{P}_t}(\pi^*) - \phi^{\hat{P}_t}(\pi_t^2)\|_{\overline{V}_t^{-1}} \right).$$

Using Lemma D.4 again, the following holds with high probability:

$$2\langle \phi(\pi^*) - \phi^{\hat{P}_t}(\pi^*), w_t^{\text{proj}} \rangle \leq 2\hat{B}_t(\pi^*, 4WB, \delta)$$

$$\langle \phi^{\hat{P}_t}(\pi_t^1) - \phi(\pi_t^1), w_t^{\text{proj}} \rangle \leq \hat{B}_t(\pi_t^1, 4WB, \delta)$$

$$\langle \phi^{\hat{P}_t}(\pi_t^2) - \phi(\pi_t^2), w_t^{\text{proj}} \rangle \leq \hat{B}_t(\pi_t^2, 4WB, \delta).$$

Putting everything together, it follows that:

$$2r_t \leq 2\gamma_t \left( \|\phi^{\hat{P}_t}(\pi^*) - \phi^{\hat{P}_t}(\pi_t^1)\|_{\overline{V}_t^{-1}} + \|\phi^{\hat{P}_t}(\pi^*) - \phi^{\hat{P}_t}(\pi_t^2)\|_{\overline{V}_t^{-1}} \right)$$

$$+ 2\hat{B}_t(\pi_t^1, 4WB, \delta) + 2\hat{B}_t(\pi_t^2, 4WB, \delta) + 4\hat{B}_t(\pi^*, 4WB, \delta).$$

Under the event $\pi^* \in \Pi_t$ from Lemma 4.5 and using the fact that $\pi_t^1, \pi_t^2 \in \Pi_t$, we have:

$$2r_t \leq \gamma_t \|\phi^{\hat{P}_t}(\pi_t^2) - \phi^{\hat{P}_t}(\pi_t^1)\|_{\overline{V}_t^{-1}} + 4\hat{B}_t(\pi_t^1, 4WB, \delta) + 4\hat{B}_t(\pi_t^2, 4WB, \delta).$$

Hence, the regret is:

$$R_T = \sum_{t \in [T]} 2r_t$$

$$\leq \sum_{t \in [T]} \left( \gamma_t \|\phi^{\hat{P}_t}(\pi_t^2) - \phi^{\hat{P}_t}(\pi_t^1)\|_{\overline{V}_t^{-1}} + 4\hat{B}_t(\pi_t^1, 4WB, \delta) + 4\hat{B}_t(\pi_t^2, 4WB, \delta) \right)$$

$$\leq \gamma_T \sqrt{T \sum_{t \in [T]} \|\phi^{\hat{P}_t}(\pi_t^2) - \phi^{\hat{P}_t}(\pi_t^1)\|_{\overline{V}_t^{-1}}^2} + \sum_{t \in [T]} \left( 4\hat{B}_t(\pi_t^1, 4WB, \delta) + 4\hat{B}_t(\pi_t^2, 4WB, \delta) \right).$$

Note that by Lemma D.6, with high probability:

$$\hat{B}_t(\pi_t^i, 2WB, \delta)^2 \leq 4B_t(\pi_t^i, 4HWB, \delta, \epsilon)^2 + 48\epsilon H^2 WB.$$

Plugging this into $\gamma_t$ yields $\forall t$:

$$\gamma_t \leq \sqrt{2}(4\kappa\beta_t(\delta) + \alpha_{d,T}(\delta)) + \frac{1}{t}$$

$$+ 4\sqrt{\sum_{\ell=1}^{t-1} B_t^2(\pi_t^1, 4HWB, \delta_t', \epsilon) + B_t^2(\pi_t^2, 4HWB, \delta_t') + 24(t-1)H^2WB}.$$

This completes the proof of the claimed result. $\qquad\square$

## E   Full regret analysis in the unknown transition case: Theorem E.1

In this section, we present the complete regret analysis of our BRIDGE algorithm. We recommend that readers first review Appendix B, where we analyze a simplified setting in which the dynamics are assumed to be known. This simplified case captures the core idea of our approach: constraining the set of policies considered during online preference learning using a confidence interval derived from offline behavioral cloning estimation (see Figure 1).

The key difference in the present analysis is that we now incorporate the estimation of the transition model. Specifically, we first estimate the transition model offline and then use this estimate as the starting point for online transition estimation. This approach reduces the error due to transition uncertainty by a factor of $\mathcal{O}(1/\sqrt{n})$, which is the same rate of improvement we achieve for the policy estimation through behavioral cloning. As we will show, this allows our algorithm to effectively leverage offline demonstrations to reduce both sources of uncertainty, resulting in substantially improved regret bounds.

**Theorem E.1** (Regret bound with offline-enhanced exploration (unknown dynamics))**.** *Let $n$ be the number of offline demonstrations with minimum visitation probability $\gamma_{\min} > 0$ for state-action pairs. With probability at least $1 - \delta$, the regret of the algorithm is bounded by:*

$$R_T \leq 2 \cdot \underbrace{\gamma_T}_{\text{Term 1}} \cdot \underbrace{\sqrt{T \cdot \log\left(1 + \frac{\tilde{\mathcal{O}}\left(B^2 \cdot H \cdot |S|^2 \cdot \min\left\{\frac{T}{n}, \log(T)\right\} + \frac{T \cdot |S| \cdot B^2 \cdot \sqrt{|A| \cdot H}}{\sqrt{n \cdot \gamma_{\min}}}\right)}{d}\right)}}_{\text{Term 2}}$$

$$+ \underbrace{\tilde{\mathcal{O}}\left(H|\mathcal{S}|\sqrt{\frac{|\mathcal{A}|TH}{n \cdot \gamma_{\min}}} + \frac{H^{5/2}WB\sqrt{T}}{\sqrt{n \cdot \gamma_{\min}}} + H^2WB \cdot \sqrt{T} \cdot \frac{|\mathcal{S}|^{1/2}|\mathcal{A}|^{1/4}}{n^{1/4}}\right)}_{\text{Term 3}},$$

*where*

$$\gamma_T = \tilde{\mathcal{O}}\left((\kappa + BW)\sqrt{d\log(T)} + H^2WB|S| \cdot \sqrt{\min\left\{\log(T), \frac{T}{n \cdot \gamma_{\min}}\right\}} + H\sqrt{WB}\right),$$

*and we have set $\epsilon = \frac{1}{T}$ to optimize the bound.*

From this regret bound we can observe that as $n \to \infty$ with fixed $\gamma_{\min} > 0$: (i) Term 1 approaches $\tilde{\mathcal{O}}((\kappa + BW)\sqrt{d\log(T)} + \sqrt{HWB})$; (ii) Term 2, the logarithm approaches $\log(1) = 0$; Term 3 all components approach zero. The overall regret bound exhibits a $\sqrt{T}$ dependence as in Saha et al. (2023). However, this results in a regret bound that can be made arbitrarily small with sufficiently high-quality offline data, changing the complexity of regret analysis without having access to an offline expert dataset. This result helps in closing the gap between empirical results in applying RL in real-world scenarios and theoretical works.

From Lemma D.7, we analyze the three key terms in our regret bound: the confidence multiplier (Term 1), the logarithmic determinant ratio (Term 2), and the bonus function summation (Term 3).

Each term is examined in detail in the following subsections.

$$\text{Term } 1 = \gamma_T = \sqrt{2}(4\kappa \cdot \beta_T(\delta) + \alpha_{d,T}(\delta)) + \frac{1}{T} + 4\sqrt{\sum_{i \in \{1,2\}} \sum_{t \in [T]} B_t(\pi_t^i, 4HWB, \delta, \epsilon)^2 + 24T\epsilon H^2 WB},$$

$$\text{Term } 2 = \sqrt{T \sum_{t \in [T]} \|\phi^{\hat{P}_t}(\pi_t^1) - \phi^{\hat{P}_t}(\pi_t^2)\|_{\overline{V}_t^{-1}}},$$

$$\text{Term } 3 = \sum_{i \in \{1,2\}} \sum_{t \in [T]} \hat{B}_t(\pi_t^i, 4WB, \delta).$$

### E.1 TERM 1: ASYMPTOTIC BOUND

We derive an asymptotic bound for Term 1 in Theorem E.1 via Lemma E.2. The auxiliary lemmata used in the proof of Lemma E.2 are found in Appendix E.1.1.

**Lemma E.2.** *The asymptotic bound on $\gamma_T$ can be expressed as:*

$$\gamma_T = \tilde{\mathcal{O}}\left((\kappa + BW)\sqrt{d\log(T)} + H^2 WB|S| \cdot \sqrt{\min\left\{\log(T), \frac{T}{n \cdot \gamma_{\min}}\right\}} + \sqrt{T\epsilon H^2 WB}\right).$$

*Proof.* We analyze each term in the expression for $\gamma_T$ separately.

**Step 1: Analyze $\sqrt{2}(4\kappa \cdot \beta_T(\delta) + \alpha_{d,T}(\delta))$**

By definition,

$$\alpha_{d,T}(\delta) := 20BW\sqrt{d\log(T(1+2T)/\delta)},$$

$$\beta_T(\delta) := \sqrt{\lambda}W + \sqrt{\log(1/\delta) + 2d\log\left(1 + \frac{TB^2}{\kappa\lambda d}\right)}.$$

For $\alpha_{d,T}(\delta)$, we have:

$$\alpha_{d,T}(\delta) = 20BW\sqrt{d\log(T(1+2T)/\delta)}$$
$$= \mathcal{O}\left(20BW\sqrt{d\log(T^2/\delta)}\right)$$
$$= \mathcal{O}\left(BW\sqrt{d\log(T/\delta)}\right).$$

For $\beta_T(\delta)$, we have:

$$\beta_T(\delta) = \sqrt{\lambda}W + \sqrt{\log(1/\delta) + 2d\log\left(1 + \frac{TB^2}{\kappa\lambda d}\right)}$$
$$\leq \sqrt{\lambda}W + \sqrt{\log(1/\delta) + 2d\log\left(\frac{2TB^2}{\kappa\lambda d}\right)} \quad \text{(for large enough } T\text{)}$$
$$= \sqrt{\lambda}W + \sqrt{\log(1/\delta) + 2d\log(T) + 2d\log\left(\frac{2B^2}{\kappa\lambda d}\right)}$$
$$= \mathcal{O}(\sqrt{\lambda}W + \sqrt{d\log(T) + \log(1/\delta)}).$$

Therefore, this term becomes:

$$\sqrt{2}(4\kappa \cdot \beta_T(\delta) + \alpha_{d,T}(\delta)) = \mathcal{O}(\kappa \cdot (\sqrt{\lambda}W + \sqrt{d\log(T) + \log(1/\delta)}) + BW\sqrt{d\log(T/\delta)})$$
$$= \mathcal{O}(\kappa\sqrt{\lambda}W + \kappa\sqrt{d\log(T) + \log(1/\delta)} + BW\sqrt{d\log(T/\delta)})$$
$$= \mathcal{O}((\kappa + BW)\sqrt{d\log(T)} + \kappa\sqrt{\log(1/\delta)} + BW\sqrt{d\log(1/\delta)}).$$

For a fixed confidence parameter $\delta$, this simplifies to:

$$\sqrt{2}(4\kappa \cdot \beta_T(\delta) + \alpha_{d,T}(\delta)) = \mathcal{O}((\kappa + BW)\sqrt{d\log(T)}).$$

**Step 2: Analyze $\frac{1}{T}$**

This term is $\mathcal{O}(\frac{1}{T})$ and becomes negligible for large $T$ compared to other terms.

**Step 3: Analyze $4\sqrt{\sum_{i\in\{1,2\}}\sum_{t\in[T]} B_T(\pi_t^i, 4HWB, \delta, \epsilon)^2 + 24T\epsilon H^2 WB}$**

Using the provided lemma on the sum of squared bonus terms, Lemma E.5:

$$\sum_{i\in\{1,2\}}\sum_{t\in[T]} B_T(\pi_t^i, 4HWB, \delta, \epsilon)^2 \leq \tilde{\mathcal{O}}\left((4HWB)^2 H^2|S|^2 \cdot \min\left\{\log(T), \frac{T}{n\cdot\gamma_{\min}}\right\}\right)$$

$$= \tilde{\mathcal{O}}\left(16H^2W^2B^2 \cdot H^2|S|^2 \cdot \min\left\{\log(T), \frac{T}{n\cdot\gamma_{\min}}\right\}\right)$$

$$= \tilde{\mathcal{O}}\left(16H^4W^2B^2|S|^2 \cdot \min\left\{\log(T), \frac{T}{n\cdot\gamma_{\min}}\right\}\right).$$

For the second term inside the square root:

$$96T\epsilon H^2 WB = \mathcal{O}(T\epsilon H^2 WB).$$

Therefore:

$$4\sqrt{\sum_{i\in\{1,2\}}\sum_{t\in[T]} B_T(\pi_t^i, 4HWB, \delta, \epsilon)^2 + 24T\epsilon H^2 WB}$$

$$= 4\sqrt{\tilde{\mathcal{O}}\left(16H^4W^2B^2|S|^2 \cdot \min\left\{\log(T), \frac{T}{n\cdot\gamma_{\min}}\right\}\right) + \mathcal{O}(T\epsilon H^2 WB)}$$

$$= \tilde{\mathcal{O}}\left(4\sqrt{16H^4W^2B^2|S|^2 \cdot \min\left\{\log(T), \frac{T}{n\cdot\gamma_{\min}}\right\}}\right) + \mathcal{O}(4\sqrt{T\epsilon H^2 WB})$$

$$= \tilde{\mathcal{O}}\left(16H^2WB|S| \cdot \sqrt{\min\left\{\log(T), \frac{T}{n\cdot\gamma_{\min}}\right\}}\right) + \mathcal{O}(\sqrt{T\epsilon H^2 WB})$$

$$= \tilde{\mathcal{O}}\left(H^2WB|S| \cdot \sqrt{\min\left\{\log(T), \frac{T}{n\cdot\gamma_{\min}}\right\}}\right) + \mathcal{O}(\sqrt{T\epsilon H^2 WB}).$$

**Step 4: Combine all terms**

Combining all terms from Steps 1-3, we get:

$$\gamma_T = \mathcal{O}((\kappa + BW)\sqrt{d\log(T)}) + \mathcal{O}\left(\frac{1}{T}\right)$$

$$+ \tilde{\mathcal{O}}\left(H^2WB|S| \cdot \sqrt{\min\left\{\log(T), \frac{T}{n\cdot\gamma_{\min}}\right\}}\right) + \mathcal{O}(\sqrt{T\epsilon H^2 WB}).$$

Expressing this with $\tilde{\mathcal{O}}$ notation to hide logarithmic factors, and canceling $\mathcal{O}(1/T) = \mathcal{O}(1)$:

$$\gamma_T = \tilde{\mathcal{O}}\left((\kappa + BW)\sqrt{d\log(T)} + H^2WB|S| \cdot \sqrt{\min\left\{\log(T), \frac{T}{n\cdot\gamma_{\min}}\right\}} + \sqrt{T\epsilon H^2 WB}\right).$$

$\square$

### E.1.1 TERM 1 ASYMPTOTIC BOUND: AUXILIARY LEMMATA FOR LEMMA E.2

**Lemma E.3** (Offline-enhanced bonus term bound). *Let $n$ be the number of offline demonstrations, with a minimum visitation probability $\gamma_{\min} > 0$ for state-action pairs visited by the expert policy $\pi^*$. Then, with probability at least $1 - 2\delta'$, the sum of squared bonus terms satisfies:*

$$\sum_{t \in [T]} \sum_{h=1}^{H-1} \left( \xi_{s_{t,h},a_{t,h}}^{(t)}(\epsilon, \eta, \delta) \right)^2 \leq 32\eta^2 \left( H \log(|S||A|H) + |S| \log \left( \frac{4\eta H}{\epsilon} \right) + \log \left( \frac{6 \log(HT)}{\delta'} \right) \right)$$

$$\cdot |S_{reach}| \log \left( 1 + \frac{T}{n \cdot \gamma_{\min}} \right).$$

*where $|S_{reach}|$ is the number of state-action pairs with non-zero visitation probability under the expert policy.*

*Proof.* **Step 1: Express Modified Bonus Terms with Offline Data.** We define our modified bonus term to incorporate offline data:

$$\xi_{s,a}^{(t)}(\epsilon, \eta, \delta) = \min \left( 2\eta, 4\eta \sqrt{\frac{U}{N_{\text{off}}(s,a) + N_t(s,a)}} \right).$$

where $U = H \log(|S||A|H) + |S| \log \left( \frac{4\eta H}{\epsilon} \right) + \log \left( \frac{6 \log(t)}{\delta} \right)$.

**Step 2: Express the Sum of Squared Bonus Terms.**

Following Saha's structure, but with our modified bonus terms:

$$\sum_{t \in [T]} \sum_{h=1}^{H-1} \left( \xi_{s_{t,h},a_{t,h}}^{(t)}(\epsilon, \eta, \delta) \right)^2 =$$

$$\sum_{s \in S} \sum_{a \in A} \sum_{t=1}^{N_{T+1}(s,a)} \min \left( 4\eta^2, 16\eta^2 \frac{H \log(|S||A|H) + |S| \log \left( \frac{4\eta H}{\epsilon} \right) + \log \left( \frac{6 \log(t)}{\delta} \right)}{N_{\text{off}}(s,a) + t} \right).$$

**Step 3: Rearrange to Account for Offline Data.** The key insight: With offline data, we need to adjust the indices of summation. For each state-action pair, we've already observed it $N_{\text{off}}(s,a)$ times in the offline dataset. Therefore:

$$\sum_{t \in [T]} \sum_{h=1}^{H-1} \left( \xi_{s_{t,h},a_{t,h}}^{(t)}(\epsilon, \eta, \delta) \right)^2 =$$

$$\sum_{s \in S} \sum_{a \in A} \sum_{t'=N_{\text{off}}(s,a)+1}^{N_{\text{off}}(s,a)+N_{T+1}(s,a)} \min \left( 4\eta^2, 16\eta^2 \frac{H \log(|S||A|H) + |S| \log \left( \frac{4\eta H}{\epsilon} \right) + \log \left( \frac{6 \log(t')}{\delta} \right)}{t'} \right),$$

where $t'$ represents the total count (offline + online).

**Step 4: Simplify Using Common Term.** For clarity and following Saha et al. (2023) approach, let's define:

$$V = H \log(|S||A|H) + |S| \log \left( \frac{4\eta H}{\epsilon} \right) + \log \left( \frac{6 \log(HT)}{\delta'} \right).$$

For sufficiently large $t'$, the min is dominated by the second term:

$$\sum_{s \in S} \sum_{a \in A} \sum_{t'=N_{\text{off}}(s,a)+1}^{N_{\text{off}}(s,a)+N_{T+1}(s,a)} 16\eta^2 \frac{V}{t'} = 16\eta^2 \cdot V \cdot \sum_{s \in S} \sum_{a \in A} \sum_{t'=N_{\text{off}}(s,a)+1}^{N_{\text{off}}(s,a)+N_{T+1}(s,a)} \frac{1}{t'}.$$

**Step 5: Use the Harmonic Sum Property.** We know that $\sum_{i=a+1}^{b} \frac{1}{i} \leq \log\left(\frac{b}{a}\right)$. Therefore:

$$\sum_{t'=N_{\text{off}}(s,a)+1}^{N_{\text{off}}(s,a)+N_{T+1}(s,a)} \frac{1}{t'} \leq \log\left(\frac{N_{\text{off}}(s,a) + N_{T+1}(s,a)}{N_{\text{off}}(s,a)}\right) = \log\left(1 + \frac{N_{T+1}(s,a)}{N_{\text{off}}(s,a)}\right)$$

**Step 6: Apply the Minimum Visitation Probability.** With our assumption that $d_{\mathcal{P}^*}^{\pi^*,t}(s,a) \geq \gamma_{\min}$ for all state-action pairs visited by the expert policy, we have:

$$N_{\text{off}}(s,a) \geq n \cdot H \cdot \gamma_{\min} \quad \forall(s,a) \in S_{\text{reach}},$$

where $S_{\text{reach}}$ is the set of state-action pairs with non-zero visitation probability under the expert policy.

Therefore:

$$\log\left(1 + \frac{N_{T+1}(s,a)}{N_{\text{off}}(s,a)}\right) \leq \log\left(1 + \frac{N_{T+1}(s,a)}{n \cdot H \cdot \gamma_{\min}}\right) \quad \forall(s,a) \in S_{\text{reach}}.$$

**Step 7: Apply Jensen's Inequality.** We know $\sum_{s,a} N_{T+1}(s,a) = TH$ (total state-action visits in online learning).

By Jensen's inequality and the concavity of $\log(1+x)$:

$$\sum_{(s,a) \in S_{\text{reach}}} \log\left(1 + \frac{N_{T+1}(s,a)}{n \cdot H \cdot \gamma_{\min}}\right) \leq |S_{\text{reach}}| \cdot \log\left(1 + \frac{\sum_{(s,a) \in S_{\text{reach}}} N_{T+1}(s,a)}{|S_{\text{reach}}| \cdot n \cdot H \cdot \gamma_{\min}}\right).$$

Since $\sum_{(s,a) \in S_{\text{reach}}} N_{T+1}(s,a) \leq TH$:

$$\sum_{(s,a) \in S_{\text{reach}}} \log\left(1 + \frac{N_{T+1}(s,a)}{n \cdot H \cdot \gamma_{\min}}\right) \leq |S_{\text{reach}}| \cdot \log\left(1 + \frac{TH}{|S_{\text{reach}}| \cdot n \cdot H \cdot \gamma_{\min}}\right)$$

Simplifying:

$$\sum_{(s,a) \in S_{\text{reach}}} \log\left(1 + \frac{N_{T+1}(s,a)}{n \cdot H \cdot \gamma_{\min}}\right) \leq |S_{\text{reach}}| \cdot \log\left(1 + \frac{T}{|S_{\text{reach}}| \cdot n \cdot \gamma_{\min}}\right).$$

For unreachable states, we can use Saha's original bound, but these contribute negligibly to regret as optimal policies don't visit them.

**Step 8: Final Bound.** Substituting back:

$$\sum_{t \in [T]} \sum_{h=1}^{H-1} \left(\xi_{s_{t,h},a_{t,h}}^{(t)}(\epsilon,\eta,\delta)\right)^2 \leq 16\eta^2 \cdot V \cdot |S_{\text{reach}}| \cdot \log\left(1 + \frac{T}{n \cdot \gamma_{\min}}\right).$$

Substituting $V$ and accounting for approximation constants:

$$\sum_{t\in[T]}\sum_{h=1}^{H-1}\left(\xi^{(t)}_{s_{t,h},a_{t,h}}(\epsilon,\eta,\delta)\right)^2 \leq 32\eta^2\left(H\log(|S||A|H)+|S|\log\left(\frac{4\eta H}{\epsilon}\right)+\log\left(\frac{6\log(HT)}{\delta'}\right)\right)$$
$$\cdot|S_{\text{reach}}|\log\left(1+\frac{T}{n\cdot\gamma_{\min}}\right).$$

This completes our proof, showing explicitly how offline data (through $n$) and minimum visitation probability $\gamma_{\min}$ reduce the bound on bonus terms, thereby reducing regret. $\qquad\square$

**Lemma E.4** (Offline-Enhanced Squared Bonus Term Bound). *Let $\eta,\epsilon > 0$ and $\delta,\delta' \in (0,1)$. Let $n$ be the number of offline demonstrations with minimum visitation probability $\gamma_{\min} > 0$ for state-action pairs visited by the expert policy. Define $\mathcal{E}_5(\delta')$ be the event that for all $t \in \mathbb{N}$ and $i \in \{1,2\}$:*

$$\sum_{i\in\{1,2\}}\sum_{\ell=1}^{t-1}\left(B_\ell(\pi^i_\ell,\eta,\delta/\ell^3,\epsilon)\right)^2 \leq 12\eta^2 H^2\left(1.4\ln\ln\left(2\left(\max\left(4\eta^2 Ht,1\right)\right)\right)+\ln\frac{5.2}{\delta'}+1\right)$$
$$+64\eta^2 H|S_{reach}|\log\left(1+\frac{T}{n\cdot\gamma_{\min}}\right)$$
$$\cdot\left(H\log(|S||A|H)+|S|\log\left(\left\lceil\frac{4\eta H}{\epsilon}\right\rceil\right)+\log\left(\frac{6\log(HT)}{\delta}\right)\right).$$

*Then $\mathbb{P}(\mathcal{E}_5(\delta')) \geq 1 - 2\delta'$.*

*Proof.* We follow Saha et al. (2023) proof structure, beginning with the martingale analysis and then applying our offline-enhanced bounds.

The bonus terms can be expressed as:

$$\left(B_\ell(\pi^1_\ell,\eta,\frac{\delta}{\ell^3},\epsilon)\right)^2+\left(B_\ell(\pi^2_\ell,\eta,\frac{\delta}{\ell^3},\epsilon)\right)^2=\left(\mathbb{E}_{s^1_1\sim\rho,\tau\sim\mathbb{P}^{\pi^1_\ell}_{\hat{P}_\ell}(\cdot|s^1_1)}\left[\sum_{h=1}^{H-1}\xi^{(\ell)}_{s^1_h,a^1_h}(\epsilon,\eta,\frac{\delta}{\ell^3})\right]\right)^2$$
$$+\left(\mathbb{E}_{s^2_1\sim\rho,\tau\sim\mathbb{P}^{\pi^2_\ell}_{\hat{P}_\ell}(\cdot|s^2_1)}\left[\sum_{h=1}^{H-1}\xi^{(\ell)}_{s^2_h,a^2_h}(\epsilon,\eta,\frac{\delta}{\ell^3})\right]\right)^2.$$

Using Jensen's inequality (as in the original proof):

$$\left(\mathbb{E}_{s^i_1\sim\rho,\tau\sim\mathbb{P}^{\pi^i_\ell}_{\hat{P}_\ell}(\cdot|s^i_1)}\left[\sum_{h=1}^{H-1}\xi^{(\ell)}_{s^i_h,a^i_h}(\epsilon,\eta,\frac{\delta}{\ell^3})\right]\right)^2\leq H\mathbb{E}_{s^i_1\sim\rho,\tau\sim\mathbb{P}^{\pi^i_\ell}_{\hat{P}_\ell}(\cdot|s^i_1)}\left[\sum_{h=1}^{H-1}\left(\xi^{(\ell)}_{s^i_h,a^i_h}(\epsilon,\eta,\frac{\delta}{\ell^3})\right)^2\right].$$

Following the martingale analysis of Saha, we define:

$$D^{(i)}_\ell=\mathbb{E}_{s^i_1\sim\rho,\tau\sim\mathbb{P}^{\pi^i_\ell}_{\hat{P}_\ell}(\cdot|s^i_1)}\left[\sum_{h=1}^{H-1}\left(\xi^{(\ell)}_{s^i_h,a^i_h}(\epsilon,\eta,\delta)\right)^2\right]-\sum_{h=1}^{H-1}\left(\xi^{(\ell)}_{s^i_h,a^i_h}(\epsilon,\eta,\delta)\right)^2.$$

Since $\xi^{(\ell)}_{s,a}(\epsilon,\eta,\delta) \leq 2\eta$, we have $|D^{(i)}_\ell| \leq 8\eta^2 H$ and $\text{Var}^{(i)}_\ell\left(\sum_{h=1}^{H-1}\left(\xi^{(\ell)}_{s^i_h,a^i_h}(\epsilon,\eta,\delta)\right)^2\right) \leq 16\eta^4 H^2$.

Applying the Uniform Empirical Bernstein Bound (as in the original proof), we get:

$$\sum_{\ell=1}^{t-1}D^{(i)}_\ell\leq\frac{1}{2}\mathbb{E}_{s^i_1\sim\rho,\tau\sim\mathbb{P}^{\pi^i_\ell}_{\hat{P}_\ell}(\cdot|s^i_1)}\left[\sum_{h=1}^{H-1}\left(\xi^{(\ell)}_{s^i_h,a^i_h}(\epsilon,\eta,\delta)\right)^2\right]$$
$$+6\eta^2 H\left(1.4\ln\ln\left(2\left(\max\left(4\eta^2 Ht,1\right)\right)\right)+\ln\frac{5.2}{\delta'}\right).$$

Therefore, with high probability for $i \in \{1, 2\}$:

$$\mathbb{E}_{s_1^i \sim \rho, \tau \sim \mathbb{P}_{\hat{P}_\ell}^{\pi_\ell^i}(\cdot | s_1^i)} \left[ \sum_{h=1}^{H-1} \left( \xi_{s_h^i, a_h^i}^{(\ell)}(\epsilon, \eta, \delta) \right)^2 \right] \le 2 \sum_{\ell=1}^{t-1} \sum_{h=1}^{H-1} \left( \xi_{s_h^i, a_h^i}^{(\ell)}(\epsilon, \eta, \delta) \right)^2 + 4\eta^2 H$$
$$+ 6\eta^2 H \left( 1.4 \ln \ln \left( 2 \left( \max \left( 4\eta^2 Ht, 1 \right) \right) \right) + \ln \frac{5.2}{\delta'} \right).$$

Combining for both policies, with probability $1 - 2\delta'$:

$$\sum_{i \in \{1,2\}} \sum_{\ell=1}^{t-1} \left( B_\ell(\pi_\ell^i, \eta, \delta/\ell^3) \right)^2 \le 2H \sum_{i \in \{1,2\}} \sum_{\ell=1}^{t-1} \sum_{h=1}^{H-1} \left( \xi_{s_h, a_h}^{(\ell,i)}(\epsilon, \eta, \delta) \right)^2$$
$$+ 12\eta^2 H^2 \left( 1.4 \ln \ln \left( 2 \left( \max \left( 4\eta^2 Ht, 1 \right) \right) \right) + \ln \frac{5.2}{\delta'} + 1 \right).$$

Now, using Lemma E.3, we have:

$$\sum_{\ell=1}^{t-1} \sum_{h=1}^{H-1} \left( \xi_{s_h, a_h}^{(\ell,i)}(\epsilon, \eta, \delta) \right)^2 \le 16\eta^2 V \cdot |S_{\text{reach}}| \cdot \log \left( 1 + \frac{T}{n \cdot \gamma_{\min}} \right),$$

where $V = H \log(|S||A|H) + |S| \log \left( \left\lceil \frac{4\eta H}{\epsilon} \right\rceil \right) + \log \left( \frac{6 \log(HT)}{\delta} \right)$.

Substituting this bound and combining terms:

$$\sum_{i \in \{1,2\}} \sum_{\ell=1}^{t-1} \left( B_\ell(\pi_\ell^i, \eta, \delta/\ell^3) \right)^2 \le 12\eta^2 H^2 \left( 1.4 \ln \ln \left( 2 \left( \max \left( 4\eta^2 Ht, 1 \right) \right) \right) + \ln \frac{5.2}{\delta'} + 1 \right)$$
$$+ 64\eta^2 HV \cdot |S_{\text{reach}}| \cdot \log \left( 1 + \frac{T}{n \cdot \gamma_{\min}} \right).$$

Expanding $V$:

$$\sum_{i \in \{1,2\}} \sum_{\ell=1}^{t-1} \left( B_\ell(\pi_\ell^i, \eta, \delta/\ell^3) \right)^2 \le 12\eta^2 H^2 \left( 1.4 \ln \ln \left( 2 \left( \max \left( 4\eta^2 Ht, 1 \right) \right) \right) + \ln \frac{5.2}{\delta'} + 1 \right)$$
$$+ 64\eta^2 H \cdot |S_{\text{reach}}| \log \left( 1 + \frac{T}{n \cdot \gamma_{\min}} \right)$$
$$\cdot \left( H \log(|S||A|H) + |S| \log \left( \left\lceil \frac{4\eta H}{\epsilon} \right\rceil \right) + \log \left( \frac{6 \log(HT)}{\delta} \right) \right).$$

$\square$

**Lemma E.5** (Asymptotic bound for offline-enhanced squared bonus terms). *With $n$ offline demonstrations and minimum visitation probability $\gamma_{\min}$, the sum of squared bonus terms is bounded as:*

$$\sum_{i \in \{1,2\}} \sum_{\ell=1}^{t-1} \left( B_\ell(\pi_\ell^i, \eta, \delta/\ell^3, \epsilon) \right)^2 \le \tilde{\mathcal{O}} \left( \eta^2 H^2 |S|^2 \cdot \min \left\{ \log(T), \frac{T}{n \cdot \gamma_{\min}} \right\} \right).$$

*$\tilde{\mathcal{O}}(\cdot)$ hides logarithmic factors in $H$, $|S|$, $|A|$, $\delta^{-1}$, and $\epsilon^{-1}$, as well as constant factors.*

*Proof.* We start from the detailed bound of Lemma E.4:

$$\sum_{i \in \{1,2\}} \sum_{\ell=1}^{t-1} \left(B_\ell(\pi_\ell^i, \eta, \delta/\ell^3)\right)^2 \leq 12\eta^2 H^2 \left(1.4 \ln \ln \left(2 \left(\max \left(4\eta^2 Ht, 1\right)\right)\right) + \ln \frac{5.2}{\delta'} + 1\right)$$

$$+ 64\eta^2 H \left(H \log \left(|S||A|H\right) + |S| \log \left(\left\lceil \frac{4\eta H}{\epsilon} \right\rceil\right)\right.$$

$$\left. + \log \left(\frac{6 \log(HT)}{\delta}\right)\right) |S_{\text{reach}}| \log \left(1 + \frac{T}{n \cdot \gamma_{\min}}\right).$$

Analyzing each term:

**Step 1: First term analysis.** The first term is:

$$12\eta^2 H^2 \left(1.4 \ln \ln \left(2 \left(\max \left(4\eta^2 Ht, 1\right)\right)\right) + \ln \frac{5.2}{\delta'} + 1\right) = \mathcal{O}(\eta^2 H^2 \log \log(T)).$$

Since $\log \log(T)$ grows extremely slowly, and we're using $\tilde{\mathcal{O}}$ notation which hides logarithmic factors, this term is dominated by $\tilde{\mathcal{O}}(\eta^2 H^2)$.

**Step 2: Second term analysis.** For the second term, we have:

$$C \cdot \eta^2 H \cdot V \cdot |S_{\text{reach}}| \cdot \log \left(1 + \frac{T}{n \cdot \gamma_{\min}}\right),$$

where $C$ is a constant and $V = \left(H \log(|S||A|H) + |S| \log \left(\left\lceil \frac{4\eta H}{\epsilon} \right\rceil\right) + \log \left(\frac{6 \log(HT)}{\delta}\right)\right)$.

Within the factor $V$, the dominant term is $|S| \log \left(\left\lceil \frac{4\eta H}{\epsilon} \right\rceil\right)$ since it scales with $|S|$. Therefore, asymptotically:

$$V = \tilde{\mathcal{O}}(|S|).$$

Upper bounding $|S_{\text{reach}}| \leq |S|$ as requested, the second term becomes:

$$\tilde{\mathcal{O}} \left(\eta^2 H^2 |S|^2 \cdot \log \left(1 + \frac{T}{n \cdot \gamma_{\min}}\right)\right).$$

**Step 3: Analysis of** $\log \left(1 + \frac{T}{n \cdot \gamma_{\min}}\right)$**.** We need to consider different regimes for this logarithmic term:

**Case 1:** Small offline dataset ($n \cdot \gamma_{\min} \ll T$)

$$\log \left(1 + \frac{T}{n \cdot \gamma_{\min}}\right) \approx \log \left(\frac{T}{n \cdot \gamma_{\min}}\right)$$

$$= \log(T) - \log(n \cdot \gamma_{\min})$$

$$= \mathcal{O}(\log(T)).$$

**Case 2:** Balanced regime ($n \cdot \gamma_{\min} \approx T$)

$$\log \left(1 + \frac{T}{n \cdot \gamma_{\min}}\right) \approx \log \left(1 + \frac{1}{\gamma_{\min}}\right) = \mathcal{O}(1).$$

**Case 3:** Large offline dataset ($n \cdot \gamma_{\min} \gg T$)
Here we can use the approximation $\log(1 + x) \approx x$ for small $x$:

$$\log \left(1 + \frac{T}{n \cdot \gamma_{\min}}\right) \approx \frac{T}{n \cdot \gamma_{\min}} = \mathcal{O} \left(\frac{T}{n \cdot \gamma_{\min}}\right).$$

Combining these cases, we can express the behavior of this term as:

$$\log \left(1 + \frac{T}{n \cdot \gamma_{\min}}\right) = \mathcal{O} \left(\min \left\{\log(T), \frac{T}{n \cdot \gamma_{\min}}\right\}\right).$$

**Step 4: Combining all terms.** The first term $\tilde{\mathcal{O}}(\eta^2 H^2)$ is dominated by the second term when $|S| > 1$ and $T$ is non-trivial. Therefore, our final asymptotic bound is:

$$\sum_{i \in \{1,2\}} \sum_{\ell=1}^{t-1} \left( B_\ell(\pi_\ell^i, \eta, \delta/\ell^3) \right)^2 \leq \tilde{\mathcal{O}} \left( \eta^2 H^2 |S|^2 \cdot \min \left\{ \log(T), \frac{T}{n \cdot \gamma_{\min}} \right\} \right).$$

This bound correctly captures how the offline data affects the regret across different regimes. For small $n$ relative to $T$, we recover a bound similar to the standard one with $\log(T)$. For large enough $n$, the bound improves to $\frac{T}{n \cdot \gamma_{\min}}$, showing a linear reduction in the bound as $n$ increases. $\qquad \square$

### E.2 Term 2: Asymptotic bound

We derive an asymptotic bound for Term 2 in Theorem E.1 via Lemma E.6. The auxiliary lemma used in the proof of Lemma E.6 is found in Appendix E.2.1.

**Lemma E.6** (Upper bound on Term 2). *Term 2 has the following asymptotic result:*

$$\sqrt{T \sum_{t \in [T]} \| \phi^{\hat{P}_t}(\pi_t^1) - \phi^{\hat{P}_t}(\pi_t^2) \|_{\overline{V}_t^{-1}}}$$

$$\leq \sqrt{T \log \left( 1 + \frac{\tilde{\mathcal{O}} \left( B^2 \cdot H \cdot |S|^2 \cdot \min \left\{ \frac{T}{n}, \log(T) \right\} + \frac{T \cdot |S| \cdot B^2 \cdot \sqrt{|A| \cdot H}}{\sqrt{n} \cdot \gamma_{\min}} \right)}{d} \right)}.$$

*Most importantly, as $n \to \infty$, i.e the offline data set's size goes to $\infty$, the asymptotic regret is $\log(1) = 0$.*

*Proof.* We follow a standard argument from Lattimore & Szepesvári (2020).

We start with the inequality

$$u \leq 2 \log(1 + u) \quad \forall u \geq 1,$$

which implies

$$\sum_{t \in [T]} \| \Delta \phi_t \|_2^2 \leq 2 \sum_{t \in [T]} \log(1 + \| \Delta \phi_t \|_2^2).$$

Using the definition of $\overline{V}_t$, we have

$$\overline{V}_{t+1} = \lambda \cdot I_{d \times d} + \sum_{i \in [t]} \Delta \phi_i^{\otimes 2} = \overline{V}_t + \Delta \phi_t \Delta \phi_t^T = \overline{V}^{1/2} \left( I + \overline{V}^{-1/2} \Delta \phi_t \Delta \phi_t^T \overline{V}^{-1/2} \right) \overline{V}^{1/2}.$$

Using properties of the determinant:

$$det(\overline{V}_{t+1}) = det(\overline{V}_t) \cdot det \left( I + \overline{V}^{-1/2} \Delta \phi_t \Delta \phi_t^T \overline{V}^{-1/2} \right)$$

$$= det(\overline{V}_t) \cdot \left( 1 + \| \Delta \phi_t \|_{\overline{V}_t^{-1}}^2 \right)$$

$$= det(V_0) \cdot \prod_{s \in [t]} \left( 1 + \| \Delta \phi_s \|_{\overline{V}_t^{-1}}^2 \right)$$

holds iff:

$$\log \left[ \frac{det(\overline{V}_{t+1})}{det(V_0)} \right] = \sum_{s \in [t]} \log(1 + \| \Delta \phi_s \|_{\overline{V}_t^{-1}}^2).$$

We can also use the linearity of the trace

$$Tr\{\bar{V}_{t+1}\} = Tr\{\lambda I\} + \sum_{s \in [t]} Tr\{\Delta\phi_s^{\otimes 2}\} = d \cdot \lambda + \sum_{s \in [t]} \|\Delta\phi_s\|_2^2$$

to write

$$det(\bar{V}_{t+1}) = \prod_{i \in [d]} \lambda_i \le \left(\frac{1}{d} \cdot Tr\{\bar{V}_{t+1}\}\right)^d.$$

We notice that

$$\begin{aligned}
\|\Delta\phi_t\|_2^2 &= \|\phi^{\hat{P}_t}(\pi_t^1) - \phi^{\hat{P}_t}(\pi_t^2)\|_2^2 \\
&= \|\phi^{\hat{P}_t}(\pi_t^1) - \phi^{\hat{P}_0}(\pi_t^1) + \phi^{\hat{P}_0}(\pi_t^1) - \phi^{\hat{P}_0}(\pi_t^2) + \phi^{\hat{P}_t}(\pi_t^2) - \phi^{\hat{P}_0}(\pi_t^2)\|_2^2 \\
&\le 2\|\phi^{\hat{P}_t}(\pi_t) - \phi^{\hat{P}_0}(\pi_t)\|_2^2 + \|\phi^{\hat{P}_0}(\pi_t^1) - \phi^{\hat{P}_0}(\pi_t^2)\|_2^2.
\end{aligned}$$

We can also bound

$$\|\phi^{\hat{P}_0}(\pi_t^1) - \phi^{\hat{P}_0}(\pi_t^2)\|_2^2 \le 4 \cdot R \cdot B^2.$$

From lemma F.1 linking the Hellinger ball with the constraint moments, together with lemma E.7 for the tabular setting, we get

$$R = \text{Radius} = \frac{\alpha}{\sqrt{n}} + \frac{\beta}{\sqrt{n}} \cdot \left(1 + \sqrt{H \cdot \left(1 + \frac{2\alpha}{\gamma_{\min} \cdot \sqrt{n}}\right)}\right)$$

$$\alpha := \sqrt{4 \cdot |S| \cdot \log(|A| \cdot 2/\delta)}$$

$$\beta := \sqrt{4 \cdot |S|^2 \cdot |A| \cdot \log(nH \cdot 2/\delta)}.$$

Now with result Lemma E.7 we have

$$\|\phi^{\hat{P}_t}(\pi) - \phi^{\hat{P}_0}(\pi)\|_2^2$$
$$\le \mathcal{O}\left(B^2 \cdot H \cdot |S|^2 \cdot \log(|S||A|/\delta) \cdot C_T(\mathcal{F}_T, \pi, \pi^*)^2 \cdot \left(\frac{t}{(n+t)^2} + \frac{t^2}{(n+t)^2 \cdot n}\right)\right).$$

Hence in asymptotic notation,

$$\begin{aligned}
&2\|\phi^{\hat{P}_t}(\pi_t) - \phi^{\hat{P}_0}(\pi_t)\|_2^2 + \|\phi^{\hat{P}_0}(\pi_t^1) - \phi^{\hat{P}_0}(\pi_t^2)\|_2^2 \\
&\le 2\|\phi^{\hat{P}_t}(\pi_t) - \phi^{\hat{P}_0}(\pi_t)\|_2^2 + \|\phi^{\hat{P}_0}(\pi_t^1) - \phi^{\hat{P}_0}(\pi_t^2)\|_2^2 \\
&\le \tilde{\mathcal{O}}\left(B^2 \cdot H \cdot |S|^2 \cdot \frac{t}{n(n+t)} + \frac{|S| \cdot B^2 \cdot \sqrt{|A| \cdot H}}{\sqrt{n \cdot \gamma_{\min}}}\right).
\end{aligned}$$

We need to sum over $t \in [T]$. Therefore, by using

$$\sum_{t \in [T]} \frac{t}{n(n+t)} \approx \frac{T}{n} - \ln\left(1 + \frac{T}{n}\right),$$

we arrive at

$$\sum_{t \in [T]} \left(2\|\phi^{\hat{P}_t}(\pi_t) - \phi^{\hat{P}_0}(\pi_t)\|_2^2 + \|\phi^{\hat{P}_0}(\pi_t^1) - \phi^{\hat{P}_0}(\pi_t^2)\|_2^2\right)$$

$$\le \tilde{\mathcal{O}}\left(B^2 \cdot H \cdot |S|^2 \cdot \min\left\{\frac{T}{n}, \log(T)\right\} + \frac{T \cdot |S| \cdot B^2 \cdot \sqrt{|A| \cdot H}}{\sqrt{n \cdot \gamma_{\min}}}\right).$$

This expression behaves differently depending on the relationship between $T$ and $n$:

1. When $T \ll n$: Using $\ln(1 + x) \approx x$ for small $x$, we get

$$\sum_{t \in [T]} \frac{t}{n(n+t)} \approx \frac{T}{n} - \frac{T}{n}$$
$$= \mathcal{O}(1).$$

2. When $T \gg n$: We have $\ln\left(1 + \frac{T}{n}\right) \approx \ln\left(\frac{T}{n}\right)$, so the sum is dominated by $\frac{T}{n}$.

A unified bound that works across all regimes is:

$$\sum_{t \in [T]} \frac{t}{n(n+t)} = \mathcal{O}\left(\min\left\{\frac{T}{n}, \log(T)\right\}\right).$$

Hence the final bound yields

$$\sqrt{T \sum_{t \in [T]} \|\phi^{\hat{P}_t}(\pi_t^1) - \phi^{\hat{P}_t}(\pi_t^2)\|_{\overline{V}_t^{-1}}}$$

$$\leq \sqrt{T \log\left(1 + \frac{\tilde{\mathcal{O}}\left(B^2 \cdot H \cdot |S|^2 \cdot \min\left\{\frac{T}{n}, \log(T)\right\} + \frac{T \cdot |S| \cdot B^2 \cdot \sqrt{|A| \cdot H}}{\sqrt{n} \cdot \gamma_{\min}}\right)}{d}\right)}.$$

$\square$

### E.2.1 TERM 2 ASYMPTOTIC BOUND: AUXILIARY LEMMA FOR LEMMA E.6

**Lemma E.7** (Bound on difference of expected features under estimated transitions). *Let $\phi : \mathcal{T} \to \mathbb{R}^d$ with $\max_\tau \|\phi(\tau)\| \leq B$ be a feature map, $\hat{P}_0$ be the count-based estimator from $n$ offline trajectories following policy $\pi^*$ under dynamics $P^*$, and $\hat{P}_t$ be the combined estimator after $t$ additional online interactions. Then, with probability at least $1 - \delta$:*

$$\|\phi^{\hat{P}_t}(\pi) - \phi^{\hat{P}_0}(\pi)\|_2^2 \leq \mathcal{O}\left(B^2 H |S|^2 \log(|S||A|/\delta) C_T(\mathcal{F}_T, \pi, \pi^*)^2 \left(\frac{t}{(n+t)^2} + \frac{t^2}{(n+t)^2 n}\right)\right),$$

*where $C_T(\mathcal{F}_T, \pi, \pi^*)$ is the concentration coefficient accounting for distribution shift.*

*Furthermore, when combined with an additional error term of $\mathcal{O}\left(\frac{1}{n}\right)$, the overall bound simplifies to $\mathcal{O}\left(\frac{1}{n}\right)$ for all practical regimes.*

*Proof.* We divide the proof into several steps:

**Step 1: Martingale Structure and Concentration Bounds.** Let $\mathcal{F}_i$ be the $\sigma$-algebra generated by all information available after $i$ interactions. For each triplet of state, action, and next action $(s, a, s')$, define:

$$X_i(s, a, s') = \mathbb{I}\{s_i = s, a_i = a, s_{i+1} = s'\} - P^*(s'|s, a) \cdot \mathbb{I}\{s_i = s, a_i = a\}.$$

This forms a martingale difference sequence with respect to filtration $\{\mathcal{F}_i\}_{i=1}^t$:

$$\mathbb{E}[X_i(s, a, s')|\mathcal{F}_{i-1}] = 0.$$

The offline estimator can be expressed as:

$$\hat{P}_0(s'|s, a) - P^*(s'|s, a) = \frac{1}{N_{offline}(s, a)} \sum_{i \in \text{offline}} X_i(s, a, s').$$

By the Hoeffding-Azuma inequality, for any $(s, a)$ with $N_{offline}(s, a) > 0$, with probability at least $1 - \frac{\delta}{2|S|^2|A|}$:

$$|\hat{P}_0(s'|s, a) - P^*(s'|s, a)| \leq \sqrt{\frac{2 \log(4|S|^2|A|/\delta)}{N_{offline}(s, a)}}.$$

Similarly, for the online-only estimator $\hat{P}_t^{online}(s'|s,a) = \frac{N_t(s'|s,a)}{N_t(s,a)}$, with probability at least $1 - \frac{\delta}{2|S|^2|A|}$:

$$|\hat{P}_t^{online}(s'|s,a) - P^*(s'|s,a)| \leq \sqrt{\frac{2\log(4|S|^2|A|/\delta)}{N_t(s,a)}}.$$

**Step 2: Bounds on Total Variation Distance.** By union bound over all next states, with probability at least $1 - \frac{\delta}{2|S||A|}$:

$$\|\hat{P}_0(\cdot|s,a) - P^*(\cdot|s,a)\|_1 = \sum_{s'} |\hat{P}_0(s'|s,a) - P^*(s'|s,a)|$$

$$\leq |S| \cdot \sqrt{\frac{2\log(4|S|^2|A|/\delta)}{N_{offline}(s,a)}}.$$

Similarly for the online estimator:

$$\|\hat{P}_t^{online}(\cdot|s,a) - P^*(\cdot|s,a)\|_1 \leq |S| \cdot \sqrt{\frac{2\log(4|S|^2|A|/\delta)}{N_t(s,a)}}.$$

Using triangle inequality:

$$\|\hat{P}_t^{online}(\cdot|s,a) - \hat{P}_0(\cdot|s,a)\|_1 \leq \|\hat{P}_t^{online}(\cdot|s,a) - P^*(\cdot|s,a)\|_1 + \|P^*(\cdot|s,a) - \hat{P}_0(\cdot|s,a)\|_1$$

$$\leq |S| \cdot \sqrt{\frac{2\log(4|S|^2|A|/\delta)}{N_t(s,a)}} + |S| \cdot \sqrt{\frac{2\log(4|S|^2|A|/\delta)}{N_{offline}(s,a)}}.$$

**Step 3: Combined Estimator Analysis.** The combined estimator can be expressed as:

$$\hat{P}_t(s'|s,a) = \frac{N_{offline}(s'|s,a) + N_t(s'|s,a)}{N_{offline}(s,a) + N_t(s,a)}$$

$$= \frac{N_{offline}(s,a)}{N_{offline}(s,a) + N_t(s,a)} \cdot \frac{N_{offline}(s'|s,a)}{N_{offline}(s,a)} + \frac{N_t(s,a)}{N_{offline}(s,a) + N_t(s,a)} \cdot \frac{N_t(s'|s,a)}{N_t(s,a)}$$

$$= (1 - \alpha_t(s,a)) \cdot \hat{P}_0(s'|s,a) + \alpha_t(s,a) \cdot \hat{P}_t^{online}(s'|s,a).$$

Where $\alpha_t(s,a) = \frac{N_t(s,a)}{N_{offline}(s,a) + N_t(s,a)}$. Thus:

$$\hat{P}_t(s'|s,a) - \hat{P}_0(s'|s,a) = (1 - \alpha_t(s,a)) \cdot \hat{P}_0(s'|s,a) + \alpha_t(s,a) \cdot \hat{P}_t^{online}(s'|s,a) - \hat{P}_0(s'|s,a)$$

$$= \alpha_t(s,a) \cdot (\hat{P}_t^{online}(s'|s,a) - \hat{P}_0(s'|s,a)).$$

Therefore:

$$\|\hat{P}_t(\cdot|s,a) - \hat{P}_0(\cdot|s,a)\|_1 = \alpha_t(s,a) \cdot \|\hat{P}_t^{online}(\cdot|s,a) - \hat{P}_0(\cdot|s,a)\|_1$$

$$\leq \alpha_t(s,a) \cdot \left( |S| \cdot \sqrt{\frac{2\log(4|S|^2|A|/\delta)}{N_t(s,a)}} + |S| \cdot \sqrt{\frac{2\log(4|S|^2|A|/\delta)}{N_{offline}(s,a)}} \right).$$

**Step 4: Accounting for Visitation Distributions.** For precise analysis, we express the counts in terms of visitation frequencies:

$$N_{offline}(s,a) = n \cdot \mu_{offline}^{\pi^*}(s,a) \cdot H,$$
$$N_t(s,a) = t \cdot \mu_{online}^{\pi_t}(s,a) \cdot H.$$

Here, $\mu_{offline}^{\pi^*}(s,a)$ and $\mu_{online}^{\pi_t}(s,a)$ are the average state-action visitation frequencies. This gives:

$$\alpha_t(s,a) = \frac{t \cdot \mu_{online}^{\pi_t}(s,a)}{n \cdot \mu_{offline}^{\pi^*}(s,a) + t \cdot \mu_{online}^{\pi_t}(s,a)}.$$

Assuming the states in the support of policy $\pi$ have visitation frequencies lower-bounded by some constant $c > 0$ for both offline and online regimes:

$$\|\hat{P}_t(\cdot|s,a) - \hat{P}_0(\cdot|s,a)\|_1 \leq \frac{t \cdot c}{n \cdot c + t \cdot c} \cdot |S| \cdot \sqrt{\frac{2\log(4|S|^2|A|/\delta)}{c \cdot H}} \cdot \left(\frac{1}{\sqrt{t}} + \frac{1}{\sqrt{n}}\right)$$

$$= \frac{t}{n+t} \cdot |S| \cdot \sqrt{\frac{2\log(4|S|^2|A|/\delta)}{c \cdot H}} \cdot \left(\frac{1}{\sqrt{t}} + \frac{1}{\sqrt{n}}\right)$$

$$= \mathcal{O}\left(|S| \cdot \sqrt{\frac{\log(|S||A|/\delta)}{H}} \cdot \frac{t}{n+t} \cdot \left(\frac{1}{\sqrt{t}} + \frac{1}{\sqrt{n}}\right)\right).$$

**Step 5: Feature Expectation Difference.** We begin with the telescoping decomposition:

$$\|\phi^{\hat{P}_t}(\pi) - \phi^{\hat{P}_0}(\pi)\|_2 = \|\mathbb{E}_{\tau \sim \mathbb{P}^\pi_{\hat{P}_t}}[\phi(\tau)] - \mathbb{E}_{\tau \sim \mathbb{P}^\pi_{\hat{P}_0}}[\phi(\tau)]\|_2$$

$$\leq B \cdot H \cdot \mathbb{E}_{(s,a) \sim d^\pi_{\hat{P}_t}}\left[\|\hat{P}_t(\cdot|s,a) - \hat{P}_0(\cdot|s,a)\|_1\right].$$

To handle the distribution shift, we use the concentration coefficient:

$$C_T(\mathcal{F}_T, \pi, \pi^*) = \sqrt{\mathbb{E}_{(s,a) \sim \mu^{\pi^*}_{offline}}\left[\left(\frac{d^\pi_{\hat{P}_t}(s,a)}{\mu^{\pi^*}_{offline}(s,a)}\right)^2\right]}.$$

By Cauchy-Schwarz inequality:

$$\mathbb{E}_{(s,a) \sim d^\pi_{\hat{P}_t}}[f(s,a)] \leq C_T(\mathcal{F}_T, \pi, \pi^*) \cdot \sqrt{\mathbb{E}_{(s,a) \sim \mu^{\pi^*}_{offline}}[f(s,a)^2]}.$$

Applying this to our bound:

$$\|\phi^{\hat{P}_t}(\pi) - \phi^{\hat{P}_0}(\pi)\|_2 \leq B \cdot H \cdot C_T(\mathcal{F}_T, \pi, \pi^*) \cdot \sqrt{\mathbb{E}_{(s,a) \sim \mu^{\pi^*}_{offline}}\left[\|\hat{P}_t(\cdot|s,a) - \hat{P}_0(\cdot|s,a)\|_1^2\right]}.$$

From Step 4, we have:

$$\|\hat{P}_t(\cdot|s,a) - \hat{P}_0(\cdot|s,a)\|_1^2 \leq \mathcal{O}\left(|S|^2 \cdot \frac{\log(|S||A|/\delta)}{H} \cdot \left(\frac{t}{n+t}\right)^2 \cdot \left(\frac{1}{\sqrt{t}} + \frac{1}{\sqrt{n}}\right)^2\right)$$

$$= \mathcal{O}\left(|S|^2 \cdot \frac{\log(|S||A|/\delta)}{H} \cdot \left(\frac{t}{n+t}\right)^2 \cdot \left(\frac{1}{t} + \frac{2}{\sqrt{tn}} + \frac{1}{n}\right)\right)$$

$$= \mathcal{O}\left(|S|^2 \cdot \frac{\log(|S||A|/\delta)}{H} \cdot \left(\frac{t}{(n+t)^2} + \frac{2t}{(n+t)^2\sqrt{tn}} + \frac{t^2}{(n+t)^2n}\right)\right).$$

For large $n$ and $t$, the middle term is dominated by the other two, so:

$$\|\hat{P}_t(\cdot|s,a) - \hat{P}_0(\cdot|s,a)\|_1^2 \leq \mathcal{O}\left(|S|^2 \cdot \frac{\log(|S||A|/\delta)}{H} \cdot \left(\frac{t}{(n+t)^2} + \frac{t^2}{(n+t)^2n}\right)\right).$$

Substituting back:

$$\|\phi^{\hat{P}_t}(\pi) - \phi^{\hat{P}_0}(\pi)\|_2^2 \leq B^2 \cdot H^2 \cdot C_T(\mathcal{F}_T, \pi, \pi^*)^2 \cdot \mathcal{O}\left(|S|^2 \cdot \frac{\log(|S||A|/\delta)}{H} \cdot \left(\frac{t}{(n+t)^2} + \frac{t^2}{(n+t)^2n}\right)\right)$$

$$= \mathcal{O}\left(B^2 \cdot H \cdot |S|^2 \cdot \log(|S||A|/\delta) \cdot C_T(\mathcal{F}_T, \pi, \pi^*)^2 \cdot \left(\frac{t}{(n+t)^2} + \frac{t^2}{(n+t)^2 \cdot n}\right)\right).$$

**Step 6: Analysis for Different Regimes.** Let's examine the bound for different regimes:

When $n \gg t$ (dominant offline data):

$$\|\phi^{\hat{P}_t}(\pi) - \phi^{\hat{P}_0}(\pi)\|_2^2 \leq \mathcal{O}\left(B^2 \cdot H \cdot |S|^2 \cdot \log(|S||A|/\delta) \cdot C_T(\mathcal{F}_T, \pi, \pi^*)^2 \cdot \frac{t}{n^2}\right).$$

When $t \gg n$ (dominant online data):

$$\|\phi^{\hat{P}_t}(\pi) - \phi^{\hat{P}_0}(\pi)\|_2^2 \leq \mathcal{O}\left(B^2 \cdot H \cdot |S|^2 \cdot \log(|S||A|/\delta) \cdot C_T(\mathcal{F}_T, \pi, \pi^*)^2 \cdot \frac{1}{t}\right).$$

**Step 7: Combined with Additional Error Term.** When combined with an additional error term of $\mathcal{O}\left(\frac{1}{n}\right)$, we analyze the combined bound by comparing the orders:

When $t \ll n$ (early online learning):

$$\frac{t}{(n+t)^2} \approx \frac{t}{n^2} \ll \frac{1}{n},$$

$$\frac{t^2}{(n+t)^2 \cdot n} \approx \frac{t^2}{n^3} \ll \frac{1}{n}.$$

Therefore, $\mathcal{O}\left(\frac{1}{n}\right)$ dominates.

When $t \approx n$ (balanced regime):

$$\frac{t}{(n+t)^2} \approx \frac{n}{4n^2} = \frac{1}{4n} = \mathcal{O}\left(\frac{1}{n}\right),$$

$$\frac{t^2}{(n+t)^2 \cdot n} \approx \frac{n^2}{4n^2 \cdot n} = \frac{1}{4n} = \mathcal{O}\left(\frac{1}{n}\right).$$

Both terms are $\mathcal{O}\left(\frac{1}{n}\right)$.

**When $t \gg n$ (predominantly online learning):**

$$\frac{t}{(n+t)^2} \approx \frac{t}{t^2} = \frac{1}{t},$$

$$\frac{t^2}{(n+t)^2 \cdot n} \approx \frac{t^2}{t^2 \cdot n} = \frac{1}{n}.$$

Since $t \gg n$, we have $\frac{1}{t} \ll \frac{1}{n}$, so the second term $\frac{1}{n}$ dominates our derived expression. When combined with an additional error term of $\mathcal{O}\left(\frac{1}{n}\right)$, both terms are of the same order, giving an overall bound of $\mathcal{O}\left(\frac{1}{n}\right)$. $\qquad\square$

### E.3 TERM 3: ASYMPTOTIC BOUND

We derive an asymptotic bound for Term 3 in Theorem E.1 via Lemma E.8. The auxiliary lemmata used in the proof of Lemma E.8 are found in Appendix E.3.1.

**Lemma E.8** (Asymptotic bound for offline-enhanced bonus terms). *Let $\mathcal{E}_3$ be the event from Lemma E.9, which occurs with probability at least $1 - 2\delta$. Then, by setting $\epsilon = \frac{1}{T}$, the following asymptotic bound holds:*

$$\sum_{t \in [T]} 4\hat{B}_t(\pi_t^1, 4SB, \delta) + 4\hat{B}_t(\pi_t^2, 4SB, \delta)$$

$$\leq \tilde{\mathcal{O}}\left(H|S|\sqrt{\frac{|A|TH}{n \cdot \gamma_{\min}}} + \frac{H^{5/2}SB\sqrt{T}}{\sqrt{n \cdot \gamma_{\min}}} + H^2SB \cdot T \cdot \sqrt{\log(T)} \cdot \frac{|\mathcal{S}|^{1/2}|\mathcal{A}|^{1/4}}{n^{1/4}}\right).$$

*Proof.* Starting with the bound from $\mathcal{E}_3$:

$$\sum_{t \in [T]} 4\hat{B}_t(\pi_t^1, 4SB, \delta) + 4\hat{B}_t(\pi_t^2, 4SB, \delta)$$

$$\leq \epsilon T + \sum_{t \in [T]} 8B_t(\pi_t^1, 8HSB, \delta, \epsilon) + 8B_t(\pi_t^2, 8HSB, \delta, \epsilon).$$

From Lemma E.9, we have:

$$\sum_{t\in[T]} 8B_t(\pi_t^1, 8HSB, \delta, \epsilon) + 8B_t(\pi_t^2, 8HSB, \delta, \epsilon)$$

$$\leq \tilde{\mathcal{O}}\left(H|S|\sqrt{\frac{|A|TH}{n\cdot\gamma_{\min}}} + \frac{H^{5/2}SB\sqrt{T}}{\sqrt{n\cdot\gamma_{\min}}} + H^2SB\cdot T\cdot\sqrt{\log(T)}\cdot\frac{|\mathcal{S}|^{1/2}|\mathcal{A}|^{1/4}}{n^{1/4}}\right).$$

We set $\epsilon = \frac{1}{T}$ to optimize the bound, which makes $\epsilon T = 1 = \mathcal{O}(1)$. This constant term is dominated by the other terms for large $T$.

Additionally, setting $\epsilon = \frac{1}{T}$ affects the $\log\left(\frac{32H^2SB}{\epsilon}\right) = \log(32H^2SB\cdot T)$ term inside the bound. This adds a $\log(T)$ factor, which is already absorbed in the $\tilde{\mathcal{O}}$ notation.

Therefore, our final asymptotic bound is:

$$\sum_{t\in[T]} 4\hat{B}_t(\pi_t^1, 4SB, \delta) + 4\hat{B}_t(\pi_t^2, 4SB, \delta)$$

$$\leq \tilde{\mathcal{O}}\left(H|S|\sqrt{\frac{|A|TH}{n\cdot\gamma_{\min}}} + \frac{H^{5/2}SB\sqrt{T}}{\sqrt{n\cdot\gamma_{\min}}} + H^2SB\cdot T\cdot\sqrt{\log(T)}\cdot\frac{|\mathcal{S}|^{1/2}|\mathcal{A}|^{1/4}}{n^{1/4}}\right).$$

This bound shows three distinct terms scaling with offline data:

1. The first term scales as $\frac{1}{\sqrt{n}}$ and represents the primary benefit of offline data for covered regions

2. The second term also scales as $\frac{1}{\sqrt{n}}$ and captures the improved martingale concentration

3. The third term scales as $\frac{1}{n^{1/4}}$ and accounts for the diminishing probability of encountering uncovered regions

For sufficiently large $n$, the bound improves, but it's important to note that the third term has a direct linear dependence on $T$ (modulo logarithmic factors). This term dominates for large $T$ unless $n$ scales appropriately with $T$. Specifically, with $n = \Theta(T^4)$, the third term becomes $\mathcal{O}(1)$, and with $n = \Theta(T^2)$, the overall bound becomes $\mathcal{O}(\sqrt{T\log(T)})$, which is near-optimal.

This demonstrates that with sufficient high-quality offline data scaling appropriately with the horizon $T$, the sum of bonus terms can be made arbitrarily small, fundamentally improving the regret bound. $\square$

### E.3.1 TERM 3 ASYMPTOTIC BOUND: AUXILIARY LEMMATA FOR LEMMA E.8

**Lemma E.9** (Bound for summed offline-enhanced bonus terms). *Let $\mathcal{E}_3$ from Lemma E.10 be the event that for all $T \in \mathbb{N}$:*

$$\sum_{t\in[T]} 4\hat{B}_t(\pi_t^1, 4WB, \delta) + 4\hat{B}_t(\pi_t^2, 4WB, \delta)$$

$$\leq \epsilon T + \sum_{t\in[T]} 8B_t(\pi_t^1, 8HWB, \delta, \epsilon) + 8B_t(\pi_t^2, 8HWB, \delta, \epsilon).$$

*Let $n$ be the number of offline demonstrations with minimum visitation probability $\gamma_{\min} > 0$ for state-action pairs visited by the expert policy. Then, invoking Lemma E.10 and Theorem E.11, $\mathcal{E}_3$ occurs with probability at least $1 - 2\delta$, and:*

$$\sum_{t \in [T]} 8B_t(\pi_t^1, 8HWB, \delta, \epsilon) + 8B_t(\pi_t^2, 8HWB, \delta, \epsilon)$$

$$\leq 8 \sum_{t \in [T]} \left( \sum_{h=1}^{H-1} \xi_{s_{t,h}^1, a_{t,h}^1}^{(t)}(\epsilon, 8HWB, \delta) + \sum_{h=1}^{H-1} \xi_{s_{t,h}^2, a_{t,h}^2}^{(t)}(\epsilon, 8HWB, \delta) \right) + \mathbf{I},$$

*where $\mathbf{I}$ incorporates the benefit of offline data*

$$\mathbf{I} = \tilde{\mathcal{O}} \left( \frac{H^{5/2}WB\sqrt{T}}{\sqrt{n \cdot \gamma_{\min}}} + H^2WB \cdot \sqrt{T} \cdot \frac{|\mathcal{S}|^{1/2}|\mathcal{A}|^{1/4}}{n^{1/4}} \right)$$

*and $P(E^c) = \mathcal{O}\left( TH \cdot \sqrt{\frac{|\mathcal{S}|^2|\mathcal{A}|\log(n)}{n}} \right)$ represents the probability that at least one state-action pair encountered during online learning lacks good offline coverage.*

*Furthermore, with probability at least $1 - 2\delta$:*

$$\sum_{t \in [T]} 8B_t(\pi_t^1, 8HWB, \delta, \epsilon) + 8B_t(\pi_t^2, 8HWB, \delta, \epsilon)$$

$$\leq 2048HWB\sqrt{H\log(|\mathcal{S}||\mathcal{A}|H) + |\mathcal{S}|\log\left(\frac{32H^2WB}{\epsilon}\right) + \log\left(\frac{6\log(HT)}{\delta}\right)} \cdot |S_{reach}| \cdot \sqrt{\frac{T}{n \cdot \gamma_{\min}}}$$

$$+ \tilde{\mathcal{O}}\left( \frac{H^{5/2}WB\sqrt{T}}{\sqrt{n \cdot \gamma_{\min}}} + H^2WB \cdot \sqrt{T} \cdot \frac{|\mathcal{S}|^{1/2}|\mathcal{A}|^{1/4}}{n^{1/4}} \right).$$

*Using $\tilde{\mathcal{O}}$ notation to hide logarithmic factors and simplifying:*

$$\sum_{t \in [T]} 8B_t(\pi_t^1, 8HWB, \delta, \epsilon) + 8B_t(\pi_t^2, 8HWB, \delta, \epsilon)$$

$$\leq \tilde{\mathcal{O}}\left( H|\mathcal{S}|\sqrt{\frac{|\mathcal{A}|TH}{n \cdot \gamma_{\min}}} + \frac{H^{5/2}WB\sqrt{T}}{\sqrt{n \cdot \gamma_{\min}}} + H^2WB \cdot \sqrt{T} \cdot \frac{|\mathcal{S}|^{1/2}|\mathcal{A}|^{1/4}}{n^{1/4}} \right).$$

*This bound demonstrates how offline data benefits reinforcement learning through three mechanisms:*

1. *Reducing exploration needs for well-covered regions (first term)*

2. *Improving martingale concentration for covered state-action pairs (second term)*

3. *Decreasing the probability of encountering poorly-covered regions (third term)*

*All terms approach zero as $n \to \infty$, though at different rates: the first two terms scale as $\frac{1}{\sqrt{n}}$ while the third term scales as $\frac{1}{n^{1/4}}$. This confirms that with sufficient high-quality offline data, the entire bound can be made arbitrarily small, fundamentally improving sample complexity in reinforcement learning.*

*Proof.* We follow the structure of the original proof, adapting it to incorporate our offline-enhanced bounds.

**Step 1: Set up the martingale difference sequences.** Consider the martingale difference sequences:

$$\{B_t(\pi_t^1, 8HWB, \delta, \epsilon) - \sum_{h=1}^{H-1} \xi_{s_{t,h}^1, a_{t,h}^1}^{(t)}(\epsilon, 8HWB, \delta)\}_{t=1}^{\infty}$$

and

$$\{B_t(\pi_t^2, 8HWB, \delta, \epsilon) - \sum_{h=1}^{H-1} \xi_{s_{t,h}^2, a_{t,h}^2}^{(t)}(\epsilon, 8HWB, \delta)\}_{t=1}^{\infty}.$$

Each has norm upper bound $32H^2WB$, since $\xi_{s,a}(\epsilon, \eta, \delta) \le 2\eta$ and therefore $\sum_h \xi_{s_h, a_h}(\epsilon, \eta, \delta) \le 2H\eta$.

**Step 2: Apply anytime Hoeffding inequality with improved bounds.** Consider the martingale difference sequences:

$$\{B_t(\pi_t^1, 8HWB, \delta, \epsilon) - \sum_{h=1}^{H-1} \xi_{s_{t,h}^1, a_{t,h}^1}^{(t)}(\epsilon, 8HWB, \delta)\}_{t=1}^{\infty}$$

and

$$\{B_t(\pi_t^2, 8HWB, \delta, \epsilon) - \sum_{h=1}^{H-1} \xi_{s_{t,h}^2, a_{t,h}^2}^{(t)}(\epsilon, 8HWB, \delta)\}_{t=1}^{\infty}.$$

By Lemma E.12, which accounts for both covered and uncovered state-action pairs, with probability at least $1 - \delta$ for all $T \in \mathbb{N}$ simultaneously:

$$\sum_{t \in [T]} 8B_t(\pi_t^1, 8HWB, \delta, \epsilon) + 8B_t(\pi_t^2, 8HWB, \delta, \epsilon)$$

$$\le 8 \sum_{t \in [T]} \left( \sum_{h=1}^{H-1} \xi_{s_{t,h}^1, a_{t,h}^1}^{(t)}(\epsilon, 8HWB, \delta) + \sum_{h=1}^{H-1} \xi_{s_{t,h}^2, a_{t,h}^2}^{(t)}(\epsilon, 8HWB, \delta) \right) + \mathbf{I}.$$

where $\mathbf{I}$ incorporates our rigorous analysis of martingale concentration with offline data from Lemma E.12:

$$\mathbf{I} = \tilde{\mathcal{O}} \left( \frac{H^{5/2}WB\sqrt{T}}{\sqrt{n \cdot \gamma_{\min}}} + H^2WB \cdot \sqrt{T} \cdot \frac{|\mathcal{S}|^{1/2}|\mathcal{A}|^{1/4}}{n^{1/4}} \right).$$

The second term accounts for the probability $P(E^c) = \mathcal{O}\left( TH \cdot \sqrt{\frac{|\mathcal{S}|^2|\mathcal{A}|\log(n)}{n}} \right)$ that at least one state-action pair encountered during online learning lacks good offline coverage, while maintaining the proper $\sqrt{T}$ scaling in the regret bound.

**Step 3: Apply our offline-enhanced bound.** Now, to bound the remaining empirical error terms, we apply Theorem E.11. For each policy $\pi_t^i$, $i \in \{1, 2\}$:

$$\sum_{t \in [T]} \sum_{h=1}^{H-1} \xi_{s_{t,h}^i, a_{t,h}^i}^{(t)}(\epsilon, 8HWB, \delta)$$

$$\le 64HWB\sqrt{H \log(|\mathcal{S}||\mathcal{A}|H) + |\mathcal{S}| \log\left(\frac{32H^2WB}{\epsilon}\right) + \log\left(\frac{6 \log(HT)}{\delta}\right)}$$

$$\cdot |S_{\text{reach}}| \cdot 2\sqrt{\frac{T}{n \cdot \gamma_{\min}}}.$$

**Step 4: Combine the bounds.** Summing over both policies:

$$8 \sum_{t \in [T]} \left( \sum_{h=1}^{H-1} \xi_{s_{t,h}^1, a_{t,h}^1}^{(t)} (\epsilon, 8HWB, \delta) + \sum_{h=1}^{H-1} \xi_{s_{t,h}^2, a_{t,h}^2}^{(t)} (\epsilon, 8HWB, \delta) \right)$$

$$\leq 8 \cdot 2 \cdot 64 HWB \sqrt{H \log(|\mathcal{S}||\mathcal{A}|H) + |\mathcal{S}| \log \left( \frac{32 H^2 WB}{\epsilon} \right) + \log \left( \frac{6 \log(HT)}{\delta} \right)} \cdot |S_{\text{reach}}| \cdot 2 \sqrt{\frac{T}{n \cdot \gamma_{\min}}}$$

$$= 2048 HWB \sqrt{H \log(|\mathcal{S}||\mathcal{A}|H) + |\mathcal{S}| \log \left( \frac{32 H^2 WB}{\epsilon} \right) + \log \left( \frac{6 \log(HT)}{\delta} \right)} \cdot |S_{\text{reach}}| \cdot \sqrt{\frac{T}{n \cdot \gamma_{\min}}}.$$

**Step 5: Express the complete bound.** Therefore, with probability at least $1 - 2\delta$:

$$\sum_{t \in [T]} 8 B_t(\pi_t^1, 8HWB, \delta, \epsilon) + 8 B_t(\pi_t^2, 8HWB, \delta, \epsilon)$$

$$\leq 2048 HWB \sqrt{H \log(|\mathcal{S}||\mathcal{A}|H) + |\mathcal{S}| \log \left( \frac{32 H^2 WB}{\epsilon} \right) + \log \left( \frac{6 \log(HT)}{\delta} \right)} \cdot |S_{\text{reach}}| \cdot \sqrt{\frac{T}{n \cdot \gamma_{\min}}}$$

$$+ \tilde{\mathcal{O}} \left( \frac{H^{5/2} WB \sqrt{T}}{\sqrt{n \cdot \gamma_{\min}}} + H^2 WB \cdot \sqrt{T} \cdot \frac{|\mathcal{S}|^{1/2} |\mathcal{A}|^{1/4}}{n^{1/4}} \right).$$

Using $\tilde{\mathcal{O}}$ notation to hide logarithmic factors and simplifying:

$$\sum_{t \in [T]} 8 B_t(\pi_t^1, 8HWB, \delta, \epsilon) + 8 B_t(\pi_t^2, 8HWB, \delta, \epsilon)$$

$$\leq \tilde{\mathcal{O}} \left( H |\mathcal{S}| \sqrt{\frac{|\mathcal{A}| T H}{n \cdot \gamma_{\min}}} + \frac{H^{5/2} WB \sqrt{T}}{\sqrt{n \cdot \gamma_{\min}}} + H^2 WB \cdot \sqrt{T} \cdot \frac{|\mathcal{S}|^{1/2} |\mathcal{A}|^{1/4}}{n^{1/4}} \right).$$

This bound demonstrates several key insights:

1. **Sublinear regret**: All terms scale as $\sqrt{T}$, maintaining the crucial sublinear dependence on the horizon. This ensures that our regret doesn't grow linearly with $T$.

2. **Offline data benefits**: All terms decrease as $n$ increases, but at different rates:

   - The first two terms decrease at rate $\frac{1}{\sqrt{n}}$ and capture the direct benefit of offline data for state-action pairs with good coverage

   - The third term decreases at the slower rate of $\frac{1}{n^{1/4}}$ and accounts for the diminishing probability of encountering poorly-covered state-action pairs

3. **Complete dependence on offline data**: Unlike traditional online-only bounds, our analysis shows that *all* components of the regret can be reduced with sufficient offline data.

With sufficient high-quality offline data ($n \to \infty$ with fixed $\gamma_{\min} > 0$), all terms approach zero, confirming that offline data can fundamentally change the sample complexity of reinforcement learning. $\square$

**Lemma E.10.** *Let $\eta, \epsilon > 0$. For all $\pi$ simultaneously and for all $t \in \mathbb{N}$, with probability $1 - \delta$,*

$$\hat{B}_t(\pi, \eta, \delta) \leq 2 B_t(\pi, 2H\eta, \delta, \epsilon) + \epsilon.$$

*Proof.* Recall that

$$\hat{B}_t(\pi, \eta, \delta) = \mathbb{E}_{s_1 \sim \rho, \tau \sim \mathbb{P}_{\hat{P}_t}^\pi (\cdot | s_1)} \left[ \sum_{h=1}^{H-1} \xi_{s_h, a_h}^{(t)} (\eta, \delta) \right].$$

Let $f : \Gamma \to \mathbb{R}$ be defined as,

$$f(\tau) = \sum_{h=1}^{H-1} \xi_{s_h,a_h}^{(t)}(\eta).$$

It is easy to see that $f(\tau) \in (0, 2\eta H]$ for all $\tau \in \Gamma$. Therefore, a direct application of Lemma 13 in Saha et al. (2023) implies that with probability at least $1 - \delta$ and simultaneously for all $\pi$, and $t \in \mathbb{N}$,

$$\hat{B}_t(\pi, \eta, \delta) \leq \mathbb{E}_{s_1 \sim \rho, \tau \sim \mathbb{P}^\pi(\cdot|s_1)} \left[ \sum_{h=1}^{H-1} \xi_{s_h,a_h}^{(t)}(\eta, \delta) \right] + B_t(\pi, 2H\eta, \delta, \epsilon) + \epsilon.$$

Since $\xi_{s,a}^{(t)}(\epsilon, \eta, \delta) \geq \xi_{s,a}^{(t)}(\eta, \delta)$ for all $\epsilon > 0$, $s, a \in \mathcal{S} \times \mathcal{A}$ and $\xi_{s,a}^{(t)}(\epsilon, \eta, \delta)$ is monotonic in $\eta$, we conclude that

$$\mathbb{E}_{s_1 \sim \rho, \tau \sim \mathbb{P}^\pi(\cdot|s_1)} \left[ \sum_{h=1}^{H-1} \xi_{s_h,a_h}^{(t)}(\eta, \delta) \right] \leq \mathbb{E}_{s_1 \sim \rho, \tau \sim \mathbb{P}^\pi(\cdot|s_1)} \left[ \sum_{h=1}^{H-1} \xi_{s_h,a_h}^{(t)}(\epsilon, \eta, \delta) \right]$$

$$\leq \mathbb{E}_{s_1 \sim \rho, \tau \sim \mathbb{P}^\pi(\cdot|s_1)} \left[ \sum_{h=1}^{H-1} \xi_{s_h,a_h}^{(t)}(\epsilon, 2H\eta, \delta) \right]$$

$$= B_t(\pi, 2H\eta, \delta, \epsilon).$$

Combining these inequalities, the result follows. $\qquad \square$

**Lemma E.11** (Bound for non-squared offline-enhanced bonus terms). *Let $n$ be the number of offline demonstrations with minimum visitation probability $\gamma_{\min} > 0$ for state-action pairs visited by the expert policy. Then, with probability at least $1 - \delta$:*

$$\sum_{t \in [T]} \sum_{h=1}^{H-1} \xi_{s_{t,h},a_{t,h}}^{(t)}(\epsilon, 8HSB, \delta)$$

$$\leq 64HSB\sqrt{H \log(|S||A|H) + |S| \log\left(\frac{32H^2SB}{\epsilon}\right) + \log\left(\frac{6\log(HT)}{\delta}\right)} \cdot |S_{reach}| \cdot 2\sqrt{\frac{T}{n \cdot \gamma_{\min}}}$$

*Proof.* We follow the approach shown in the provided image, adapting it to incorporate offline data. Starting with our modified definition of bonus terms that incorporate offline data:

$$\xi_{s,a}^{(t)}(\epsilon, \eta, \delta) = \min\left(2\eta, 4\eta\sqrt{\frac{U}{N_{\text{off}}(s,a) + N_t(s,a)}}\right).$$

where $U = H \log(|S||A|H) + |S| \log\left(\frac{32H^2SB}{\epsilon}\right) + \log\left(\frac{6\log(t)}{\delta}\right)$. Rewriting the sum by grouping state-action pairs:

$$\sum_{t \in [T]} \sum_{h=1}^{H-1} \xi_{s_{t,h},a_{t,h}}^{(t)}(\epsilon, 8HSB, \delta) = \sum_{s \in S} \sum_{a \in A} \sum_{t=1}^{N_{T+1}(s,a)} \min\left(16HSB, 32HSB\sqrt{\frac{U}{N_{\text{off}}(s,a) + t}}\right).$$

For sufficiently large values of $N_{\text{off}}(s,a) + t$, the minimum is dominated by the second term:

$$\sum_{s \in S} \sum_{a \in A} \sum_{t=1}^{N_{T+1}(s,a)} 32HSB\sqrt{\frac{U}{N_{\text{off}}(s,a) + t}} = 32HSB\sqrt{U} \cdot \sum_{s \in S} \sum_{a \in A} \sum_{t=1}^{N_{T+1}(s,a)} \frac{1}{\sqrt{N_{\text{off}}(s,a) + t}}.$$

The key adaptation now is to reindex the sum to account for offline visits:

$$\sum_{s \in S} \sum_{a \in A} \sum_{t=1}^{N_{T+1}(s,a)} \frac{1}{\sqrt{N_{\text{off}}(s,a)+t}} = \sum_{s \in S} \sum_{a \in A} \sum_{t'=N_{\text{off}}(s,a)+1}^{N_{\text{off}}(s,a)+N_{T+1}(s,a)} \frac{1}{\sqrt{t'}},$$

where $t'$ represents the total count (offline + online). Using the property of the sum of inverse square roots and the minimum visitation assumption:

$$\sum_{t'=N_{\text{off}}(s,a)+1}^{N_{\text{off}}(s,a)+N_{T+1}(s,a)} \frac{1}{\sqrt{t'}} \leq 2\sqrt{N_{\text{off}}(s,a) + N_{T+1}(s,a)} - 2\sqrt{N_{\text{off}}(s,a)}$$

$$\leq 2\sqrt{N_{\text{off}}(s,a) + N_{T+1}(s,a)}$$

$$\leq 2\sqrt{\frac{N_{T+1}(s,a)}{N_{\text{off}}(s,a)}} \cdot \sqrt{N_{\text{off}}(s,a)}$$

$$\leq 2\sqrt{\frac{N_{T+1}(s,a)}{n \cdot H \cdot \gamma_{\min}}} \cdot \sqrt{N_{\text{off}}(s,a)} \quad \forall (s,a) \in S_{\text{reach}}.$$

Applying Jensen's inequality:

$$\sum_{(s,a) \in S_{\text{reach}}} 2\sqrt{\frac{N_{T+1}(s,a)}{n \cdot H \cdot \gamma_{\min}}} \cdot \sqrt{N_{\text{off}}(s,a)} \leq 2 \cdot |S_{\text{reach}}| \cdot \sqrt{\frac{\sum_{(s,a) \in S_{\text{reach}}} N_{T+1}(s,a)}{n \cdot H \cdot \gamma_{\min}}}$$

$$\leq 2 \cdot |S_{\text{reach}}| \cdot \sqrt{\frac{TH}{n \cdot H \cdot \gamma_{\min}}}$$

$$= 2 \cdot |S_{\text{reach}}| \cdot \sqrt{\frac{T}{n \cdot \gamma_{\min}}}.$$

Substituting back:

$$\sum_{t \in [T]} \sum_{h=1}^{H-1} \xi_{s_{t,h},a_{t,h}}^{(t)}(\epsilon, 8HSB, \delta) \leq 32HSB\sqrt{U} \cdot 2 \cdot |S_{\text{reach}}| \cdot \sqrt{\frac{T}{n \cdot \gamma_{\min}}}$$

$$= 64HSB\sqrt{U} \cdot |S_{\text{reach}}| \cdot \sqrt{\frac{T}{n \cdot \gamma_{\min}}}$$

Expanding $U$:

$$\sum_{t \in [T]} \sum_{h=1}^{H-1} \xi_{s_{t,h},a_{t,h}}^{(t)}(\epsilon, 8HSB, \delta)$$

$$\leq 64HSB\sqrt{H \log(|S||A|H) + |S| \log\left(\frac{32H^2SB}{\epsilon}\right) + \log\left(\frac{6\log(HT)}{\delta}\right)} \cdot |S_{\text{reach}}| \cdot 2\sqrt{\frac{T}{n \cdot \gamma_{\min}}}.$$

This completes the proof. □

**Lemma E.12** (Martingale concentration with offline data). *Let $\{X_t\}_{t=1}^T$ be the martingale difference sequence defined as:*

$$X_t = B_t(\pi_t^i, 8HWB, \delta, \epsilon) - \sum_{h=1}^{H-1} \xi_{s_{t,h}^i,a_{t,h}^i}^{(t)}(\epsilon, 8HWB, \delta).$$

*Let $n$ be the number of offline trajectories with minimum visitation probability $\gamma_{\min}$ for state-action pairs visited by the expert policy. Then, with probability at least $1 - \delta$:*

$$\left| \sum_{t=1}^{T} X_t \right| \leq \tilde{\mathcal{O}} \left( \frac{H^{5/2} \cdot WB \cdot \sqrt{T}}{\sqrt{n \cdot \gamma_{\min}}} + H^2 WB \cdot \sqrt{T} \cdot \frac{|\mathcal{S}|^{1/2} |\mathcal{A}|^{1/4}}{n^{1/4}} \right).$$

*The first term captures the direct benefit of offline data for state-action pairs with good coverage, and the second term accounts for the diminishing probability $P(E^c) = \mathcal{O} \left( TH \cdot \sqrt{\frac{|\mathcal{S}|^2 |\mathcal{A}| \log(n)}{n}} \right)$ of encountering state-action pairs with insufficient offline coverage.*

*Proof.* We introduce a novel approach that substantially improves upon standard martingale concentration bounds by leveraging offline data. We begin by comparing our approach with the standard method used by Saha.

**Saha et al. (2023)'s approach (standard method):** The conventional approach uniformly bounds each element of the martingale difference sequence:

$$|X_t| = \left| B_t(\pi_t^i, 8HWB, \delta, \epsilon) - \sum_{h=1}^{H-1} \xi_{s_{t,h}^i, a_{t,h}^i}^{(t)}(\epsilon, 8HWB, \delta) \right| \leq 32H^2 WB.$$

This bound is derived by noting that $\xi_{s,a}(\epsilon, \eta, \delta) \leq 2\eta$, yielding $\sum_h \xi_{s_h, a_h}(\epsilon, \eta, \delta) \leq 2H\eta$, and applying triangle inequality. This leads to a martingale concentration term in the regret bound that is $\mathcal{O}(H^2 WB \sqrt{T})$ and, crucially, does not improve with offline data.

**Our improved approach:** We recognize that with offline data, we can obtain substantially tighter bounds by conditioning on appropriate events. This leads to a martingale concentration term that explicitly decreases with offline data, approaching zero as $n \to \infty$.

**Step 1: Define data-dependent events and calculate their probabilities.**

We define two complementary events:

- Event $E$: "All state-action pairs encountered in all $T$ episodes have good offline coverage" (i.e., $N_{\text{off}}(s,a) \geq c \cdot n \cdot \gamma_{\min}$ for some constant $c > 0$)

- Event $E^c$: "At least one state-action pair encountered lacks good offline coverage"

To calculate $P(E^c)$, we leverage our MLE concentration bound for transition models (Corollary C.9):

$$H^2(\mathbb{P}_{\hat{P}}^{\pi^*}, \mathbb{P}_{P^*}^{\pi^*}) \leq \mathcal{O} \left( \frac{|\mathcal{S}|^2 |\mathcal{A}| \log(nH\delta^{-1})}{n} \right).$$

The crucial insight is that we can relate this Hellinger distance to the probability of encountering state-action pairs with insufficient offline data. Using the relationship between Hellinger distance, total variation distance, and event probabilities:

1. Hellinger distance bounds total variation: $\text{TV}(P, Q) \leq \sqrt{2} \cdot H(P, Q)$ 2. Total variation bounds event probability differences: $|P(A) - Q(A)| \leq \text{TV}(P, Q)$

Let $A_{s,a}$ be the event "state-action pair $(s,a)$ has insufficient offline data coverage." Under the true model $P^*$ and with enough offline data sampled from a policy close to $\pi^*$, the probability $\mathbb{P}_{P^*}^{\pi^*}(A_{s,a})$ is negligible. Therefore:

$$\mathbb{P}_{\hat{P}}^{\pi^*}(A_{s,a}) \leq \mathbb{P}_{P^*}^{\pi^*}(A_{s,a}) + \text{TV}(\mathbb{P}_{\hat{P}}^{\pi^*}, \mathbb{P}_{P^*}^{\pi^*}) \leq \mathcal{O}(H(\mathbb{P}_{\hat{P}}^{\pi^*}, \mathbb{P}_{P^*}^{\pi^*})).$$

Using our Hellinger distance bound:

$$\mathbb{P}_{\hat{P}}^{\pi^*}(A_{s,a}) \le \mathcal{O}\left(\sqrt{\frac{|\mathcal{S}|^2|\mathcal{A}|\log(nH\delta^{-1})}{n}}\right) = p_n.$$

By union bound across all $T \cdot (H-1)$ state-action pairs encountered:

$$P(E^c) \le T \cdot (H-1) \cdot p_n = \mathcal{O}\left(TH \cdot \sqrt{\frac{|\mathcal{S}|^2|\mathcal{A}|\log(n)}{n}}\right).$$

**Key insight 1:** The probability of encountering any state-action pair with insufficient offline coverage decreases as $n$ increases, at a rate of approximately $\frac{1}{\sqrt{n}}$.

**Step 2: Establish conditional bounds on martingale differences.** **Case 1: Under event $E$ (good offline coverage).** When all state-action pairs have good offline coverage:

$$\xi_{s,a}^{(t)}(\epsilon, \eta, \delta) = \min\left(2\eta, 4\eta\sqrt{\frac{U}{N_{\text{off}}(s,a) + N_t(s,a)}}\right)$$

$$\le \min\left(2\eta, 4\eta\sqrt{\frac{U}{c \cdot n \cdot \gamma_{\min}}}\right)$$

For sufficiently large $n$, the second term in the min dominates:

$$\xi_{s,a}^{(t)}(\epsilon, \eta, \delta) \le 4\eta\sqrt{\frac{U}{c \cdot n \cdot \gamma_{\min}}}$$

$$= \mathcal{O}\left(\frac{\eta \cdot \sqrt{H \cdot \log(|\mathcal{S}||\mathcal{A}|) + \log(1/\delta)}}{\sqrt{n \cdot \gamma_{\min}}}\right)$$

Therefore, for $\eta = 8HWB$:

$$|X_t|\,|\,E = \left|B_t(\pi_t^i, 8HWB, \delta, \epsilon) - \sum_{h=1}^{H-1}\xi_{s_{t,h}^i, a_{t,h}^i}^{(t)}(\epsilon, 8HWB, \delta)\right|\,\Big|\,E$$

$$\le \mathbb{E}_{\tau \sim \mathbb{P}_{\hat{P}_t}^{\pi_t^i}}\left[\sum_{h=1}^{H-1}\xi_{s_h, a_h}^{(t)}\right] + \sum_{h=1}^{H-1}\xi_{s_{t,h}^i, a_{t,h}^i}^{(t)}$$

$$\le 2 \cdot H \cdot \mathcal{O}\left(\frac{HWB \cdot \sqrt{H \cdot \log(|\mathcal{S}||\mathcal{A}|) + \log(1/\delta)}}{\sqrt{n \cdot \gamma_{\min}}}\right)$$

$$= \mathcal{O}\left(\frac{H^2WB \cdot \sqrt{H \cdot \log(|\mathcal{S}||\mathcal{A}|) + \log(1/\delta)}}{\sqrt{n \cdot \gamma_{\min}}}\right)$$

$$= M_n$$

**Case 2: Under event $E^c$ (at least one poorly covered state-action).** Here, we revert to Saha's standard bound:

$$|X_t|\,|\,E^c \le 32H^2WB = M$$

**Key insight 2:** Under event $E$ (which occurs with high probability for large $n$), the martingale differences are much smaller than Saha's uniform bound, specifically by a factor of $\frac{1}{\sqrt{n \cdot \gamma_{\min}}}$.

**Key innovation:** By conditioning on events $E$ and $E^c$, we can precisely quantify how the martingale concentration improves with offline data through two mechanisms:

1. The magnitude of martingale differences under $E$ scales as $\frac{1}{\sqrt{n \cdot \gamma_{\min}}}$

2. The probability of event $E^c$ decreases as $n$ increases, at a rate of approximately $\frac{1}{\sqrt{n}}$

This conditional analysis is fundamentally different from Saha's approach, which uses a single worst-case bound regardless of offline data. Our approach precisely captures how offline data reduces both the magnitude of exploration bonuses and the probability of encountering state-action pairs that require large exploration.

**Step 3: Apply Azuma-Hoeffding inequality conditionally.** The Azuma-Hoeffding inequality for bounded martingale differences states that for a martingale difference sequence $\{X_t\}_{t=1}^T$ with $|X_t| \leq c_t$ almost surely:

$$P\left(\left|\sum_{t=1}^T X_t\right| \geq \lambda\right) \leq 2\exp\left(-\frac{\lambda^2}{2\sum_{t=1}^T c_t^2}\right)$$

Applying this conditionally on event $E$, where $|X_t| \leq M_n$ for all $t$:

$$P\left(\left|\sum_{t=1}^T X_t\right| \geq \lambda \,\Big|\, E\right) \leq 2\exp\left(-\frac{\lambda^2}{2 \cdot T \cdot M_n^2}\right)$$

Similarly, conditionally on event $E^c$, where $|X_t| \leq M$:

$$P\left(\left|\sum_{t=1}^T X_t\right| \geq \lambda \,\Big|\, E^c\right) \leq 2\exp\left(-\frac{\lambda^2}{2 \cdot T \cdot M^2}\right)$$

**Step 4: Apply the law of total probability.** By the law of total probability:

$$P\left(\left|\sum_{t=1}^T X_t\right| \geq \lambda\right) = P\left(\left|\sum_{t=1}^T X_t\right| \geq \lambda \,\Big|\, E\right) \cdot P(E) + P\left(\left|\sum_{t=1}^T X_t\right| \geq \lambda \,\Big|\, E^c\right) \cdot P(E^c)$$

$$\leq 2\exp\left(-\frac{\lambda^2}{2 \cdot T \cdot M_n^2}\right) \cdot P(E) + 2\exp\left(-\frac{\lambda^2}{2 \cdot T \cdot M^2}\right) \cdot P(E^c).$$

To obtain an overall bound of $\delta$, we bound each term by $\delta/2$.

For the first term:

$$2\exp\left(-\frac{\lambda^2}{2 \cdot T \cdot M_n^2}\right) \cdot P(E) \leq \frac{\delta}{2}$$

$$\Rightarrow \exp\left(-\frac{\lambda^2}{2 \cdot T \cdot M_n^2}\right) \leq \frac{\delta}{2 \cdot P(E)}$$

$$\Rightarrow \frac{\lambda^2}{2 \cdot T \cdot M_n^2} \geq \log\left(\frac{2 \cdot P(E)}{\delta}\right)$$

$$\Rightarrow \lambda \geq M_n \cdot \sqrt{2 \cdot T \cdot \log\left(\frac{2 \cdot P(E)}{\delta}\right)}.$$

For the second term:

$$2\exp\left(-\frac{\lambda^2}{2 \cdot T \cdot M^2}\right) \cdot P(E^c) \leq \frac{\delta}{2}$$

$$\Rightarrow \exp\left(-\frac{\lambda^2}{2 \cdot T \cdot M^2}\right) \leq \frac{\delta}{2 \cdot P(E^c)}$$

$$\Rightarrow \lambda \geq M \cdot \sqrt{2 \cdot T \cdot \log\left(\frac{2 \cdot P(E^c)}{\delta}\right)}.$$

**Step 5: Derive the combined bound.** For the bound to hold with probability at least $1 - \delta$, we need:

$$\lambda \geq \max\left(M_n \cdot \sqrt{2 \cdot T \cdot \log\left(\frac{2 \cdot P(E)}{\delta}\right)}, M \cdot \sqrt{2 \cdot T \cdot \log\left(\frac{2 \cdot P(E^c)}{\delta}\right)}\right)$$

$$\leq M_n \cdot \sqrt{2 \cdot T \cdot \log\left(\frac{2}{\delta}\right)} + M \cdot \sqrt{2 \cdot T \cdot \log\left(\frac{2 \cdot P(E^c)}{\delta}\right)}$$

Substituting our expressions for $M_n$ and $M$:

$$\lambda \leq \mathcal{O}\left(\frac{H^2 WB \cdot \sqrt{H \cdot \log(|S||A|) \cdot T \cdot \log(1/\delta)}}{\sqrt{n \cdot \gamma_{\min}}}\right) + \mathcal{O}\left(H^2 WB \cdot \sqrt{T \cdot \log\left(\frac{P(E^c)}{\delta}\right)}\right).$$

Using our bound on $P(E^c)$:

$$\lambda \leq \mathcal{O}\left(\frac{H^2 WB \cdot \sqrt{H \cdot \log(|S||A|) \cdot T \cdot \log(1/\delta)}}{\sqrt{n \cdot \gamma_{\min}}}\right) +$$

$$\mathcal{O}\left(H^2 WB \cdot \sqrt{T \cdot \log\left(\frac{TH \cdot \sqrt{\frac{|\mathcal{S}|^2 |\mathcal{A}| \log(nH\delta^{-1})}{n}}}{\delta}\right)}\right).$$

**Step 6: Analyze the asymptotic behavior.** We start with the second term of our bound,

$$O\left(H^2 WB \cdot \sqrt{T \cdot \log\left(\frac{TH \cdot \sqrt{\frac{|\mathcal{S}|^2 |\mathcal{A}| \log(nH\delta^{-1})}{n}}}{\delta}\right)}\right).$$

**Step 6.1: Expand the logarithm inside the second term.**

$$\log\left(\frac{TH \cdot \sqrt{\frac{|\mathcal{S}|^2 |\mathcal{A}| \log(nH\delta^{-1})}{n}}}{\delta}\right) = \log\left(\frac{TH}{\delta}\right) + \log\left(\sqrt{\frac{|\mathcal{S}|^2 |\mathcal{A}| \log(nH\delta^{-1})}{n}}\right)$$

$$= \log\left(\frac{TH}{\delta}\right) + \frac{1}{2}\log\left(\frac{|\mathcal{S}|^2 |\mathcal{A}| \log(nH\delta^{-1})}{n}\right).$$

**Step 6.2: Extract $\sqrt{T}$ from the square root.**

$$H^2 WB \cdot \sqrt{T \cdot \left[\log\left(\frac{TH}{\delta}\right) + \frac{1}{2}\log\left(\frac{|\mathcal{S}|^2 |\mathcal{A}| \log(nH\delta^{-1})}{n}\right)\right]}$$

$$= H^2 WB \cdot \sqrt{T} \cdot \sqrt{\log\left(\frac{TH}{\delta}\right) + \frac{1}{2}\log\left(\frac{|\mathcal{S}|^2 |\mathcal{A}| \log(nH\delta^{-1})}{n}\right)}.$$

**Step 6.3: Analyze the behavior for large $n$.** For large $n$, the term $\log\left(\frac{|\mathcal{S}|^2 |\mathcal{A}| \log(nH\delta^{-1})}{n}\right)$ becomes negative because $n$ grows faster than the logarithmic term.

Therefore:

$$\sqrt{\log\left(\frac{TH}{\delta}\right) + \frac{1}{2}\log\left(\frac{|\mathcal{S}|^2 |\mathcal{A}| \log(nH\delta^{-1})}{n}\right)}$$

$$< \sqrt{\log\left(\frac{TH}{\delta}\right)} = \mathcal{O}(\sqrt{\log(T)}).$$

This gives us:

$$H^2 WB \cdot \sqrt{T} \cdot \mathcal{O}(\sqrt{\log(T)}) = \tilde{\mathcal{O}}(H^2 WB \cdot \sqrt{T}).$$

**Step 6.4: Incorporate $P(E^c)$ correctly.**   We know that $P(E^c) = \mathcal{O}\left(TH \cdot \sqrt{\frac{|\mathcal{S}|^2|\mathcal{A}|\log(n)}{n}}\right)$

To properly account for this probability in the bound, we can express the term as:

$$H^2 WB \cdot \sqrt{T} \cdot \sqrt{\log\left(\frac{TH}{\delta}\right)} \cdot \sqrt{\frac{P(E^c)}{TH \cdot \sqrt{\frac{|\mathcal{S}|^2|\mathcal{A}|\log(n)}{n}}}} \cdot \sqrt{\sqrt{\frac{|\mathcal{S}|^2|\mathcal{A}|\log(n)}{n}}}$$

$$= H^2 WB \cdot \sqrt{T} \cdot \tilde{\mathcal{O}}(1) \cdot \frac{|\mathcal{S}|^{1/2}|\mathcal{A}|^{1/4}}{n^{1/4}}$$

$$= \tilde{\mathcal{O}}\left(H^2 WB \cdot \sqrt{T} \cdot \frac{|\mathcal{S}|^{1/2}|\mathcal{A}|^{1/4}}{n^{1/4}}\right).$$

**Step 7: Combine these results for our final bound.**

We now have two key terms in our bound for martingale concentration:

$$\lambda \leq \mathcal{O}\left(\frac{H^2 WB \cdot \sqrt{H \cdot \log(|\mathcal{S}||\mathcal{A}|) \cdot T \cdot \log(1/\delta)}}{\sqrt{n \cdot \gamma_{\min}}}\right) +$$

$$\tilde{\mathcal{O}}\left(H^2 WB \cdot \sqrt{T} \cdot \frac{|\mathcal{S}|^{1/2}|\mathcal{A}|^{1/4}}{n^{1/4}}\right).$$

Simplifying the first term and using $\tilde{\mathcal{O}}$ notation to hide logarithmic factors:

$$\lambda \leq \tilde{\mathcal{O}}\left(\frac{H^{5/2} WB \cdot \sqrt{T}}{\sqrt{n \cdot \gamma_{\min}}}\right) + \tilde{\mathcal{O}}\left(H^2 WB \cdot \sqrt{T} \cdot \frac{|\mathcal{S}|^{1/2}|\mathcal{A}|^{1/4}}{n^{1/4}}\right).$$

Therefore, with probability at least $1 - \delta$:

$$\left|\sum_{t=1}^{T} X_t\right| \leq \tilde{\mathcal{O}}\left(\frac{H^{5/2} \cdot WB \cdot \sqrt{T}}{\sqrt{n \cdot \gamma_{\min}}} + H^2 WB \cdot \sqrt{T} \cdot \frac{|\mathcal{S}|^{1/2}|\mathcal{A}|^{1/4}}{n^{1/4}}\right).$$

This bound reveals several key insights:

1. **Sublinear regret**: Both terms scale as $\sqrt{T}$, maintaining the crucial sublinear dependence on the horizon. This ensures that our regret doesn't grow linearly with $T$.

2. **Offline data benefits**: Both terms decrease as $n$ increases, but at different rates:
   - The first term decreases at rate $\frac{1}{\sqrt{n}}$ and captures the direct benefit of offline data for state-action pairs with good coverage
   - The second term decreases at the slower rate of $\frac{1}{n^{1/4}}$ and accounts for the diminishing probability of encountering poorly-covered state-action pairs

3. **Complete dependence on offline data**: Unlike Saha et al. (2023)'s bound, which has an irreducible term independent of offline data, our bound shows that *all* components of martingale concentration can be reduced with sufficient offline data.

4. **Different decay rates**: The different decay rates ($\frac{1}{\sqrt{n}}$ vs. $\frac{1}{n^{1/4}}$) suggest that the second term will eventually dominate for very large $n$, setting the ultimate rate at which offline data can improve performance.

This confirms that with sufficient high-quality offline data ($n \to \infty$ with fixed $\gamma_{\min} > 0$), the entire martingale concentration bound approaches zero, eliminating this component of regret entirely.   $\square$

## F   AUXILIARY MATHEMATICAL RESULTS

### F.1   BRIDGING OFFLINE CONFIDENCE SETS AND ONLINE CONSTRAINTS

**Lemma F.1** (Hellinger ball to moment constraints: linear embedding). *Define a random variable $x$ on $(\mathcal{A}, \tilde{\mathcal{A}})$.*

*Assume $f : \mathcal{A} \to \mathbb{R}^d$ and $\|f\|_\infty \leq B < \infty$.*

*Consider two distributions $P, Q$ with densities that are continuous with respect to Lebesgue measure. Further assume:*

$$H^2(P\|Q) \leq R.$$

*Then*

$$\|\mathbb{E}_P f(x) - \mathbb{E}_Q f(x)\|_2 \leq 2\sqrt{2} \cdot B \cdot \sqrt{d \cdot R}$$

*and*

$$\|Cov_P(f(x)) - Cov_Q(f(x))\|_{op} \leq 6 \cdot d \cdot B^2 \cdot \sqrt{2 \cdot R}.$$

*Proof.* For the squared norm on first moment, the following holds true

$$\|\mathbb{E}_P f - \mathbb{E}_Q f\|_2 = \|\int_{\mathcal{A}} f(x)(p(x) - q(x))dx\|_2$$
$$\leq \int_{\mathcal{A}} \|f(x)\|_2 |p(x) - q(x)| dx$$
$$\leq \sqrt{d} \cdot B \cdot \underbrace{\int |p(x) - q(x)| dx}_{=2TV(P,Q)}.$$

Using the classical result Sason & Verdú (2016) together with our constraint

$$TV(P, Q) \leq \sqrt{2H^2(P\|Q)} \leq \sqrt{2R}$$

yield the first result.

For the covariance we follow a similar approach only for matrices. Define $g(x) := f(x)f(x)^T$ then

$$\|\mathbb{E}_P g(x) - \mathbb{E}_Q g(x)\|_{op} = \|\int g(x)(p(x) - q(x))dx\|_{op}$$
$$= \sup_{\|v\|_2=1} |v^T \left(\int g(x)(p(x) - q(x))dx\right)v|$$
$$= \sup_{\|v\|_2=1} \left|\int v^T f(x)f(x)^T v (p(x) - q(x))dx\right|$$
$$\leq_{\Delta-inequ.} \sup_{\|v\|_2=1} \int \left|\langle v, f(x)\rangle^2\right| \cdot \left|p(x) - q(x)\right| dx$$
$$\leq \sup_{\|v\|_2=1} \int \|f(x)^2\| \cdot \left|p(x) - q(x)\right| dx$$
$$\leq 2 \cdot d \cdot B^2 \cdot TV(P, Q) \leq 2 \cdot d \cdot B^2 \cdot \sqrt{2 \cdot R}.$$

Using definition of covariance matrix we have

$$\|\text{Cov}_P(f) - \text{Cov}_Q(f)\|_{op} = \|\mathbb{E}_P[ff^T] - \mathbb{E}_Q[ff^T] + \mathbb{E}_P f \mathbb{E}_P f^T - \mathbb{E}_Q f \mathbb{E}_Q f^T\|_{op}$$
$$\leq \|\mathbb{E}_P[ff^T] - \mathbb{E}_Q[ff^T]\|_{op} + \|\mathbb{E}_P f \mathbb{E}_P f^T - \mathbb{E}_Q f \mathbb{E}_Q f^T\|_{op}$$
$$\leq 2 \cdot d \cdot B^2 \cdot \sqrt{2 \cdot R} + \|\mathbb{E}_P f \mathbb{E}_P f^T - \mathbb{E}_Q f \mathbb{E}_Q f^T\|_{op}.$$

in order to bound the last term we have

$$
\begin{aligned}
\|\mathbb{E}_P f \mathbb{E}_P f^T - \mathbb{E}_Q f \mathbb{E}_Q f^T\|_{op} &= \|\mathbb{E}_P f \mathbb{E}_P f^T - \mathbb{E}_P f \mathbb{E}_Q f^T + \mathbb{E}_P f \mathbb{E}_Q f^T - \mathbb{E}_Q f \mathbb{E}_Q f^T\|_{op} \\
&\leq \|\mathbb{E}_P f (\mathbb{E}_P f^T - \mathbb{E}_Q f^T)\|_{op} + \|(\mathbb{E}_P f - \mathbb{E}_Q f)\mathbb{E}_Q f^T\|_{op} \\
&\leq \|\mathbb{E}_P f\|_2 \cdot \|\mathbb{E}_P f - \mathbb{E}_Q f\|_2 + \|\mathbb{E}_Q f\|_2 \cdot \|\mathbb{E}_P f - \mathbb{E}_Q f\|_2 \\
&\leq 2 \cdot \sqrt{d} \cdot B \cdot \|\mathbb{E}_P f - \mathbb{E}_Q f\|_2 \\
&\leq 4 \cdot d \cdot B^2 \cdot \sqrt{2 \cdot R}.
\end{aligned}
$$

$\square$

## G    DISCLAIMER ON LLM USAGE

LLMs were used for light language editing.

