# OpenReview forum: "Fine-tuning Behavioral Cloning Policies with Preference‑Based Reinforcement Learning"
_ICLR.cc/2026/Conference — ICLR 2026 Poster_

### Official Review · Reviewer_g6Yh · 2025-10-28

**Soundness:** 3
**Presentation:** 3
**Contribution:** 3
**Rating:** 6
**Confidence:** 3

**Summary:**

This paper proposes a hybrid algorithm, BRIDGE, that combines offline behavioral cloning with online preference-based RL to address data scarcity and unsafe exploration challenges. The method first constructs a confidence set over policies using the Hellinger distance in trajectory distribution space, and then fine-tunes the policy online using preference feedback constrained to this confidence set. Theoretically, the paper establishes a regret bound that converges asymptotically to zero as the number of offline demonstrations increases, formally characterizing the relationship between offline data quantity and online learning efficiency. Empirical results on both discrete and continuous control environments show that BRIDGE outperforms offline BC and online PbRL baselines.

**Strengths:**

- The theoretical results clearly support the necessity of the hybrid algorithm. The provided regret bound is sharper and offers meaningful insight into how offline demonstration data improves online sample efficiency. While I did not verify the detailed proofs in the appendix, the proof sketch is mathematically sound and grounded in established theoretical tools.
- The writing and organization are clear, with a strong logical flow from motivation to theoretical results and experiments.

**Weaknesses:**

- Some experimental details are deferred to the appendix but are essential for understanding the results (see specific questions below).
- The computational complexity of the optimization step (line 7 in Algorithm 1) is unclear.

**Questions:**

- In Figure 2, the number of offline demonstrations used should be specified. Without this, the performance comparison with online PbRL (which starts without offline data) seems incomplete. Also, what policy does the online PbRL baseline start from at iteration 0?
- In Figure 2, why do the dotted lines (offline BC) lack CI? Is the offline dataset fixed across the 20 runs?
- In Figure 3, how is the number of policies quantified in continuous environments? Is the state-action space discretized?
- What motivates the use of the Hellinger distance over other divergence measures? Is it mainly for analytical convenience?

---

> ### Author Response · Authors · 2025-11-21
>
> We would like to thank you for your time and effort, and appreciate your positive, constructive feedback. We are pleased to hear that you found our theory result insightful and mathematically sound, and our writing and organization clear with strong logical flow. We have addressed your feedback in the revision (which we will upload later today), and would like to answer your question in detail below.
>
> **Computational complexity of the optimization (cf. Weakness 2):**
>
> The optimization step in line 7 of Algorithm 1 corresponds to selecting the policy in the class $\Pi_t$ that minimizes the empirical objective. In the finite-state and finite-action setting, $\Pi$ is finite, thus so are the filtered $\Pi^{offline}$ and $\Pi_t$, and this step reduces to evaluating the objective for each policy, which requires $O(∣\Pi_t|^2)$ computations. We have clarified this complexity in the revised manuscript. In the continuous control case, this exact optimization is not tractable, as the induced $\Pi$ is no longer finite. As we described in our answer above on “Continuous implementation vs discrete theory”, we construct a finite approximation of $\Pi$, which reduces theoretical complexity to the discrete case. Added complexity in continuous domains comes from simulating trajectories in more complex settings (e.g., MuJoCo).
> We would like to emphasize that most algorithms with theoretical guarantees in imitation learning [1] and preference-based RL [3,5] do not address this practical optimization challenge explicitly. For this reason, we view our practical implementation as an additional contribution. We now highlight this point more clearly in the experimental section of the revised manuscript.
>
> **PbRL as a baseline (cf. Question 1):**
>
> We thank the reviewer for their suggestion. We have added the number of offline demonstrations to the main Figure 2’s caption in the revision.
>
> The online baseline at online iteration 0, does not start from a single policy, but from an unrefined policy search set that consists of the full policy space $\Pi$. In contrast, BRIDGE starts from the confidence set $\Pi^{offline} \subseteq \Pi$, which it constructs as a Hellinger ball around the behavior cloned policy $\pi^{BC}$. In our practical implementations, since $\Pi$ becomes intractably large for MDPs larger than our StarMDP experiment, we’ve chosen as a tractable search set for PbRL a union of the BRIDGE confidence set (Hellinger ball around $\pi^{BC}$) and a large number of other policies. In the discrete implementation, the other policies are random, and in the continuous implementation, the other policies are obtained by adding Gaussian noise to the parameters of $\pi^{BC}$ (so giving some advantage also at the online PbRL algorithm).
>
> **Offline dataset dynamic or fixed across runs (cf. Question 2):**
>
> We construct the offline demonstration dataset freshly for each seed (and added a comment regarding this to the revision). For the BC regret (the dotted green line), we only show the mean for visual clarity.
>
> **Finite policy search set in continuous experiments (cf. Question 3):**
>
> We explain this in Appendix A.4.2, where we describe the practical implementation in detail for both discrete and continuous versions. We have added additional clarity to the revision. In the continuous implementation, the state-action space is not discretized; it is continuous. The policy search space is infinite, but for tractability, we construct a set of candidate policies consisting of
>
> - The BC policy $\pi^{BC}$
>
> - Policies obtained by adding Gaussian noise to the BC policy’s parameters
>
> BRIDGE then filters this set by distance to $\pi^{BC}$ to obtain its confidence set $\Pi^{offline}$. The online PbRL baseline adds an additional set of ‘other' policies obtained from adding even larger Gaussian noise to $\pi^{BC}$’s parameters. We saw, and decided against, one alternative to this approach: Generate a large set of truly random policies $\Pi^{rand}$, such that PbRL searches in the full $\Pi^{rand}$ and BRIDGE filters it to obtain $\Pi^{offline}$. We found that the space of functional/usable policies in e.g. MuJoCo tasks is tiny compared to $\Pi$, so we would have to use an extremely large amount of random policies, which would not be computationally tractable to search. Our approach guarantees that policies in the search space are not fully useless by being derived from $\pi^{BC}$, and through the two different levels of noise, allows generating candidates with a wide range of quality.
>
> Related work uses e.g. sampling-based approaches [2], though we decided on the one described above as it adheres closer to our theoretical framework.

---

> ### Author Response · Authors · 2025-11-21
>
> **Motivation for use of Hellinger distance (cf. Question 4):**
>
> The choice of Hellinger distance is architecturally necessary for:
>
> - Using [1]’s offline estimation directly
>
> - Maintaining the modular covering number framework
>
> - Enabling triangle inequality decomposition for confidence sets (e.g., proof of Lemma C.13)
>
> - Capturing trajectory-level rather than point-level statistics.
>
> Alternative (f-) divergences like KL, TV, $\chi^2$ would require finding workarounds, like translating [1]’s results with additional constants, or changing parts of our proof entirely, like the modular covering number structure, metric properties, and trajectory-level view.
>
> References:
>
> [1] Foster, D. J., Block, A., & Misra, D. (2024). Is behavior cloning all you need? understanding horizon in imitation learning. Advances in Neural Information Processing Systems, 37, 120602-120666.
>
> [2] Pace, A. et al. (2025). Preference Elicitation for Offline Reinforcement Learning. In The Thirteenth International Conference on Learning Representations.
>
> [3] Chen, X., Zhong, H., Yang, Z., Wang, Z., & Wang, L. (2022, June). Human-in-the-loop: Provably efficient preference-based reinforcement learning with general function approximation. In International Conference on Machine Learning (pp. 3773-3793). PMLR.
>
> [4] Ye, C., Xiong, W., Zhang, Y., Dong, H., Jiang, N., & Zhang, T. Online Iterative Reinforcement Learning from Human Feedback with General Preference Model. In The Thirty-eighth Annual Conference on Neural Information Processing Systems.
>
> [5] Drago, S., Mussi, M., & Metelli, A. M. (2025). Towards Theoretical Understanding of Sequential Decision Making with Preference Feedback. In 42nd International Conference on Machine Learning, ICML 2025 (pp. 1-16).
>
> [6] Aadirupa Saha, Aldo Pacchiano, and Jonathan Lee. Dueling rl: Reinforcement learning with trajectory preferences. In International Conference on Artificial Intelligence and Statistics, pp. 6263– 6289. PMLR, 2023.

---

### Official Review · Reviewer_UYkF · 2025-10-28

**Soundness:** 2
**Presentation:** 2
**Contribution:** 2
**Rating:** 4
**Confidence:** 2

**Summary:**

This paper addresses the challenge of applying reinforcement learning (RL) in real-world settings where specifying rewards is difficult and exploration is risky. The authors propose a two-stage framework: first, learn an initial policy using Behavioral Cloning (BC) from reward-free offline expert demonstrations; second, fine-tune this policy online using preference-based feedback. The paper introduces BRIDGE (Bounded Regret with Imitation Data and Guided Exploration), an algorithm designed for this "offline imitation + online preference fine-tuning" setting. The main contributions are:

1.  **The first theoretical framework** for analyzing this offline-to-online preference learning paradigm without ground-truth rewards during fine-tunin.
2.  **Regret bounds** (Theorem 4.1) that explicitly quantify how the amount of offline expert data ($n$) improves online sample efficiency, showing regret approaches zero as $n \rightarrow \infty$. The bounds rely on constructing confidence sets using the Hellinger distance between trajectory distributions, with radii shrinking as $O(1/\sqrt{n})$ (Theorem 4.2).
3.  **Empirical validation** of BRIDGE in discrete (StarMDP, Gridworld) and continuous control (MuJoCo Reacher, Ant) environments, showing lower cumulative regret compared to standalone BC and online preference-based RL (PbRL) baselines.

The core idea is to use offline data to construct a Hellinger ball confidence set around the BC policy, constraining the online preference learning phase to this smaller, safer region of the policy space.

**Strengths:**

**Novel Theoretical Contribution:** Provides the first rigorous theoretical framework and regret analysis for the widely used but previously theoretically unexplored paradigm of offline imitation learning followed by online preference-based fine-tuning.

**Clear Connection Between Offline Data and Online Efficiency:** The derived regret bounds (Theorem 4.1) explicitly and quantitatively show how the amount of offline data ($n$) reduces online regret, formally justifying the approach's sample efficiency benefits. The use of Hellinger distance confidence sets with radii shrinking as $O(1/\sqrt{n})$ provides a clear mechanism.

**Unified Algorithm (BRIDGE):** Proposes a concrete algorithm that integrates offline BC and online PbRL via uncertainty weighting and constrained exploration within the derived confidence set.

**Weaknesses:**

**Complexity of Bounds:** While insightful, the final regret bound in Theorem 4.1 and particularly the expanded version in Theorem E.1 are quite complex, making direct interpretation challenging. While the dependence on $n$ is clear, understanding the interplay of all MDP parameters ($S, A, H$), $\gamma_{min}$, $d$, $B$, $W$, etc., is less straightforward. The simplified analysis in Appendix B is helpful but doesn't cover the full algorithm.

**Computability of Hellinger Confidence Set:** Constructing the Hellinger confidence set $\Pi_{1-\delta}^{offline}$ involves estimating the Hellinger distance between the BC policy and candidate policies under the estimated dynamics $\hat{P}$. While Appendix A.4.3 provides an efficient calculation for the discrete case, the practical implementation for continuous environments resorts to an $L_2$ distance proxy in embedding space and constructive sampling around $\pi^{BC}$. This discrepancy between the theory (Hellinger distance over trajectory distributions) and continuous implementation requires further justification or analysis regarding its impact on the guarantees.

**Dependence on $\gamma_{min}$:** The bounds depend on $\gamma_{min}$, the minimum visitation probability of the expert policy. This value is unknown and hard to estimate. While Assumption 3 is milder than typical offline RL coverage assumptions, the practical implications of a potentially very small $\gamma_{min}$ (specialized expert) on the bounds could be discussed further.

**Choice of Embedding Function:** The experiments and Appendix A.1 highlight the critical impact of the trajectory embedding function $\phi$ on performance. The theory assumes $\phi$ is known, but provides little guidance on choosing or learning a good embedding, which remains a key practical challenge.

**Questions:**

**Questions**

**Continuous Implementation vs. Theory:** Could you elaborate on the theoretical implications of using the $L_2$ distance proxy in embedding space and the constructive sampling method for $\Pi^{offline}$ in the continuous experiments, compared to the Hellinger distance over trajectory distributions used in the theory? Does this approximation affect the validity of the regret bounds in the continuous setting?

**Sensitivity to $\gamma_{min}$:** How sensitive are the theoretical bounds and the practical performance of BRIDGE to the value of $\gamma_{min}$ (Assumption 3)? What happens if $\gamma_{min}$ is extremely small, corresponding to a highly specialized expert?

**Offline Data Quality:** The theory assumes offline data comes from an expert policy $\pi$. The ablation in Appendix A.1 briefly explores noisy data. Could you comment further on how the theoretical guarantees might degrade if the offline data is suboptimal or contains non-expert trajectories? Theorem 4.1 seems robust to some noise via $\gamma_{min}$ but assumes the *demonstrations* come from $\pi^*$.

**Computational Cost:** Calculating the Hellinger distance (even with the efficient method in Appendix A.4.3) and optimizing over the policy set $\Pi_t$ (Line 7, Algorithm 1) seems potentially computationally expensive, especially as the state/action space grows. Could you discuss the scalability of BRIDGE? The appendix mentions runtime but doesn't explicitly compare the overhead vs PbRL.

**Trajectory Embedding $\phi$:** Given the significant impact of the embedding function shown in the ablations, do you have insights into how one might systematically choose or even learn an effective $\phi$ in practical scenarios where the underlying reward/preference structure is unknown?

---

> ### Author Response · Authors · 2025-11-21
>
> We thank you for your detailed feedback and effort put into evaluating our paper. We are happy to read that you found our theoretical contributions novel and insightful. We have expanded clarity in the revision based on your feedback and would like to answer your questions in detail below (we will upload the updated version later in the day).
>
> **Improve clarity of dependence of regret bound on MDP parameters (cf. Weakness 1):**
>
> The regret bound we derived most importantly depends on the number of online iterations $T$ and offline demonstrations $n$. It also depends on the following constants which depend on the MDP and embedding function:
>
> - MDP properties $S,A,H$
>
> - The concentration coefficient $\gamma_\min$ (see answer below)
>
> - $d$, the dimension of the policy embedding
>
> - $B$ and $W$, the upper bound on the $L^2$-norm of trajectory embeddings and preference vectors
>
> Central to our theoretical contribution is the understanding that regret can be shrunk arbitrarily with increasing numbers of offline demonstrations $n$ as its dependence is $O(1/\sqrt{n})$, and that the dependence on the number of online iterations is the standard $O(\sqrt{T})$. We hope that this clarifies our regret bound.
>
>
> **Continuous implementation vs discrete Theory (cf. Weakness 2 & Question 1):**
>
> As this question seems related to reviewer LAgh’s question, our answer there could provide additional context.
>
> We view the continuous experiments as an empirical demonstration that the principles of our theory hold robustly even when approximations are necessary. Below, we provide theoretical motivation for the two specific approximations in our continuous implementation.
>
> - 1. Substituting the Hellinger distance in the trajectory distribution space with $L^2$ distance in the policy embedding space
>
> We use the Hellinger distance to create the confidence set by filtering the policy space $\Pi$ to only include policies $\pi$ that fulfill the relation $\sqrt{H^2(\mathbb{P}^{\pi}, \mathbb{P}^{\pi^{BC}})} \leq R$. In our continuous implementation, we instead use the $L^2$ metric as a practical surrogate, replacing the radius condition with $L^2(\phi(\pi), \phi(\pi^{BC}))$. Remember that $\phi(\pi) = E_{\tau \sim \pi}[\phi(\tau)]$ and that we assume bounded features $||\phi(\cdot)||_2 \leq B$. The theoretical link is:
> Hellinger distance controls total variation (TV) via $TV(\mathbb{P}^{\pi}, \mathbb{P}^{\pi^{BC}}) \leq \sqrt{2} H(\mathbb{P}^{\pi}, \mathbb{P}^{\pi^{BC}})$
> Total variation controls $L^2$ distance of policy embeddings, since we assume bounded embeddings: $||\phi(\pi) - \phi(\pi^{BC})||_2 \leq 2B \cdot TV(\mathbb{P}^{\pi}, \mathbb{P}^{\pi^{BC}}) \leq 2\sqrt{2}B \cdot R$.
> Then, if policies lie in a Hellinger ball (theory), they also lie in an $L^2$ ball in embedding space, which is what our implementation uses.
>
> - \2. Substituting the full policy search space $\Pi$ with a constructed subspace
>
> In discrete environments, we can iterate over a policy class $\Pi$ to filter it. In continuous control, $\Pi$ is infinite, and the dimensionality renders uniform rejection sampling computationally intractable. Randomly sampled policies in environments like Ant or Reacher are functionally useless and statistically extremely far from the expert distribution, such that an incomputably large set of random policies would be required to approximate $\Pi$. Our “constructive sampling” strategy is motivated by the theoretical requirement that the confidence set $\Pi^{offline}$ must contain the expert $\pi^\ast$ with high probability (Lemma 4.4). Since $\pi^{BC}$ is a consistent estimator of $\pi^\ast$, we assume $\pi^\ast$ lies in a local region around $\pi^{BC}$. We construct this region by applying Gaussian noise to $\pi^{BC}$ parameters. This ensures that our candidate set densely populates the area of $\Pi$ where $\pi^\ast$ is statistically likely to be located, rather than wasting samples on the irrelevant regions of the infinite policy space.
>
> To summarize, the continuous implementation is a practical approximation of our theoretical results.

---

> ### Author Response · Authors · 2025-11-21
>
> **Sensitivity to $\gamma_min$ (cf. Question 2):**
>
> We appreciate the reviewer’s comment. The dependence on the minimum visitation probability $\gamma_{\min}$​ arises naturally from concentrating statistical estimation error on states where expert support exists, and plays the same role as concentrability or coverage coefficients commonly seen in offline RL and imitation learning theory (e.g. [4,5] ). While $\gamma_{\min}$​​ is unknown and may indeed be small, depending on the transition model and the expert’s policy, this reflects a fundamental limitation of learning from demonstrations when certain states are rarely visited by the expert. From a practical perspective, this can be alleviated by (i) collecting demonstrations that span the relevant state regions, (ii) leveraging stochastic or suboptimal experts, or (iii) benefiting from the online preference-based phase, which provides additional exploration and corrective signals beyond expert coverage.
> To illustrate the effect of $\gamma_min$ on regret, we have added an ablation to the revision.
>
> **Sensitivity to offline data quality (cf. Question 3):**
>
> We thank the reviewer for the question, they are correct that our current theoretical analysis assumes that demonstrations are generated from the expert. However, our framework can theoretically and empirically accommodate suboptimal data through a bias-variance decomposition of the confidence set radius.
>
> Theoretical implications: expanding the Hellinger Ball. The core mechanism of BRIDGE is constructing the confidence set $\Pi^{offline}$ centered on the behavior cloned (BC) policy $\pi^{BC}$ with radius $R$. In the case that offline data comes from the expert, $R$ only accounts for estimation error (variance), scaling with $\mathcal{O}(1/\sqrt{n})$. If the data comes from the suboptimal policy $\pi^{sub}$, the distance to the optimal policy satisfies the triangle inequality $H(\pi^{BC}, \pi^\ast) \leq H(\pi^{BC}, \pi^{sub}) + H(\pi^{sub}, \pi^\ast)$. To preserve the guarantee that the optimal policy is in the confidence set, $\pi^\ast \in \Pi^{offline}$, the radius $R$ must be increased by the bias $H(\pi^{sub}, \pi^\ast)$.
>
> Effect on the \emph{search space}: the larger radius leads to a bigger confidence set $\Pi^{offline}$.
> Effect on the \emph{regret}: our regret bound (Thm 4.1) depends on the diameter of $\Pi^{offline}$. With more expert demonstrations, the diameter shrinks to 0 as $n \rightarrow \infty$. With suboptimal data, it floors at $H(\pi^{sub}, \pi^\ast)$. The online regret bound would still scale with $\sqrt(T)$, but the “acceleration” provided by offline data diminishes as the data quality degrades, and the algorithm’s performance approaches standard online preference learning as $\Pi^{offline} \approx \Pi$.
>
> Our ablation study in Appendix A.1 validates this behavior: Table 1 shows that as noise increases from 0 to 20%, at $n=1000$, the confidence set grows from 8% to 95.4% of the policy space. Figure 4 shows that we can manage this trade-off empirically via the radius. A larger radius, necessary to capture $\pi^\ast$ when data is noisy, results in higher cumulative regret compared to the tight radius used for the expert data. Importantly, it still outperforms the pure online baseline in regret and runtime by pruning the most irrelevant regions of the policy space.

---

> ### Author Response · Authors · 2025-11-21
>
> **Computational cost and scalability (cf. Question 4):**
>
> For calculating the Hellinger distance, we derive an efficient algorithm and analyze its complexity in Appendix A.4.3. The algorithm we derive (and use in the experiments) has computational complexity $O(\log H |S| \log_2^7)$, and memory complexity $O(|S|^2)$.
>
> For the complexity of optimizing over the policy set $\Pi_t$, we give the same answer as to reviewer g6Yh’s identical question: The optimization step in line 7 of Algorithm 1 corresponds to selecting the policy in the class $\Pi_t$ that minimizes the empirical objective. In the finite-state and finite-action setting, $\Pi$ is finite, thus so are the filtered $\Pi^{offline}$ and $\Pi_t$, and this step reduces to evaluating the objective for each policy, which requires $O(∣\Pi_t|^2)$ computations. We have clarified this complexity in the revised manuscript. In the continuous control case, this exact optimization is not tractable, as the induced $\Pi$ is no longer finite. As we describe in our answer above on “Continuous implementation vs discrete theory”, we construct a finite approximation of $\Pi$, which reduces theoretical complexity to the discrete case. Added complexity in continuous domains comes from simulating trajectories in more complex settings (e.g., MuJoCo).
>
> We would like to emphasize that most algorithms with theoretical guarantees in imitation learning [1] and preference-based RL [2,3] do not address these practical optimization challenges explicitly. For this reason, we view our practical implementation as an additional contribution.
>
> The online optimization step’s $O(|\Pi_t|^2)$ dominates computational complexity. As BRIDGE’s $\Pi_t \subseteq \Pi^{offline}$ is smaller than PbRL’s $\Pi_t \subseteq \Pi$ due to the behavior cloning-based confidence set approach, BRIDGE’s online part has a lower runtime than PbRL. These gains far outweigh the comparatively small complexity of BRIDGE’s added BC phase.
>
> **How to choose or learn embeddings in practice? (cf. Question 5):**
>
> We give a similar answer as to reviewer LAgh’s question here. As both [5] and our ablation (Appendix A.1, Fig. 6) show, the choice of policy embedding function has a big impact on learning performance. The better the policy embedding function captures the expert’s preference structure, the better the learning performance. In the case the reviewer mentions, where nothing is known about the expert’s preference structure, one has to resort to “naïve” policy embeddings. These perform worse than informed ones, but they can still work well enough, as our ablation in the Reacher environment (Fig. 6) shows. The naïve “average state-action” (which we use for the main paper’s plot in Fig 2) and “final state” embeddings both allow our algorithm to beat the behavioral cloning (BC) baseline in regret. Using the knowledge that the expert’s preferences are set to correspond to true environment reward, the embedding that replicates the Reacher environment’s rewards (and thus expert’s preferences) considerably improves performance with the algorithm converging near-instantly.
> As for learning embeddings, we see this as a natural extension of our work, as they could provide a much richer representation than the heuristic embeddings (e.g., state counts) used in our experiments. Since BRIDGE uses an offline dataset $\mathbb{D}$ for behavioral cloning, we can utilize this data to learn the embedding function $\phi$ before the online phase begins. We imagine one could apply self-supervised objectives [6] to $\mathbb{D}$.
> To add clarity to these points, we have expanded discussion on embeddings in the main body’s experiment section and the newly added section 4.4 on limitations & assumptions of BRIDGE, and we have adjusted Figure 2’s caption to include the embeddings used.

---

> ### Author Response · Authors · 2025-11-21
>
> References:
> [1] Foster, D. J., Block, A., & Misra, D. (2024). Is behavior cloning all you need? understanding horizon in imitation learning. Advances in Neural Information Processing Systems, 37, 120602-120666.
> [2] Chen, X., Zhong, H., Yang, Z., Wang, Z., & Wang, L. (2022, June). Human-in-the-loop: Provably efficient preference-based reinforcement learning with general function approximation. In International Conference on Machine Learning (pp. 3773-3793). PMLR.
>
> [3] Drago, S., Mussi, M., & Metelli, A. M. (2025). Towards Theoretical Understanding of Sequential Decision Making with Preference Feedback. In 42nd International Conference on Machine Learning, ICML 2025 (pp. 1-16).
>
> [4] Rashidinejad, P., Zhu, B., Ma, C., Jiao, J., & Russell, S. (2021). Bridging offline reinforcement learning and imitation learning: A tale of pessimism. Advances in Neural Information Processing Systems, 34, 11702-11716.
>
> [5] Xie, T., Jiang, N., Wang, H., Xiong, C., & Bai, Y. (2021). Policy finetuning: Bridging sample-efficient offline and online reinforcement learning. Advances in neural information processing systems, 34, 27395-27407.
>
> [6] Oord, A.v.d., Li, Y., Vinyals, O. (2018). Representation Learning with Contrastive Predictive Coding. arXiv preprint arXiv:180703748

---

### Official Review · Reviewer_LAhg · 2025-10-29

**Soundness:** 3
**Presentation:** 4
**Contribution:** 4
**Rating:** 8
**Confidence:** 3

**Summary:**

The paper "fine tuning behavioral cloning policies with preference-based RL" is a novel piece of work exploring the links between imitation learning and online pereference-based RL. The authors present a unified framework (BRIDGE) which combines these two - aiming to address the potential limitations of RL in domains where a) writing down appropriate reward functions is difficult and/or b) letting your agent perform open exploration would be risky or impractical.

Their key idea can be summarized as: firstly, one learns a "safe" warm-start policy from expert demonstrations only (no rewards), secondly you construct a "confidence set" in the space of policies based on this, and lastly the policy is updated using preference-based feedback which takes the form of a binary score between pairs of presented trajectories.

In Theorem 4.1 the authors provide the main theoretical result of the paper that offline data reduces online regret. They present empirical results on discrete environments (StarMDP and Gridworld) and continuous environments (Reacher and Ant) showing that their method achieves lower regret than two baselines.

**Strengths:**

The work is highly original and a very interesting idea. The paper's most significant contribution is as the first theoretical framework for studying the interplay from offline to online preference learning. The novel idea centres on using the Hellinger distance between distributions of trajectories to build the above mentioned confidence sets. It is the radii of these sets which the authors demonstrate is reduced with more offline data. They derive a non-trivial regret bound in terms of the amount of offline data and equivalent the radii of these sets.

The theoretical developments appear technically sound and well-motivated, although the proofs are extensive and largely relegated to the appendices. Due to time constraints I was not able to verify every step in detail.

The paper's presentation is very clear and well-written, with extensive detail. Each stage is logically introduced with clear figures and empirical data. There are extensive appendices.

**Weaknesses:**

An important point which is relegated to a footnote on page 6 is the applicability of the theoretical results. Section 3 introduces finite MDP settings before the theoretical results, and the footnote mentions that the results are presented for "tabular, stationary transitions here", but that the framework "readily adapts to other transition model classes". I think this could be made more clear, either by making this adaption more evident or presenting the scope of the theoretical results as distinct from the empirical ones (which include continuous control environments).

The embedding functions are clearly an important consideration and are shown to affect performance. The authors briefly touch on the limitations this could impose - a more detailed analysis of this could be insightful. For example, what are the consequences for more practical domains where the appropriate underlying structure may be unknown?

There are various assumptions made throughout the paper which are mostly introduced as standard to the field, but the implications or limitations of these are not drawn out in detail. Relatedly, a more general discussion of the limitations of the work is missing and would benefit the paper.

The majority of the theory sections are in the appendix. For accessibility and interest it might be nice to include a high-level proof summary. It could also be interesting to see a discussion comparing BRIDGE to modern RLHF or preference-based fine-tuning algorithms.

**Questions:**

- Related to the comment about the footnote and the theoretical result applicability, the confidence set construction and theoretical results often rely on the finiteness of S and A (for example, theorem 4.2 explicitly contains |S| and |A| terms. How are these results adapted to the continuous setting?
- Since the performance heavily depends on the choice of embedding function, have you thought about proposing ways to learn such embeddings from data?
- Have you considered the sensitivity of your algorithm to a noisy feedback signal?
- In practise, how should someone choose the confidence set radius or the number of offline trajectories?

---

> ### Author Response · Authors · 2025-11-21
>
> We appreciate the reviewer’s time and effort, and thank them for their constructive feedback. We are glad to hear that they found our idea highly original, our theory technically sound, and our presentation very clear and well-written. We have integrated their feedback into the revision (which will be uploaded later today). Below, we answer the raised questions in detail.
>
> **Continuous experiments & discrete theory (cf. Weakness 1 & Question 1):**
>
> We thank the reviewer for raising this important point. We have clarified the scope and applicability of our theoretical results in the revised manuscript (line 054 and Footnote 2).
>
> Our theoretical regret analysis is indeed developed under finite (tabular) state and action spaces and for general finite policy classes $\Pi$, as also done in [1]. This setting is standard in theoretical reinforcement learning [3,4], where finite classes are used as a clean test bed for understanding general function approximation: they impose no structural assumptions (such as linearity) that may lead to misleading conclusions when transitioning to more expressive model families. We have updated Footnote 2 to make this scope explicit.
>
> Regarding applicability beyond the tabular case, our framework does not rely on properties specific to tabular MDPs; rather, the main components (the offline construction of an uncertainty set over policies and the online selection rule) extend naturally to general transition models. What is currently missing for a full theoretical treatment in continuous or function-approximation settings is an analysis of the corresponding complexity measures (e.g., suitable $\varepsilon$-bracketing numbers). Developing such results is a promising direction for future work.
>
> Finally, while our theory is restricted to the finite setting, we highlight that the practical algorithm we evaluate is the natural extension of our framework to continuous control. Our empirical results, therefore, provide preliminary evidence that the method can work effectively beyond the regimes covered by our current theoretical guarantees.
>
> **Choosing or learning embedding functions (cf. Weakness 2, Question 2):**
>
> As both [5] and our ablation (Appendix A.1, Fig. 6) show, the choice of policy embedding function has a big impact on learning performance. The better the policy embedding function captures the expert’s preference structure, the better the learning performance. In the case the reviewer mentions, where nothing is known about the expert’s preference structure, one has to resort to “naïve” policy embeddings. To answer the reviewer’s question, naïve embeddings perform worse than informed ones, but they can still work well enough.
> Our ablation in the Reacher environment (Fig. 6) illustrates this. The naïve “average state-action” (which we use for the main paper’s plot in Fig 2) and “final state” embeddings both allow our algorithm to beat the behavioral cloning (BC) baseline in regret. Using the knowledge that the expert’s preferences are set to correspond to true environment reward, the embedding that replicates the Reacher environment’s rewards (and thus expert’s preferences) considerably improves performance with the algorithm converging near-instantly.
> As for learning embeddings, we see this as a natural extension of our work, as they could provide a much richer representation than the heuristic embeddings (e.g., state counts) used in our experiments. Since BRIDGE uses an offline dataset $\mathbb{D}$ for behavioral cloning, we can utilize this data to learn the embedding function $\phi$ before the online phase begins. We imagine one could apply self-supervised objectives [2] to $\mathbb{D}$.
> To address these points, we have expanded the discussion on embeddings in the main body’s experiment section & adjusted Figure 2’s caption to include the embeddings used.

---

> > ### Comment · Reviewer_LAhg · 2025-11-23
> >
> > Thank you very much for the detailed responses. The footnote and updated caption/ discussion around embeddings are much appreciated.

---

> ### Author Response · Authors · 2025-11-21
>
> **Discussion on limitations of work (cf. Weakness 3):**
>
> We agree that expanding the discussion of the limitations of our assumptions strengthens the paper, and have added the following comments in subsection 4.4 in the revision:
>
> - Linear preferences and bounded features: the chosen feature embedding must be sufficiently expressive to linearly capture the expert’s preferences. Cf. our answer to the preceding question.
>
> - Realizability and expert data: We assume the expert policy is realizable and that the offline data comes from the expert. As a result, the quality of the offline data is key to our algorithm’s performance. If the offline data is noisy/suboptimal, the resulting BC policy will be a poor center for our Hellinger ball and the confidence set may require a larger radius to contain the expert policy - or, worse, may not contain it at all with useful radii. We test this in our suboptimality ablation (Appendix A.1, Table 1). Results confirm that as data is noisier, the effectiveness of our offline filtering (search space reduction) decreases.
>
> - Expert policy concentration ($\gamma_{min}>0$): This assumption is weaker than comparable ones in the literature that require minimal offline data coverage across all state-action pairs ([6], Def’n 1). We only require that the expert's policy has a minimum visitation probability for the states it visits.  The dependence of regret on the minimum visitation probability $\gamma_{\min}$​ arises naturally from concentrating statistical estimation error on states where expert support exists
>
> **High-level proof summary (cf. Weakness 4)**
>
> This is a great suggestion. Fig. 9 at the start of Appendix B already gives such an overview, and for the revision, we have added a paragraph to the end of Section 4 that steps through the individual parts of constructing the regret bound.
>
> **Discussion comparing BRIDGE to modern RLHF/pref-based fine-tuning algorithms (cf. Weakness 4)**
>
> We agree that this would strengthen the paper further. We have added a discussion to the revision (in Appendix A.4). Our core idea can potentially be used with every RLHF/pref-based fine-tuning algorithm, and also potentially with DPO. All of these algorithms can potentially use our constraint on the policy space to improve the online sample efficiency. However, extending the theory to methods like DPO will require different techniques that could be an interesting future work direction.
>
> **Sensitivity to noisy preference feedback (cf. Question 3):**
>
> Noisy preference feedback increases the amount of online iterations needed to learn the preference model. We have ran an additional ablation, added to the revision in Appendix A.1, which shows that as feedback gets more noisy, convergence slows down and cumulative regret increases. It indicates that BRIDGE can handle moderate amounts of noise in the preference feedback (0-20%) before performance deteriorates significantly.
>
> **Choosing confidence set radius or number of offline trajectories (cf. Question 4):**
>
> The number of offline trajectories should be chosen as large as possible within available resources, since the regret scales as $O(1/\sqrt{n})$, where $n$ is the size of the offline dataset.
> For finite state–action settings, the confidence set radius can be selected following the expression derived in our theoretical analysis, which directly relates the radius to the concentration guarantees established for the finite policy class.
> In continuous-control environments, the confidence radius cannot be computed in closed form and instead needs to be tuned as a function of the offline dataset. In this case, its appropriate scale depends on factors such as the stochasticity of the environment and the variability of the expert demonstrations, both of which affect the estimation accuracy of the behavioral cloning policy. One could use a cross-validation-like approach: do a run of BRIDGE on a limited number of online queries for a number of plausible radii, then for the full run, select the radius that results in the best-performing policy on the limited runs.

---

> > ### Comment · Reviewer_LAhg · 2025-11-23
> >
> > This thorough and thoughtful response is very much appreciated. The limitations section is very clear, and responses to questions are very interesting - thank you for these!

---

> ### Author Response · Authors · 2025-11-21
>
> References:
>
> [1] Foster, D. J., Block, A., & Misra, D. (2024). Is behavior cloning all you need? understanding horizon in imitation learning. Advances in Neural Information Processing Systems, 37, 120602-120666.
>
> [2] Oord, A.v.d., Li, Y., Vinyals, O. (2018). Representation Learning with Contrastive Predictive Coding. arXiv preprint arXiv:180703748
>
> [3] Agarwal, A. et al. (2019). Reinforcement learning: Theory and algorithms. CS Dept., UW Seattle, Seattle, WA, USA, Tech. Rep, 32, 96.
>
> [4] Foster, D. J., & Rakhlin, A. (2023). Foundations of reinforcement learning and interactive decision making. arXiv preprint arXiv:2312.16730.
>
> [5] Pacchiano, A. et al. (2020). Learning to score behaviors for guided policy optimization. arXiv preprint arXiv:1906.04349.
>
> [6] Rashidinejad, P. et al. (2023). Bridging offline reinforcement learning and imitation learning: A tale of pessimism. arXiv preprint arXiv:2103.12021.
>
> [7] Ouyang, L. et al. (2022). Training language models to follow instructions with human feedback. arXiv preprint arXiv:2203.02155.
>
> [8] Rafailov, R. et al. (2024). Direct preference optimization: Your language model is secretly a reward model. arXiv preprint arXiv:2305.18290.
>
> [9] Pacchiano, A., Saha, A., Lee, J. (2023). Dueling RL: Reinforcement learning with trajectory preferences. arXiv preprint arXiv:2111.04850.

---

### Official Review · Reviewer_xVVw · 2025-11-01

**Soundness:** 4
**Presentation:** 4
**Contribution:** 3
**Rating:** 8
**Confidence:** 4

**Summary:**

The authors provide the first fundamental analysis of the offline-to-online approach and introduce BRIDGE, a unified algorithm that integrates two signals through an uncertainty-weighted objective function. They derive a regret bound that narrows as the number of offline demonstrations increases, thus explicitly linking the amount of offline data to the efficiency of online samples.

**Strengths:**

The theoretical part is excellent.

**Weaknesses:**

The experimental section could be more comprehensive.

**Questions:**

Have you considered experiments in other cases, or multi-scale scenarios?

---

> ### Author Response · Authors · 2025-11-21
>
> We thank the reviewer for the very positive review. We answer the question below.
>
> Our current evaluation already considers distinct problem classes, discrete tabular (GridWorld) and continuous control (MuJoCo) environments, showing that our method applies across qualitatively different MDP scales (changing offline dataset size, noise demonstrations, different embeddings). We think that exploring additional domains, for example, real-world robotic platforms, would be a valuable next step, especially given that preference-based fine-tuning naturally fits settings where reward specification is complex. We will incorporate this perspective into the discussion on future work, and we hope this aligns with the reviewer’s intended meaning of “multi-scale scenarios.”

---

### Author Response · Authors · 2025-12-03
**For the AC: Summary of feedback & discussion**

In light of the changed rebuttal period procedure, this comment is intended as a summary of the feedback and discussion for the AC. We’re thankful for the positive and constructive reviews and believe they’ve helped us improve the paper further in the revision.

## 1. Positive feedback
Reviewers found our work “highly original and a very interesting idea” (R2), our theory “technically sound and well-motivated” (R2), “excellent” (R1), and “novel” (R3), our presentation “very clear and well-written, with extensive detail” (R2), and our “writing and organization [...] clear, with a strong logical flow from motivation to theoretical results and experiments” (R4).

## 2. Reviewer points & our responses
We addressed all points the reviewers raised in detail. By the time responses were frozen, Reviewer 2 had replied, stating our response was “thorough and thoughtful”, and that our added Section 4.4 on limitations of our work was “very clear”.

### 2.1 Gap between discrete theory & continuous implementation (R2, R3, R4)
The reviewers noted that our theory contributions (i.e., regret bounds for our BRIDGE algorithm) rely on Hellinger distance sets in policy distribution space for finite policy spaces in finite MDPs, while experiments are done for both finite and continuous MDPs, and the continuous implementation instead uses $L^2$ distance sets in feature embedding space for infinite policy spaces.

Response: we clarified that the $L^2$ metric used in continuous experiments is a valid proxy derived from the Hellinger distance via Total Variation bounds. We justified “constructive sampling” (adding noise to $\pi^{BC}$) as a tractable approximation for the infinite policy space. We expanded discussion of this in the revision in Section 4 and the appendix.

### 2.2 Robustness to noisy feedback & data and choice of embeddings (R2, R3)
Reviewers asked about our algorithm’s theoretical and empirical robustness to suboptimal offline data, low expert coverage ($\gamma_{min}$) and choice of embedding function.

Response: we added new ablation studies to appendix A.1.
* Noise: ablations show BRIDGE degrades gracefully with noisy feedback and data and maintains lower regret than the baseline.
* Embeddings: naïve embeddings (e.g., average state) outperform the baseline, though informed embeddings perform best.
* Expert coverage: we clarify that our $\gamma_{min}$ is a weaker assumption than usual minimum expert coverage assumptions in the literature, and verify in the ablation that regret increases for smaller $\gamma_{min}$.

### 2.3 Computational complexity (R3, R4)
Reviewers asked about the complexity of our algorithm’s optimization step (Algorithm 1, line 7) and the Hellinger distance calculation.

Response: we provided the complexity analysis ($O(|\Pi_t|^2)$). We clarified that BRIDGE is computationally faster than the pure PbRL baseline because the confidence set $\Pi^{offline}$ significantly prunes the search space for the online learning stage compared to the baseline’s full policy space $\Pi$.

### 2.4 Requests for clarification (R1, R2, R3, R4)
Reviewers requested clarification on several points, which we all addressed:
* Applicability of our approach to domains of various scales (R1): we clarified that our experiments cover a range of MDP scales with tabular MDPs and continuous control (MuJoCo), and that real-world robot environments would be a desirable next step.
* Limitations of our approach & assumptions (R2): we’ve added a new Section 4.5 to the revision discussing this.
* Providing a high-level proof summary (R2): we’ve added a new Section 4.4 to the revision giving intuition and a step-by-step sketch of the main theorem’s proof.
* Dependence of our regret bound on MDP parameters (R3): we explained that the main dependence of the bound is on the number of offline data $n$ (i.e., $O(1/\sqrt{n})$) and online iterations $T$ (i.e., $O(\sqrt{T})$), and that all other variables appearing in the bound can be regarded as constants determined by the given MDP. This point is addressed several times in the main body (L62-67, L211-214, L429-431).
* Details on experiment setup (R4): we’ve addressed R4’s questions on our experiment setup and added that information to the revision to clarify.

## 3. Conclusion
We believe we have addressed all technical points regarding the theory-practice differential and robustness with explanations and ablations, and strengthened the paper further by adding to the revision requested sections on the proof sketch and limitations of our work, as well as numerous small additions to improve clarity. Given the strong endorsements from R1 and R2, and having addressed all of R3 and R4’s concerns and questions, we hope the specific merits of BRIDGE, as the “first rigorous theoretical framework and regret analysis for the widely used but previously theoretically unexplored paradigm of offline imitation learning followed by online preference-based fine-tuning” (R3), warrant acceptance.

---

### Meta-Review · Area_Chair_8Ed6 · 2025-12-29

**Summary:**

This paper explored fine tuning behavioral cloning policies with preference-based RL, with the key idea outlined  as follows: firstly, one learns a warm-start policy from expert demonstrations), and furthermore,   a "confidence set" is constructed  in the space of policies based on this; then, the policy is updated using online preference-based feedback which takes the form of a binary score between pairs of presented trajectories.

One distinction in the problem formulation is that it considers a reward-free setting, which  is distinct from related work that assumes access to a ground-truth reward signal for online fine-tuning.

**Reviewer Concerns:**

The reviewers are generally positive about the paper, and agree that the theoretic results are well motivated and  technically sound.  One review (R1) is by a  MS student, who did not participated in the discussion,  has no substance, so it can be largely ignored.

 There are two main concerns  in the reviews:1) the discrepancy between discrete theory & continuous implementation (R2, R3, R4); 2) the impact of the quality of demonstration data and noisy feedback and the choice of embeddings. The authors made effort in addressing these concerns.  I believe that the assumption with known embedding function is definitely too strong, and deserves further investigation.

**Reviewer Scores:**

The scores are 8/8/4/6.  The first socre 8 is by a  MS student, who did not participated in the discussion, and it has no substance, so can be largely ignored.

---

### Decision · Program_Chairs · 2026-01-26

Accept (Poster)